# MDGAs are fast-diffusing molecules that delay excitatory synapse development by altering neuroligin behavior

Andrea Toledo[1], Mathieu Letellier[1†], Giorgia Bimbi[1†], Béatrice Tessier[1], Sophie Daburon[1], Alexandre Favereaux[1], Ingrid Chamma[1], Kristel Vennekens[2], Jeroen Vanderlinden[2], Matthieu Sainlos[1], Joris de Wit[2], Daniel Choquet[1,3*‡], Olivier Thoumine[1*‡]

[1]University of Bordeaux, CNRS UMR 5297, Interdisciplinary Institute for Neuroscience, Bordeaux, France; [2]VIB Center for Brain & Disease Research and KU Leuven, Department of Neurosciences, Leuven Brain Institute, Leuven, Belgium; [3]University of Bordeaux, CNRS UAR 3420, INSERM, Bordeaux Imaging Center, Bordeaux, France

*For correspondence:
daniel.choquet@u-bordeaux.fr
(DC);
olivier.thoumine@u-bordeaux.
fr (OT)

†These authors contributed
equally to this work
‡These authors are co-senior
authors to this work

Competing interest: The authors
declare that no competing
interests exist.

Reviewing Editor: Gary L
Westbrook, Oregon Health and
Science University, United States

**Abstract** MDGA molecules can bind neuroligins and interfere with trans-synaptic interactions to neurexins, thereby impairing synapse development. However, the subcellular localization and dynamics of MDGAs, or their specific action mode in neurons remain unclear. Here, surface immunostaining of endogenous MDGAs and single molecule tracking of recombinant MDGAs in dissociated hippocampal neurons reveal that MDGAs are homogeneously distributed and exhibit fast membrane diffusion, with a small reduction in mobility across neuronal maturation. Knocking-down/out MDGAs using shRNAs and CRISPR/Cas9 strategies increases the density of excitatory synapses, the membrane confinement of neuroligin-1, and the phosphotyrosine level of neuroligins associated with excitatory post-synaptic differentiation. Finally, MDGA silencing reduces the mobility of AMPA receptors, increases the frequency of miniature EPSCs (but not IPSCs), and selectively enhances evoked AMPA-receptor-mediated EPSCs in CA1 pyramidal neurons. Overall, our results support a mechanism by which interactions between MDGAs and neuroligin-1 delays the assembly of functional excitatory synapses containing AMPA receptors.

## Editor's evaluation

The authors used immunostaining and single-molecule tracking analyses in cultured hippocampal neurons to address some unresolved issues on MDGA molecules that are regarded as negative regulators of synapse development. MDGAs were highly mobile and homogenously distributed in cultured neurons with localization and dynamics of NLGN1 and GluA2 altered upon loss of MDGA2.

## Introduction

During brain development, synapse assembly and maturation are critical processes involving several families of adhesion molecules, among which the neurexin-neuroligin complex has been one of the most extensively studied (*Bemben et al., 2015b*; *Chanda et al., 2017*; *Letellier et al., 2018*; *Südhof, 2017*; *Wu et al., 2019*). Neuroligins (NLGNs) are post-synaptic proteins that comprise four members in rodents (NLGN1-4), NLGN1 being primarily localized at excitatory synapses, NLGN2 and NLGN4 at inhibitory synapses, and NLGN3 at both types of synapses (*Budreck and Scheiffele, 2007*; *Varoqueaux et al., 2004*). At the structural level, NLGNs form both homo- and hetero-dimers

through *cis*-interactions between their acetylcholinesterase (AchE)-like domains (*Araç et al., 2007*; *Chen et al., 2008*; *Dean et al., 2003*; *Fabrichny et al., 2007*; *Poulopoulos et al., 2012*). NLGNs also contain a stalk region that can be cleaved by proteases (*Peixoto et al., 2012*; *Suzuki et al., 2012*), a single pass transmembrane domain, and a conserved intracellular tail whose binding to post-synaptic scaffolding molecules can be modulated by phosphorylation and thereby influence AMPA receptor (AMPAR) recruitment (*Antonelli et al., 2014*; *Bemben et al., 2014*; *Bemben et al., 2015a*; *Giannone et al., 2013*; *Letellier et al., 2020*; *Letellier et al., 2018*; *Poulopoulos et al., 2009*). Post-synaptic NLGNs bind pre-synaptic neurexins (NRXNs) through extracellular interactions, thus making a bridge between sub-micron adhesive modules across the synaptic cleft that precisely position AMPARs (*Chamma et al., 2016a*; *Haas et al., 2018*; *Trotter et al., 2019*).

Apart from NRXNs, few direct NLGN binding partners have been identified (*Südhof, 2017*). Recently, MAM-domain GPI-anchored molecules (MDGAs) were reported to bind in cis to NLGNs with high affinity and compete with their binding to NRXNs (*Connor et al., 2019*). In the co-culture assay, the expression of MDGAs together with NLGNs in heterologous cells impairs the synapse-inducing activity of NLGNs on contacting axons (*Elegheert et al., 2017*; *Lee et al., 2013*; *Pettem et al., 2013*). Crystal structures of MDGA-NLGN complexes revealed that MDGAs bind through their first Ig1-Ig2 domains to the two lobes of the NLGN extracellular dimer, at sites which overlap with the NRXN binding interface (*Elegheert et al., 2017*; *Gangwar et al., 2017*; *Kim et al., 2017*). Manip-ulations of MDGA1 protein levels by over-expression (OE), knock-down (KD), or knock-out (KO) in neurons have led to the common view that MDGA1 selectively inhibits inhibitory synapse formation by primarily repressing NLGN2-NRXN interactions (*Connor et al., 2017*; *Lee et al., 2013*; *Loh et al., 2016*; *Pettem et al., 2013*). Similar studies performed on MDGA2 have led to more debated results, i.e. some reports showing that MDGA2 KO specifically inhibits excitatory synapse formation in vivo (*Connor et al., 2016*), while others showing no effect of MDGA2 KD on either excitatory or inhibitory synapses in culture (*Loh et al., 2016*).

Notwithstanding the foregoing, there are still a number of critical questions that need to be answered in order to get a more complete picture of the role of MDGAs in synapse differentiation and function (*Connor et al., 2019*; *Thoumine and Marchot, 2017*). 1 / What is the surface dynamics and nanoscale localization of endogenous MDGAs at the neuronal membrane? Indeed, in the absence of highly specific and efficient antibodies to MDGAs allowing reliable immunostaining in tissue, over-expression approaches have yielded contrasting results about the presence of MDGA1 and MDGA2 at excitatory versus inhibitory synapses (*Loh et al., 2016*; *Pettem et al., 2013*). Given the absence of an intracellular domain with potential synaptic retention motifs, MDGAs are in fact expected to display fast membrane diffusion and not accumulate at synapses. 2 / Considering that the binding of MDGAs and NRXNs to NLGNs is mutually exclusive, what is the effect of MDGAs on the dynamic distribution of NLGN in dendrites and on the NLGN-dependent phosphotyrosine signaling pathway known to regulate post-synaptic differentiation (*Giannone et al., 2013*; *Letellier et al., 2018*)? 3 / Consequently, what is the impact of MDGAs on AMPAR surface dynamics and synaptic function, which were shown to be tightly regulated by NLGN1 (*Czöndör et al., 2013*; *Haas et al., 2018*; *Letellier et al., 2020*; *Mondin et al., 2011*)?

To address those questions, we examined the surface localization and dynamics of MDGAs, using both custom-made antibodies to endogenous MDGA1 as well as replacement strategies with recom-binant MDGAs bearing small tags and labelled with monomeric fluorescent probes. Using a combi-nation of single-molecule imaging and electrophysiology, we also assessed the effects of single-cell MDGA knock-down or knock-out on NLGN1 and AMPAR membrane diffusion and localization in relation to synapse maturation, as well as on synaptic transmission. Finally, we examined the impact of MDGA knock-down on endogenous NLGN phosphotyrosine levels by biochemistry. Together, our data indicate that MDGAs are highly mobile and homogeneously distributed molecules, that alter both NLGN1 and AMPAR dynamics, localization, and function, thereby significantly delaying the differentiation of excitatory post-synapses.

## Results

### Endogenous MDGA1 is homogeneously distributed in hippocampal pyramidal neurons

To examine the localization of endogenous MDGAs in neurons, we produced and purified full-length recombinant Fc-tagged MDGA1 and MDGA2, and custom-ordered the generation of rabbit polyclonal antibodies against those proteins. We then characterized the collected antisera using immunohistochemistry and western blots. The MDGA1 antiserum recognized recombinant HA-MDGA1 (but not HA-MDGA2) extracted from HEK-293T cells as a 130 kDa band on immunoblots (*Figure 1A*), above the molecular weight of 101 kDa expected from the amino-acid sequence, suggesting glycosylation of the protein i.e. through the addition of N-linked sugar chains. The reactivity to MDGA1 was abolished by pre-incubation of the antiserum with an excess of recombinant MDGA1-Fc antigen (*Figure 1B*). The MDGA1 antiserum recognized a single band around 130 kDa in brain homogenates from wild-type mice, which was not present in brain homogenates from *Mdga1* KO mice, demonstrating antibody specificity (*Figure 1C*). In some samples, the MDGA1 antibody recognized a doublet of bands, which may suggest a differential glycosylation pattern seemingly regulated across neuronal development (*Figure 1—figure supplement 1C*). Immunohistochemistry on brain sections showed abundant MDGA1 localization in the hippocampus, with prominent labeling in CA3 and CA1 stratum radiatum and stratum oriens containing pyramidal neuron dendrites (*Figure 1D*). MDGA1 staining was absent in brain sections from *Mdga1* KO mice (*Ishikawa et al., 2011*), showing antibody specificity in tissue. MDGA1 was detected both in pre-synaptic and post-synaptic fractions from synaptosome preparations, revealing its presence in synaptic compartments (*Figure 1E*). Unfortunately, the MDGA2 antiserum was not specific enough to be further used. However, we detected abundant levels of MDGA2 mRNAs by RT-qPCR in hippocampal cultures (*Figure 1—figure supplement 1A, B*), in agreement with previous in situ hybridization and β-galactosidase staining (*Connor et al., 2016*; *Lee et al., 2013*), together suggesting that the MDGA2 protein is also expressed. Interestingly, we also detected mRNAs for both MDGA1 and MDGA2 in astrocyte cultures (*Figure 1—figure supplement 1A, B*).

We then examined the sub-cellular surface distribution of endogenous MDGA1 in dissociated rat hippocampal cultures at different developmental stages (DIV 7, 14, 21), by performing live staining of neurons with MDGA1 antiserum before fixation and counter immuno-labelling of either MAP-2 as a dendritic marker, excitatory pre- and post-synaptic proteins VGLUT1 and PSD-95, or inhibitory pre- and post-synaptic markers VGAT and gephyrin, respectively (*Figure 1F, I*). Live labelling with the MDGA1 antibody was first tested in COS-7 cells expressing recombinant MDGA1 or MDGA2 molecules. Strong surface staining with the MDGA1 antiserum was observed in cells expressing MDGA1, but not in cells expressing MDGA2, validating this application and demonstrating no cross-reactivity of the antibody (*Figure 1—figure supplement 2*). In neurons, the MDGA1 staining revealed many sub-micron clusters, most likely a consequence of artifactual MDGA1 aggregation due to live incubation with the divalent polyclonal antibody (*Chamma et al., 2016a*). Those small MDGA1 puncta were distributed all over the dendritic shaft, with relatively constant total fluorescence intensity and cluster area over the developmental period analyzed (*Figure 1—figure supplement 3*), and a small decrease in MDGA1 cluster intensity at DIV 21, in accordance to an overall decrease of total cellular MDGA1 level at later time points, as shown by RT-qPCR and western blot analyses (*Figure 1—figure supplement 1A-D*). MDGA1 clusters were present, but not particularly enriched, at excitatory or inhibitory synapses. Quantitatively, the fraction of excitatory post-synapses containing MDGA1 clusters was 45% and 40% at DIV 7 and 14, respectively, and decreased to 25% at DIV 21 (*Figure 1G*), while the proportion of inhibitory post-synapses containing MDGA1 was 20% at DIV 10 and increased to 30% at DIV 14 and 21 (*Figure 1J*), suggesting that MDGA1 partially redistributes from excitatory to inhibitory synapses upon neuronal maturation. Overall, among synapses that contained MDGA1, the area overlap between PSD-95 and MDGA1, or between gephyrin and MDGA1 was around 15–30% (*Figure 1H and K*), pointing to a minor occupancy of both excitatory and inhibitory synapses by MDGA1.

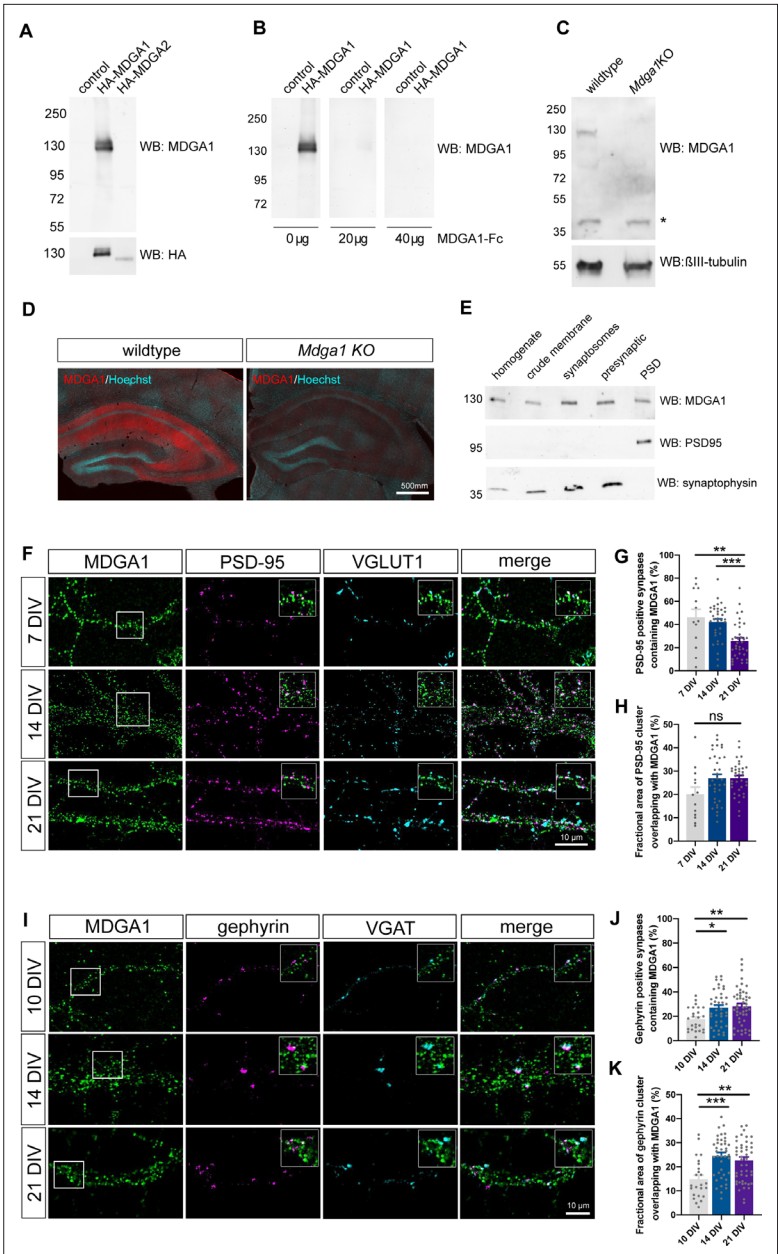

**Figure 1.** Validation of MDGA1 antibody and distribution of endogenous MDGA1 in brain slices and dissociated hippocampal cultures. (**A**) MDGA1 antiserum recognizes recombinant HA-MDGA1, but not HA-MDGA2, transiently expressed in HEK-293T cells (top membrane). Mock-transfected HEK-293T cells were used as controls. Anti-HA antibody labels both HA-MDGA1 and HA-MDGA2 (bottom membrane). Molecular weight markers in kDa indicated on the left. (**B**) Competition with different amounts (0, 20, and 40 μg) of excess recombinant MDGA1-Fc blocks detection of HA-MDGA1 by MDGA1 antiserum. (**C**) MDGA1 antiserum detects a single 130 kDa band in brain homogenate from wild-type mice, which was absent in brain homogenate from *Mdga1* KO mice (top membrane). Asterisk indicates non-specific band. ßIII-tubulin was used as loading control (bottom membrane). (**D**) Immunohistochemistry with MDGA1 antiserum (red) reveals strong immunoreactivity in CA3 and CA1 regions of the hippocampus in wild-type adult mice, which was absent in *Mdga1* KO mice. Nuclear marker Hoechst (cyan) was used to visualize tissue architecture. (**E**) Rat brain subcellular fractionation probed for MDGA1, postsynaptic excitatory marker PSD-95, and presynaptic marker synaptophysin. PSD: postsynaptic density. For original immunoblot images presented in panels (**A,B,C,E**), refer to *Figure 1—source data 1–7*. (**F, I**) Representative confocal images of dendritic segments from dissociated hippocampal neurons at different times in culture (7, 14, and 21 DIV) that were immunolabeled with MDGA1 antibody, and counterstained for either PSD-95 and VGLUT1 (**F**), or gephyrin and VGAT (**I**). (**G,H,J,K**) Quantification of the co-localization level and area overlap between

*Figure 1 continued on next page*

*Figure 1 continued*

endogenous MDGA1 and the excitatory post-synaptic marker PSD-95 (**G, H**), or the inhibitory post-synaptic marker gephyrin (**J, K**) as a function of time in culture. Data represent mean ± SEM of n > 13 neurons for all conditions and from three independent experiments, and were compared by a Kruskal–Wallis test followed by Dunn's multiple comparison test (*p < 0.05; **p < 0.01; ***p < 0.001). For the statistics of the data presented in panels (**G,H,J,K**), see *Supplementary file 1* and *Figure 1—source data 8*.

The online version of this article includes the following source data and figure supplement(s) for figure 1:

**Source data 1.** Source image of anti-MDGA1 and anti-HA immunoblots related to *Figure 1A*.

**Source data 2.** Source image of anti-MDGA1 immunoblot related to *Figure 1B*.

**Source data 3.** Source image of anti-MDGA1 immunoblot related to *Figure 1C*.

**Source data 4.** Source image of anti-βIII tubulin immunoblot related to *Figure 1C*.

**Source data 5.** Source image of anti-MDGA1 immunoblot related to *Figure 1E*.

**Source data 6.** Source image of anti-synaptophysin immunoblot related to *Figure 1E*.

**Source data 7.** PDF file showing all the immunoblots in *Figure 1* where the relevant bands chosen for illustration are highlighted by red rectangles.

**Source data 8.** Excel file containing all raw data and statistical tests used in *Figure 1*.

**Figure supplement 1.** RT-qPCR evaluation of MDGA1 and MDGA2 mRNA expression levels and western blot evaluation of protein expression during in vitro differentiation of hippocampal neurons and astrocytes.

**Figure supplement 1—source data 1.** Source image of anti-NLGN1 immunoblot related to *Figure 1—figure supplement 1C*.

**Figure supplement 1—source data 2.** Source image of anti-NLGN2 immunoblot related to *Figure 1—figure supplement 1C*.

**Figure supplement 1—source data 3.** Source image of anti-NLGN3 immunoblot related to *Figure 1—figure supplement 1C*.

**Figure supplement 1—source data 4.** Source image of anti-PSD-95 immunoblot related to *Figure 1—figure supplement 1C*.

**Figure supplement 1—source data 5.** Source image of anti-gephyrin immunoblot related to *Figure 1—figure supplement 1C*.

**Figure supplement 1—source data 6.** Source image of anti-GluA1 immunoblot related to *Figure 1—figure supplement 1C*.

**Figure supplement 1—source data 7.** Source image of anti-GluA2 immunoblot related to *Figure 1—figure supplement 1C*.

**Figure supplement 1—source data 8.** Source image of anti-MDGA1 immunoblot related to *Figure 1—figure supplement 1C*.

**Figure supplement 1—source data 9.** Source image of anti-actin immunoblot related to *Figure 1—figure supplement 1C*.

**Figure supplement 1—source data 10.** PDF file showing all the immunoblots in *Figure 1—figure supplement 1C* where the relevant bands chosen for illustration are highlighted by red rectangles.

**Figure supplement 1—source data 11.** Excel file containing all raw data and statistical tests used in *Figure 1—figure supplement 1*.

**Figure supplement 2.** Surface labeling of COS-7 cells expressing recombinant MDGA1 or MDGA2 with the MDGA1 antiserum.

**Figure supplement 3.** Surface expression of endogenous MDGA1.

**Figure supplement 3—source data 1.** Excel file containing all raw data and statistical tests used in *Figure 1—figure supplement 3*.

## Recombinant MDGA1 and MDGA2 are homogeneously distributed in dendrites at the nanoscale level

Next, we examined the nanoscale membrane organization of MDGAs using super-resolution microscopy (*Figure 2*). The formation of small endogenous MDGA1 aggregates observed upon live antibody

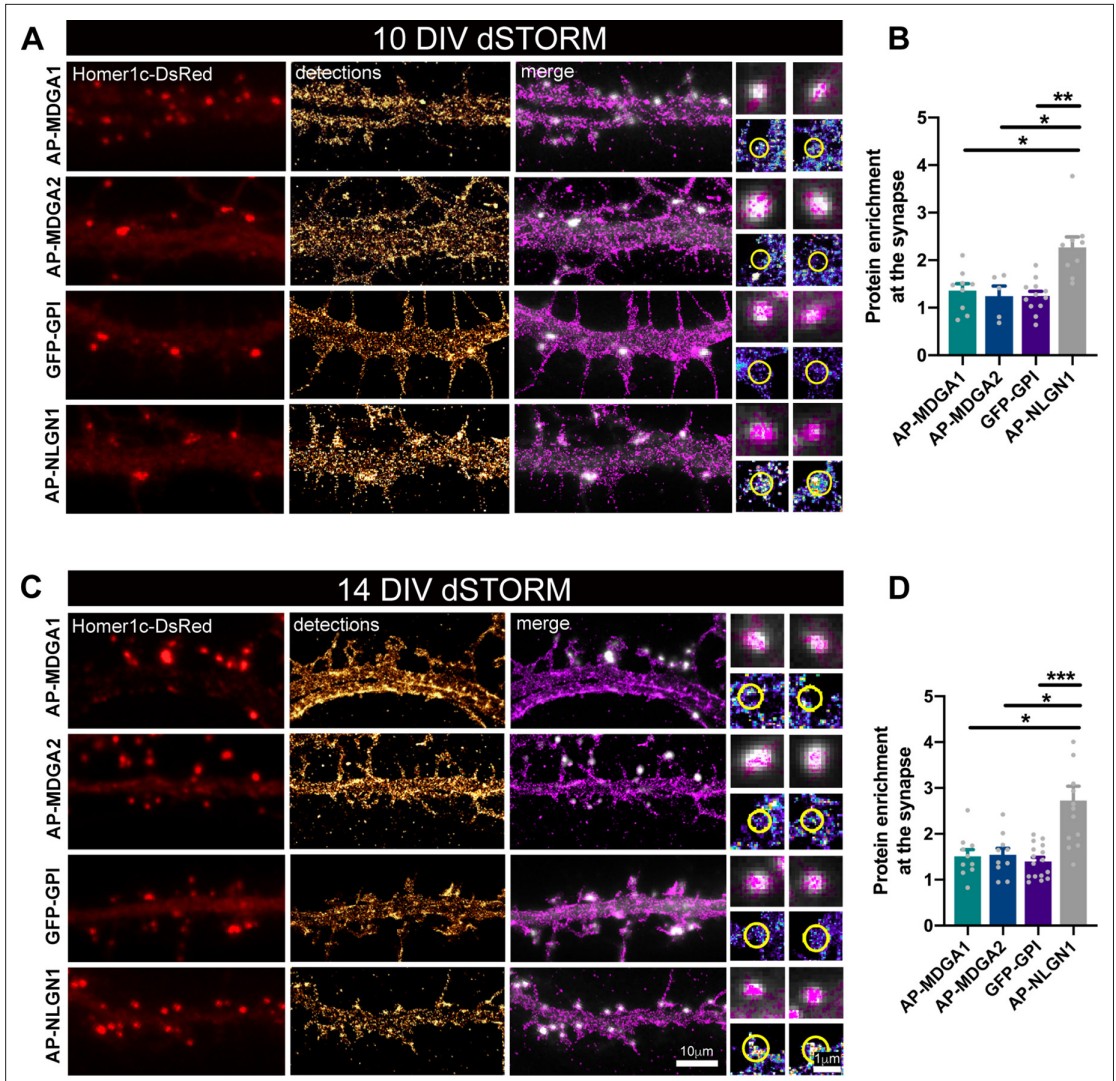

**Figure 2.** Nanoscale distribution of MDGA1 and MDGA2 in the neuronal membrane. Hippocampal neurons were electroporated at DIV 0 with a combination of shRNAs to MDGA1 or MDGA2, rescue AP-MDGA1 or AP-MDGA2 (respectively), biotin ligase BirA[ER], and Homer1c-DsRed. Alternatively, neurons were electroporated with shRNA to NLGN1, rescue AP-NLGN1, biotin ligase (BirA[ER]), and Homer1c-DsRed, or with GFP-GPI and Homer1c-DsRed. dSTORM experiments were performed at DIV 10 or 14, after labelling neurons with Alexa 647-conjugated mSA (for AP-tagged MDGAs and NLGN1) or Alexa 647-conjugated GFP nanobody (for GFP-GPI). (**A, C**) Representative images of dendritic segments showing Homer1c-DsRed positive synapses (in red), the super-resolved localization map of all AP-MDGA1, AP-MDGA2, GFP-GPI, or AP-NLGN1 single molecule detections (gold), and merged images (Homer1c-DsRed in white and detections in magenta). Insets on the right show zoomed images of different examples of Homer1c-DsRed-positive puncta overlapped with localizations (magenta) or pseudo-coloured localizations in a synaptic area marked by a yellow circle. (**B, D**) Bar plots representing the enrichment of AP-MDGA1, AP-MDGA2, GFP-GPI, and AP-NLGN1 localizations at synapses. Values were obtained from n > 5 neurons for each experimental condition and from at least three independent experiments. Data were compared by a Kruskal–Wallis test followed by Dunn's multiple comparison test (*p < 0.05; **p < 0.01; ***p < 0.001). For the statistics of the data presented in panels (**B,D**), see *Supplementary file 1* and *Figure 2—source data 1*.

The online version of this article includes the following source data and figure supplement(s) for figure 2:

**Source data 1.** Excel file containing all raw data and statistical tests used in *Figure 2*.

**Figure supplement 1.** Validation of shRNA and rescue MDGA constructs in COS-7 cells.

**Figure supplement 1—source data 1.** Source image of anti-MDGA1 immunoblot related to *Figure 2—figure supplement 1A*.

**Figure supplement 1—source data 2.** Source image of anti-HA immunoblot related to *Figure 2—figure supplement 1A*.

**Figure supplement 1—source data 3.** Source image of anti-Tubulin immunoblot related to *Figure 2—figure supplement 1A*.

**Figure supplement 1—source data 4.** Source image of anti-HA immunoblot related to *Figure 2—figure supplement 1B*.

*Figure 2 continued on next page*

*Figure 2 continued*

**Figure supplement 1—source data 5.** Source image of anti-actin immunoblot related to *Figure 2—figure supplement 1B*.

**Figure supplement 1—source data 6.** PDF file showing all the immunoblots in *Figure 2—figure supplement 1* where the relevant bands chosen for illustration are highlighted by red rectangles.

**Figure supplement 1—source data 7.** Excel file containing all raw data and statistical tests used in *Figure 2—figure supplement 1*.

**Figure supplement 2.** Validation of shRNA and rescue MDGA constructs in neuronal cultures.

**Figure supplement 2—source data 1.** Excel file containing all raw data and statistical tests used in *Figure 2—figure supplement 2*.

**Figure supplement 3.** The labeling strategy does not impair the NLGN1-MDGA interaction.

**Figure supplement 3—source data 1.** Source image of anti-NLGN1 immunoblot related to *Figure 2—figure supplement 3A*.

**Figure supplement 3—source data 2.** Source image of anti-MDGA1 immunoblot related to *Figure 2—figure supplement 3A*.

**Figure supplement 3—source data 3.** Source image of anti-NLGN1 immunoblot related to *Figure 2—figure supplement 3B*.

**Figure supplement 3—source data 4.** Source image of anti-HA immunoblot related to *Figure 2—figure supplement 3B*.

**Figure supplement 3—source data 5.** PDF file showing all the immunoblots in *Figure 2—figure supplement 3* where the relevant bands chosen for illustration are highlighted by red rectangles.

labelling prevented a reliable estimation of MDGA distribution, as previously documented for NLGN1 (*Chamma et al., 2016a*). Moreover, we were lacking a good antibody to MDGA2 for surface staining. Thus, to monitor the precise localization of MDGAs expressed at near endogenous levels, we replaced native MDGAs by recombinant tagged counterparts allowing for their detection at the ensemble and single molecule levels. To knock down native MDGAs, we used previously published shRNAs (*Loh et al., 2016*; *Pettem et al., 2013*). The efficiency and specificity of MDGA1 and MDGA2 silencing in our conditions was first assessed in COS-7 cells by co-expressing shRNAs to MDGA1 or MDGA2 with recombinant MDGA1 or MDGA2 followed by Western blot (*Figure 2—figure supplement 1*). In neurons, we measured a 75% decrease of surface MDGA1 immunofluorescence level in cells expressing shMDGA1 as compared with non-electroporated cells or cells expressing shCTRL (*Figure 2—figure supplement 2A,C*). In neurons electroporated with shMDGA2, we estimated a 40% reduction in mRNA level as compared with shCTRL by RT-qPCR (*Figure 2—figure supplement 2E*). We then rescued endogenous MDGAs with recombinant rat MDGA1 or MDGA2 bearing the short N-terminal biotin acceptor peptide (AP), which is biotinylated upon the co-expression of biotin ligase (BirA$^{ER}$) allowing for its detection with streptavidin (*Howarth et al., 2005*). Neurons co-expressing shMDGA1 and rescue AP-MDGA1 showed a 1.5-fold increase in MDGA1 surface immunostaining compared to non-electroporated cells, reflecting a mild over-expression (*Figure 2—figure supplement 2B, D*).

We then performed direct STochastic Optical Reconstruction Microscopy (dSTORM) experiments (*Dani et al., 2010*) after high density live labeling of AP-MDGA1/2 with monomeric streptavidin (mSA) (*Chamma et al., 2017*; *Demonte et al., 2013*) conjugated to Alexa 647 (100 nM concentration), reaching an optical resolution of about 30 nm. Since MDGAs are GPI-anchored membrane molecules, we electroporated neurons with GFP-GPI as a control, and labeled them with an anti-GFP nanobody also conjugated to Alexa 647, a strategy previously validated for GFP-NRXN1β (*Chamma et al., 2016a*). Using this approach, AP-MDGA1 and AP-MDGA2 displayed a fairly uniform distribution at DIV 10 and 14, filling the whole dendritic shaft without specific accumulation at synapses, similarly to the negative control GFP-GPI (*Figure 2A and C*). In post-synapses labeled by Homer1c-DsRed, MDGAs and GFP-GPI displayed a disperse localization (insets). For comparison, AP-NLGN1 expressed under similar replacement conditions (shRNA + rescue) and labeled identically with mSA-Alexa 647, showed a strong accumulation at synapses as previously shown (*Chamma et al., 2016a*). Synaptic enrichment at DIV 10 and 14 was around 1.3 and 1.5 for both MDGAs and GFP-GPI, and significantly higher for NLGN1 (2.3 and 2.7, respectively) (*Figure 2B and D*). These data show that MDGAs are not particularly enriched at excitatory synapses, and their differential localization with respect to NLGN1 suggest that the majority of NLGN1 molecules accumulated at post-synapses are not associated to MDGAs.

To rule out the possibility that the mSA probe was hindering the binding of MDGAs to NLGNs, and hence the penetration of MDGAs in synapses, we performed a series of control biochemical and immunocytochemical experiments. Streptavidin pull-down of proteins extracted from COS-7 cells expressing AP-MDGA1, BirA$^{ER}$, and HA-NLGN1, followed by anti-MDGA1 and anti-NLGN1

immunoblots, revealed that biotinylated AP-MDGA1 strongly recruits HA-NLGN1 (*Figure 2—figure supplement 3A*). This finding suggests that mSA, which is four times smaller than regular streptavidin (*Demonte et al., 2013*), should easily access AP-tagged MDGAs bound to endogenous NLGN1 in neurons. Given the high sequence and structure similarity between MDGA1 and MDGA2 (*Elegheert et al., 2017*), we expect AP-MDGA2 to also bind NLGN1 in this assay. To confirm that the interaction between MDGAs and NLGN1 also occurs when these molecules are bound to external probes in living cells, we performed cross-linking experiments using a mixture of a primary mouse anti-biotin and secondary anti-mouse antibodies in COS-7 cells expressing AP-MDGA1, HA-NLGN1 and BirA^ER, or in cells expressing HA-MDGA2, AP-NLGN1 and BirA^ER (*Figure 2—figure supplement 3C-F*). In both cases, the fluorescent antibody clusters that aggregated AP-tagged proteins contained the HA-tagged co-expressed protein, demonstrating no hindrance caused by antibodies (which are much larger than mSA) on the MDGA-NLGN1 interaction. Strengthened by these controls, our dSTORM data clearly indicate that MDGAs are not enriched at post-synapses, supporting the concept that MDGAs do not bind NRXN-occupied NLGNs at synapses.

## Individual recombinant MDGA1 and MDGA2 are highly diffusive in the neuronal membrane

To characterize the surface dynamics of MDGAs at the individual level, we sparsely labelled bioti-nylated AP-MDGAs at the cell membrane using 1 nM mSA conjugated to the robust fluorophore STAR 635 P, and performed single-molecule tracking by universal Point Accumulation In Nanoscale Topography (uPAINT) (*Figure 3*), as described earlier (*Chamma et al., 2016a*). Experiments were performed at DIV 8, 10, or 14, a time window of active excitatory synapse differentiation (*Chanda et al., 2017*). As a control, we electroporated neurons with GFP-GPI and labeled them with an anti-GFP nanobody conjugated to Atto 647 N, as described (*Lagardère et al., 2020*). At DIV 8, recombinant AP-MDGA1 and AP-MDGA2 diffused very fast in the dendritic membrane, showing a single peak of diffusion coefficient around 0.30 μm²/s, very similar to GFP-GPI (*Figure 3A and B*). Considering a small fraction (around 20%) of slowly mobile molecules, defined as molecules exploring an area smaller than the pointing accuracy of the optical system, i.e. D < 0.01 μm²/s (*Chamma et al., 2016a*), the median diffusion coefficient of the overall distribution was around 0.13–0.15 μm²/s across conditions (*Figure 3G and H*). Upon neuronal maturation (at DIV 10 and 14), the fraction of slowly mobile molecules increased for MDGA1 and MDGA2, with a concomitant decrease in median diffusion coefficient, while those parameters remained fairly constant for GFP-GPI (*Figure 3C–H*), suggesting a specific immobilization of MDGAs at these developmental stages.

## Individual recombinant MDGA1 and MDGA2 molecules are not trapped at synapses

Using the same set of data obtained from uPAINT experiments, we then examined the membrane domains explored by AP-tagged MDGA1 and MDGA2 in relation to the co-expressed post-synaptic marker Homer1c-DsRed, by constructing images integrating all single-molecule localizations over time. We found that, at the individual level, neither AP-MDGA1 nor AP-MDGA2 molecules were particularly retained at synapses (*Figure 4A and C*), confirming the ensemble picture given by dSTORM. As shown in the insets, both MDGA1 and MDGA2 displayed a panel of localizations including: (i) a complete absence from the post-synapse, (ii) the formation of small clusters reflecting confined trajectories localized at the periphery of Homer1c-DsRed puncta, and (iii) a more dispersed distribution filling the whole post-synapse (*Figure 4A and C*). For comparison, GFP-GPI exhibited essentially the third type of behavior, that is it explored the whole post-synapse with fast diffusion. Very rarely did MDGAs or GFP-GPI display confined trajectories at the core of the post-synaptic density like NLGN1 or LRRTM2 (*Chamma et al., 2016a*), suggesting an absence of synaptic retention. To directly compare the localization of MDGAs and LRRTM2, we expressed those molecules fused to an N-terminal V5 tag, and tracked them by uPAINT using a V5 Fab conjugated to STAR 635 P. V5-MDGA1 and V5-MDGA2 showed similar peri- and extra-synaptic distribution as their AP-tagged counterparts (*Figure 4—figure supplement 1*), while V5-LRRTM2 exhibited striking post-synaptic confinement as previously reported (*Chamma et al., 2016a*). To quantitatively characterize the presence of individual AP-MDGA1 and AP-MDGA2 molecules at the post-synapse, we measured a parameter called synaptic coverage, and defined as the fraction of the area of Homer1c-DsRed puncta occupied by AP-MDGAs or GFP-GPI

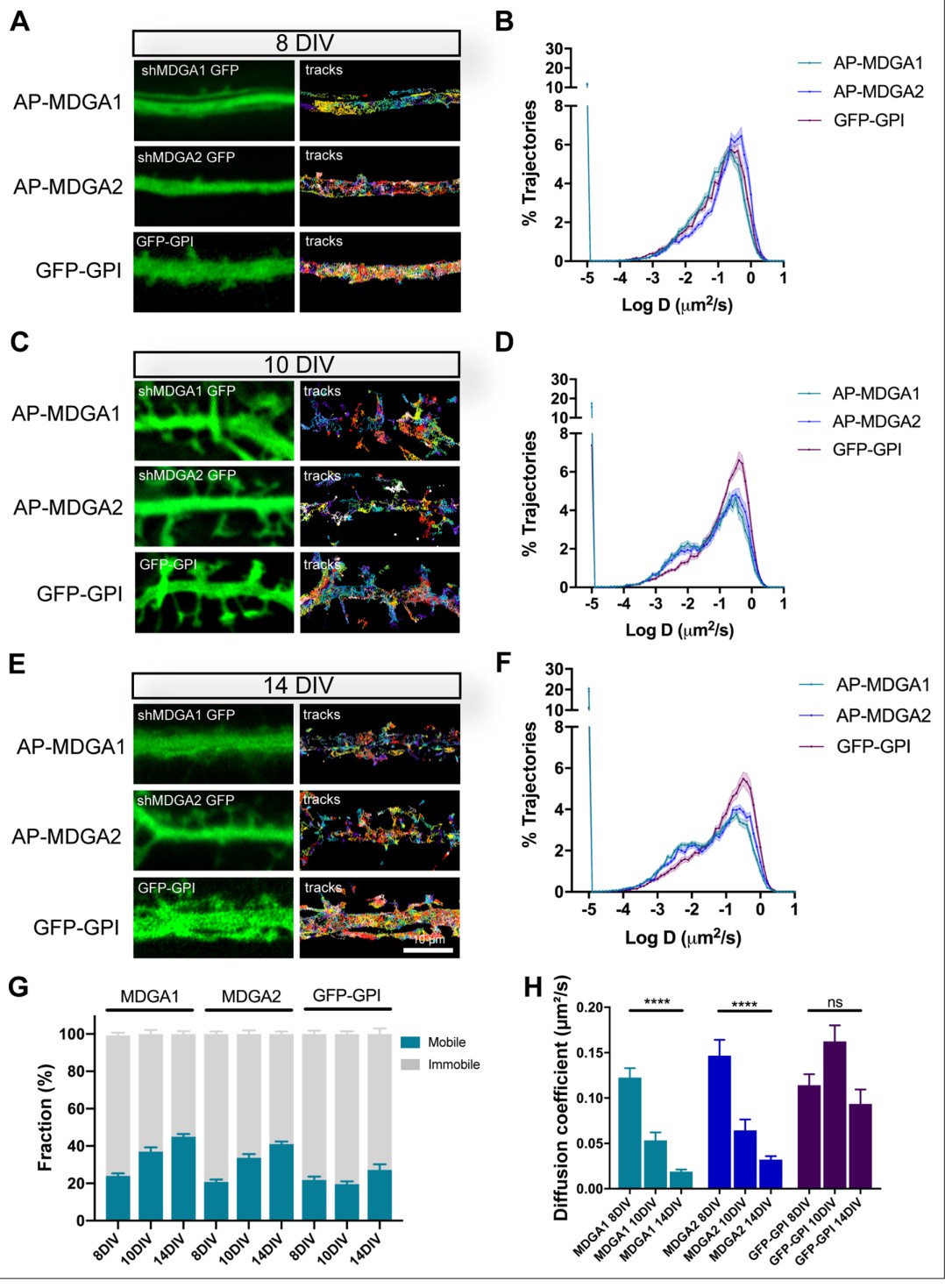

**Figure 3.** Lateral mobility of recombinant MDGAs across neuronal development. Dissociated rat hippocampal neurons were electroporated at DIV 0 with a combination of shRNAs to MDGA1 or MDGA2 (both carrying a GFP reporter), rescue AP-tagged MDGA1 or MDGA2 (respectively), and BirA[ER]. Control neurons were electroporated with GFP-GPI. uPAINT experiments were performed at DIV 8, 10, or 14, after labelling neurons expressing AP-MDGA1 or AP-MDGA2 with 1 nM STAR 635P-conjugated mSA, and labelling neurons expressing GFP-GPI with 1 nM Atto 647N-conjugated anti-GFP nanobody. (**A, C, E**) Representative images of dendritic segments showing the GFP signal (green) and the corresponding single molecule trajectories (random colors) acquired during an 80 s stream, for the indicated time in culture. (**B, D, F**) Corresponding semi-log plots of the distributions of diffusion coefficients for AP-MDGA1, AP-MDGA2, and GFP-GPI, at the three different developmental times. (**G**) Graph

*Figure 3 continued on next page*

of the mobile and immobile fractions of MDGA1, MDGA2, and GFP-GPI, as a function of time in culture. The threshold between mobile and immobile molecules was set at D = 0.01 µm²/s. (**H**) Graph of the median diffusion coefficient, averaged per cell, in the different conditions. Data represent mean ± SEM of n > 10 neurons for each experimental condition from at least three independent experiments, and were compared by a Kruskal–Wallis test followed by Dunn's multiple comparison test (**** p < 0.0001). For the statistics of the data presented in panels (**B,D,F,G,H**), see *Supplementary file 1* and *Figure 3—source data 1*.

The online version of this article includes the following source data for figure 3:

**Source data 1.** Excel file containing all raw data and statistical tests used in *Figure 3*.

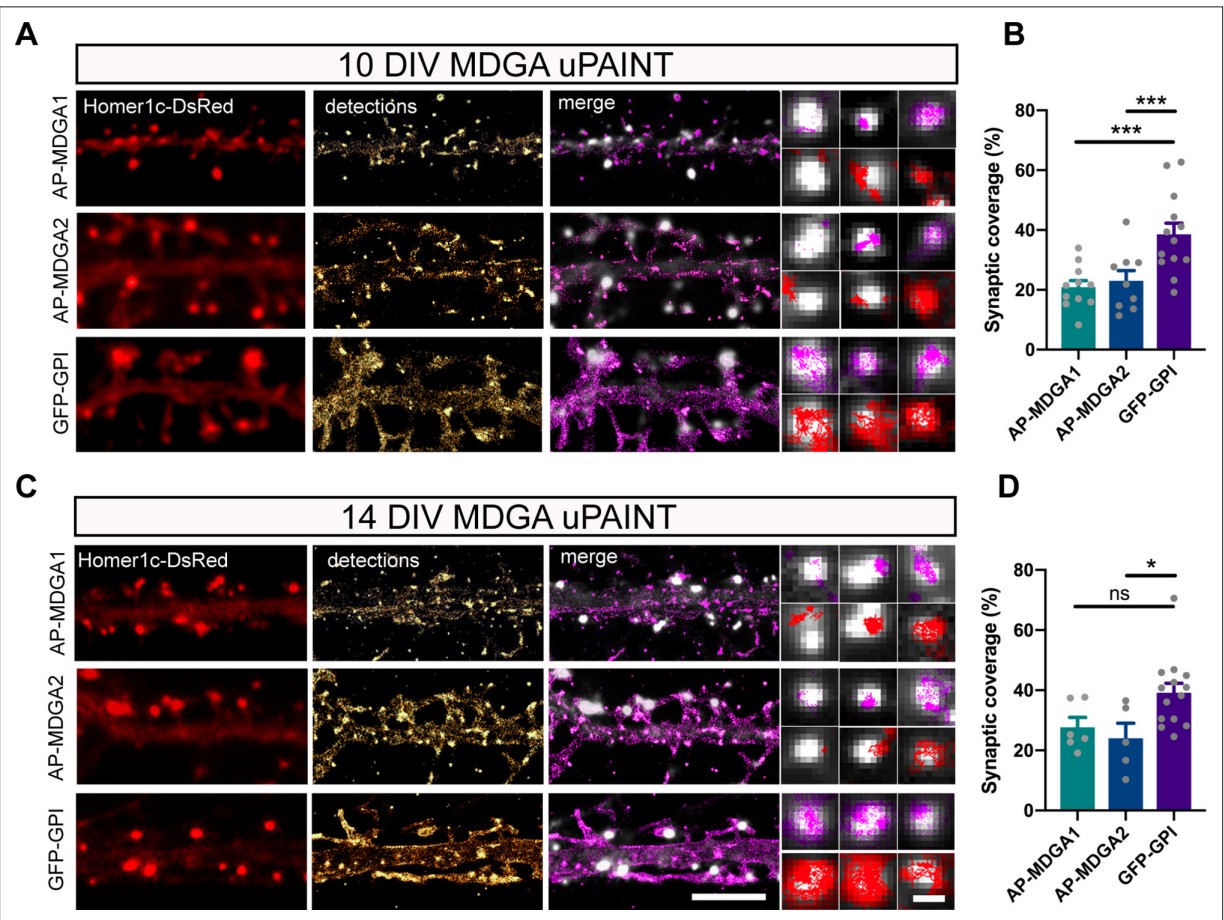

**Figure 4.** Single molecule localization of recombinant MDGAs with respect to post-synaptic densities. Hippocampal neurons were electroporated at DIV 0 with a combination of shRNAs to MDGA1 or MDGA2, rescue AP-MDGA1 or AP-MDGA2 (respectively), biotin ligase (BirA^ER), and Homer1c-DsRed. Control neurons were electroporated with GFP-GPI and Homer1c-DsRed. uPAINT experiments were performed at DIV 10 or 14, after labelling neurons with 1 nM STAR 635P-conjugated mSA or Atto 647N-conjugated anti-GFP nanobody, respectively. (**A, C**) Representative images of dendritic segments showing the Homer1c-DsRed signal (red), the super-resolved localization map of all AP-MDGA1, AP-MDGA2, or GFP-GPI single molecule detections (gold), and the corresponding trajectories (magenta) super-imposed to Homer1c-DsRed (white). Insets represent zooms on individual post-synapses in the different conditions (Homer1c-DsRed in white, detections in magenta and trajectories in red). (**B, D**) Bar plots representing synaptic coverage of AP-MDGA1, AP-MDGA2, or GFP-GPI at synapses, based on single molecule detections, for the two developmental stages (DIV 10 and 14), respectively. Data represent the mean ± SEM of 5–13 neurons for each experimental condition from at least three independent experiments, and were compared by a Kruskal–Wallis test followed by Dunn's multiple comparison test (*p < 0.05; ***p < 0.001). For the statistics of the data presented in panels (**B,D**), see *Supplementary file 1* and *Figure 4—source data 1*.

The online version of this article includes the following source data and figure supplement(s) for figure 4:

**Source data 1.** Excel file containing all raw data and statistical tests used in *Figure 2*.

**Figure supplement 1.** Lateral mobility and nanoscale localization of recombinant V5-MDGAs in hippocampal neurons.

based on single-molecule detections (*Figure 4B and D*). Synaptic coverage of MDGA1 and MDGA2 was only 20% at both DIV 10 and 14, while it reached 40% for GFP-GPI, indicating that MDGAs dynamically explore only a minor fraction of the synaptic cleft.

## MDGA2 knock-down increases synapse number and NLGN1 synaptic confinement

To characterize the influence of MDGAs on the behavior of their primary binding partner NLGN1, we knocked down MDGAs with shRNAs to MDGA1 (shMDGA1), to MDGA2 (shMDGA2), or to the non-related protein MORF4L1 as a control (shCTRL) (*Pettem et al., 2013*). Neurons were co-electro-porated at DIV 0 with these constructs together with Homer1c-DsRed. At DIV 10, a 2.5-fold increase in the density of Homer1c-DsRed puncta was observed in neurons expressing shMDGA2 relatively to shCTRL, whereas no significant effect of shMDGA1 was observed on the density of excitatory post-synaptic clusters (*Figure 5—figure supplement 1B,D*). At DIV 14, both shMDGA1 and shMDGA2 induced a modest 25% increase in the density of post-synaptic puncta (*Figure 5—figure supplement 1C,D*), suggesting an attenuation of the effect at later developmental stages. This differential effect of MDGA silencing on synape formation accross neuronal maturation resembles that of NLGN1 over-expression which exhibits major synaptogenic potential in younger neurons (DIV 6–7) and less so in older neurons (DIV 12–13) (*Dagar and Gottmann, 2019*). Considering the stronger effects of shMDGA2 and the selective role of MDGA2 on excitatory synapses reported earlier (*Connor et al., 2016*), we focused thereafter on the effects of MDGA2 on the dynamics, organization, and signaling mechanisms associated with NLGN1.

We first examined the diffusion properties of recombinant surface AP-NLGN1 sparsely labeled with STAR 635P-conjugated mSA with uPAINT. The presence of the AP tag and labeling with mSA should not interfere with the binding of NLGN1 to native MDGAs, as shown by streptavidin pull-down of proteins extracted from COS-7 cells expressing AP-NLGN1 and HA-MDGA2 (*Figure 2—figure supplement 3B*). By comparing neurons at DIV 10 and 14, there was a shift in NLGN1 mobility towards lower diffusion coefficients, which reflects a synaptic immobilization of NLGN1 upon neuronal matu-ration, as previously reported (*Chamma et al., 2016a*). In DIV 10 neurons, shMDGA2 had no effect on the NLGN1 diffusion coefficient, whose distribution looked very similar to the shCTRL condition (*Figure 5A–C*). In contrast, at DIV 14, shMDGA2 decreased the global diffusion coefficient of NLGN1 as compared to shCTRL, in particular by reducing the mobile pool of NLGN1 molecules (the peak centered at $D = 0.1\ \mu m^2/s$), and concomitantly raising the fraction of confined NLGN1 molecules (peak at $D = 0.01\ \mu m^2/s$) that are most likely retained at synapses (*Figure 5E–H*). This effect was reversed upon the co-expression of an HA-MDGA2 construct resistant to shMDGA2. These data indicate that MDGA2 impairs the synaptic immobilization of NLGN1, that is MDGA2 knock-down exacerbates the confinement of NLGN1 that normally occurs during neuronal maturation.

Second, we examined the nanoscale distribution of surface AP-NLGN1 densely labeled with Alexa 647-conjugated mSA using dSTORM. In DIV 10 neurons, there was no significant effect of shMDGA2 on NLGN1 enrichment at Homer1c-DsRed positive puncta compared to shCTRL (*Figure 6A and B*). In DIV 14 neurons, an increase from 3 to 4 in the synaptic enrichment of AP-NLGN1 was observed upon shMDGA2 expression as compared to shCTRL, albeit not significant (*Figure 6C and D*). In neurons co-expressing a MDGA2 construct resistant to the shRNA, the AP-NLGN1 synaptic enrichment was at the control level at both DIV 10 and 14. Taken together, uPAINT and dSTORM data suggest that MDGA2 impairs the immobilization of NLGN1 at newly formed synapses, but not its intrinsic post-synaptic accumulation.

## MDGA knock-down enhances NLGN tyrosine phosphorylation

In view of our previous findings that the effects of NLGN1 on synapse number and AMPAR-mediated synaptic transmission are regulated by the phosphorylation of a unique intracellular tyrosine (Y782) in NLGN1 (*Letellier et al., 2018*; *Letellier et al., 2020*), we examined whether MDGAs could affect NLGN1 phosphotyrosine level. Our rationale was that by shielding NLGN1, MDGAs could impair the NLGN1 phosphorylation signaling mechanism which is dependent on NRXN binding (*Giannone et al., 2013*). We electroporated neurons at DIV 0 with shMDGA2 or shCTRL and analyzed the phos-photyrosine level of immunoprecipitated NLGNs by performing immunoblot at DIV 10, when NLGN phosphorylation is maximal (*Letellier et al., 2018*). The NLGN phosphotyrosine level was almost

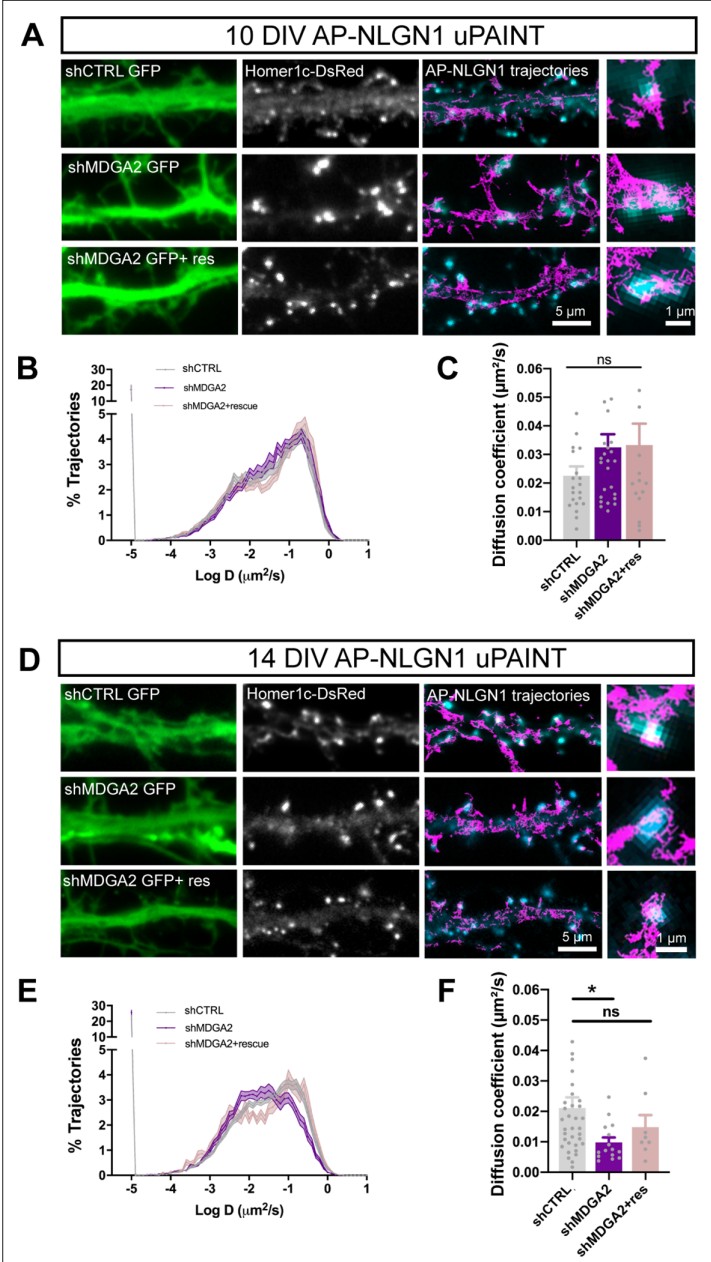

**Figure 5.** Effect of MDGA2 knock-down on NLGN1 membrane mobility. Neurons were electroporated at DIV 0 with AP-NLGN1, BirA[ER] and Homer1c-DsRed, plus shCTRL, shMDGA2, or shMDGA2 + rescue HA-MDGA2, and imaged at DIV 10 or 14 using uPAINT. (**A, D**) AP-NLGN1 was sparsely labelled using 1 nM STAR 635P-conjugated mSA for single molecule tracking in DIV 10 and 14 neurons, respectively. The GFP reporter of the different shRNAs appears in green, and the Homer1c-DsRed signal in white. On the right of each panel, merged images of Homer1c-DsRed (cyan) and AP-NLGN1 trajectories (magenta) acquired during an 80 s stream are shown. Insets represent zooms on individual post-synapses in the different conditions. (**B, E**) Semi-log plots of the distribution of AP-NLGN1 diffusion coefficients in DIV 10 and 14 neurons, respectively. (**C, F**) Median diffusion coefficient of AP-NLGN1. Data represent the mean ± SEM obtained from n = 21/26/16 neurons at DIV 10, and n = 35/15/8 neurons at DIV 14 out of three independent experiments, and were compared by a Kruskal–Wallis test followed by Dunn's multiple comparison test (*p < 0.05). For the statistics of the data presented in panels (**B,C,E,F**), see *Supplementary file 1* and *Figure 5—source data 1*.

The online version of this article includes the following source data and figure supplement(s) for figure 5:

**Source data 1.** Excel file containing all raw data and statistical tests used in *Figure 5*.

*Figure 5 continued on next page*

*Figure 5 continued*

**Figure supplement 1.** MDGAs knock-down increases excitatory synaptic density.

**Figure supplement 1—source data 1.** Excel file containing all raw data and statistical tests used in *Figure 5— figure supplement 1*.

two-fold higher in neurons expressing shMDGA2 compared to shCTRL, with no change in the total amount of NLGNs (*Figure 6E–G*). This result demonstrates that endogenous MDGAs negatively regulate NLGN tyrosine phosphorylation.

Given that NLGN1 tyrosine phosphorylation is likely influenced by NRXN binding (*Giannone et al., 2013*) and that MDGA2 inhibits NLGN1 binding to NRXNs (*Elegheert et al., 2017*), we characterized the dynamics of GFP-NRXN1β in axons making contacts with the dendrites of neurons in which MDGA2 was knocked-down, expecting a preferential reduction in mobility and/or increase in confinement of GFP-NRXN1β at contact sites. To this aim, we co-cultured neurons electroporated with GFP-NRXN1β with neurons electroporated with shMDGA2 or shCTRL (both containing an EBFP2 reporter) plus an intrabody to PSD-95 as a post-synaptic marker (Xph20-mRuby2) (*Rimbault et al., 2019*), and searched at DIV 10 for GFP-NRXN1β positive axons contacting EBFP2-positive dendrites. Then, we analyzed the GFP-NRXN1β enrichment at axon-dendrite contact sites, or we sparsely labelled GFP-NRXN1β with Atto 647N-conjugated anti-GFP nanobody and performed uPAINT (*Chamma et al., 2016a*; *Klatt et al., 2021*). The GFP-NRXN1β enrichment at pre-synapses was around 2 whether axons made contacts with dendrites expressing shMDGA2 or shCTRL (*Figure 6—figure supplement 1A,B*). In addition, the global diffusion coefficient of GFP-NRXN1β followed a broad distribution reflecting both fast diffusion in the axon and confinement at pre-synapses (*Chamma et al., 2016a*; *Klatt et al., 2021*; *Neupert et al., 2015*), but this distribution was not altered by the presence of contacting dendrites from neurons expressing either shMDGA2 or shCTRL (*Figure 6—figure supplement 1C,D*). Even though these results might suggest that MDGA KD does not directly affect the trans-synaptic NRXN1β-NLGN interaction, we have to moderate this explanation by considering that GFP-NRXN1β expressing axons make simultaneous contacts with dendrites from many neurons, such that the effect of shMDGA2 in sparsely electroporated cells is diluted.

## MDGA2 knock-down reduces AMPAR diffusion

Given the previously reported effects of NLGN1 expression level and phosphotyrosine signaling on AMPAR surface trafficking and synaptic recruitment (*Haas et al., 2018*; *Letellier et al., 2020*; *Letellier et al., 2018*; *Mondin et al., 2011*), and seeing here the impact of MDGA2 knock-down on NLGN1 dynamics and phosphotyrosine level, we then questioned the role of MDGA2 on AMPAR surface diffusion. We electroporated hippocampal neurons at DIV 0 with shMDGA2 or shCTRL and tracked native AMPARs at the single molecule level by uPAINT upon sparse labeling with an antibody to the GluA2 N-terminal domain conjugated to Atto 647 N (*Czöndör et al., 2013*; *Haas et al., 2018*; *Nair et al., 2013*). Expression of shMDGA2 significantly decreased the global AMPAR diffusion coefficient at DIV 10 compared to shCTRL (*Figure 7A–D*). Specifically, the mobile pool of AMPARs (D centered at 0.1 µm²/s) was reduced to the profit of slowly diffusing AMPARs (D < 0.01 µm²/s), most likely corresponding to synaptic receptors (*Nair et al., 2013*). This effect is consistent with the fact that shMDGA2 simultaneously increases the density of post-synapses (*Figure 5—figure supplement 1*), which act as trapping elements for surface-diffusing AMPARs (*Czöndör et al., 2012*; *Mondin et al., 2011*), resulting in an overall decrease in AMPAR mobility. At DIV 14, the distribution of AMPAR diffusion coefficients was shifted to the left as compared to DIV 10, reflecting AMPAR trapping at new synapses formed during this time interval (*Figure 7E–H*). Expression of shMDGA2 caused a further small decrease in diffusion coefficient, matching the observation that neurons expressing shMDGA2 show slightly higher numbers of excitatory synapses at DIV 14 as compared to neurons expressing shCTRL (*Figure 5—figure supplement 1C, D*).

## MDGA1 and MDGA2 knock-out selectively promote excitatory post-synaptic maturation

To achieve a stronger suppression of MDGAs than that obtained with shRNAs and further highlight the roles played by MDGA1 and MDGA2 in synapse development, we designed new DNA vectors

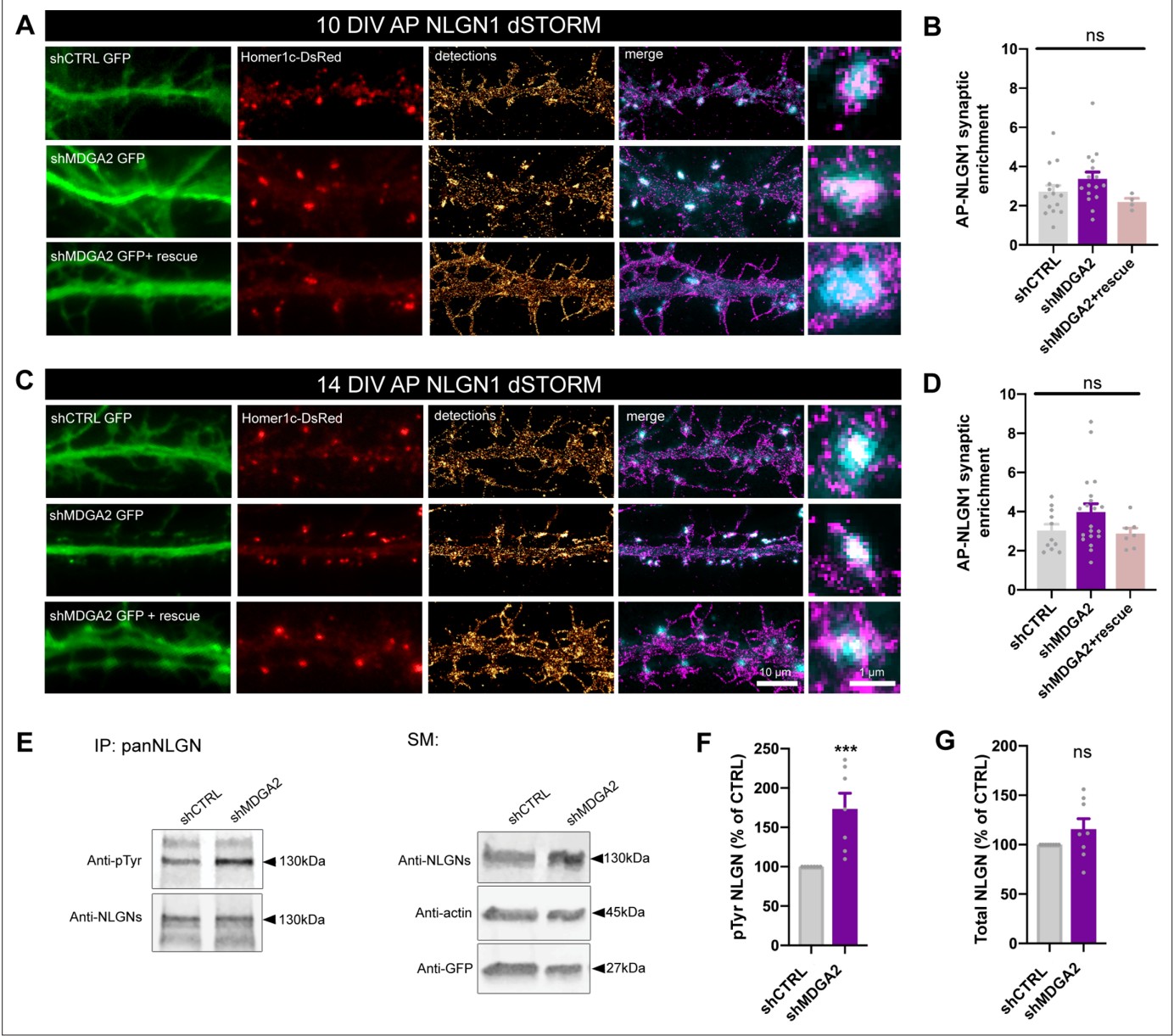

**Figure 6.** Effect of MDGA2 knock-down on NLGN1 nanoscale membrane localization and phosphorylation. (**A, C**) Neurons were electroporated at DIV 0 with AP-NLGN1, BirA^ER, and Homer1c-DsRed, plus shCTRL, shMDGA2, or shMDGA2 + rescue HA-MDGA2, and imaged at DIV 10 or 14 using dSTORM after high density labelling with Alexa 647-conjugated mSA. Representative images of dendritic segments show the GFP reporter of shRNAs (green), Homer1c-DsRed (red), and the integration of all AP-NLGN1 single molecule localizations (gold). Merged images show Homer1c-DsRed (cyan) and AP-NLGN1 localizations (magenta). Insets on the right show zoomed examples of Homer1c-DsRed positive puncta overlapped with AP-NLGN1 localizations. (**B, D**) Bar plots representing the enrichment of AP-NLGN1 at Homer1c-DsRed puncta. Data represent mean ± SEM from three independent experiments and were compared by a Kruskal–Wallis test followed by Dunn's multiple comparison test (n > 4 neurons at DIV 10 and n > 7 neurons at DIV 14 for each construct). (**E**) Hippocampal neurons were electroporated at DIV 0 with shCTRL or shMDGA2 and cultured for 10 days. Protein extracts were immunoprecipitated with a pan NLGN antibody. Phosphotyrosine (pTyr) and total NLGN levels were detected by Western blot in the immunoprecipitation (IP) samples, and pan NLGN, actin, and GFP were revealed in the starting material (SM). For original immunoblot images presented in panel (**E**), refer to *Figure 1—source data 1–5*. (**F, G**) Bar plots showing the average pTyr signal from the pan NLGN immunoprecipitate normalized to the total amount of immunoprecipitated NLGN, and the total amount of starting NLGN material in shCTRL and shMDGA2 electroporated cells, respectively. Data expressed as percentage of the shCTRL condition, represent the mean ± SEM from seven independent experiments and were compared by a Mann-Whitney test (***p < 0.001). For the statistics of the data presented in panels (**B,D,F,G**), see *Supplementary file 1* and *Figure 6—source data 6*.

The online version of this article includes the following source data and figure supplement(s) for figure 6:

*Figure 6 continued on next page*

*Figure 6 continued*

**Source data 1.** Source image of anti-pTyr immunoblot on NLGN pull-down related to *Figure 6E*.

**Source data 2.** Source image of anti-NLGN immunoblot on NLGN pull-down related to *Figure 6E*.

**Source data 3.** Source image of anti-NLGN immunoblot on starting material, related to *Figure 6E*.

**Source data 4.** Source image of anti-actin and anti-GFP immunoblots on starting material, related to *Figure 6E*.

**Source data 5.** PDF file showing all the immunoblots in *Figure 6* where the relevant bands chosen for illustration are highlighted by red rectangles.

**Source data 6.** Excel file containing all raw data and statistical tests used in *Figure 6*.

**Figure supplement 1.** Effect of MDGA knock down on NRXN1β localization and mobility.

**Figure supplement 1—source data 1.** Excel file containing all raw data and statistical tests used in *Figure 6—figure supplement 1*.

based on the CRISPR/Cas9 strategy to achieve single-cell knock-out of MDGA1 or MDGA2 in dissociated neurons (*Ran et al., 2013*). Specifically, hippocampal neurons were electroporated at DIV 0 with vectors containing the Cas9 gene, a guide RNA targeting either MDGA1, MDGA2, or a control sequence, plus a GFP or nuclear EBFP reporter. We first verified by genomic DNA cleavage that Cas9 was cutting the expected region of MDGA1 or MDGA2 genes only when the respective gRNA was present (*Figure 8—figure supplement 1A*). We also performed an extensive patch-seq analysis (*Cadwell et al., 2016*; *Fuzik et al., 2016*) of MDGA mRNAs and off target genes potentially affected by the electroporation of neurons with CRISPR/Cas9 constructs against MDGA1 or MDGA2 (Tables 1 and 2). The analysis performed in DIV 10 neurons expressing the GFP reporter clearly shows that MDGA1 and MDGA2 mRNA levels are significantly diminished by their respective CRISPR constructs as compared with CRISPR CTRL, while off target genes are not significantly affected (*Figure 8—figure supplement 2*). At the protein level, we observed an 80% reduction of endogenous MDGA1 immunostaining in neurons expressing CRISPR-Cas9 and gRNA to MDGA1, compared to neurons expressing control gRNA, revealing efficient MDGA1 knock-out (*Figure 8—figure supplement 1B, C*). No side

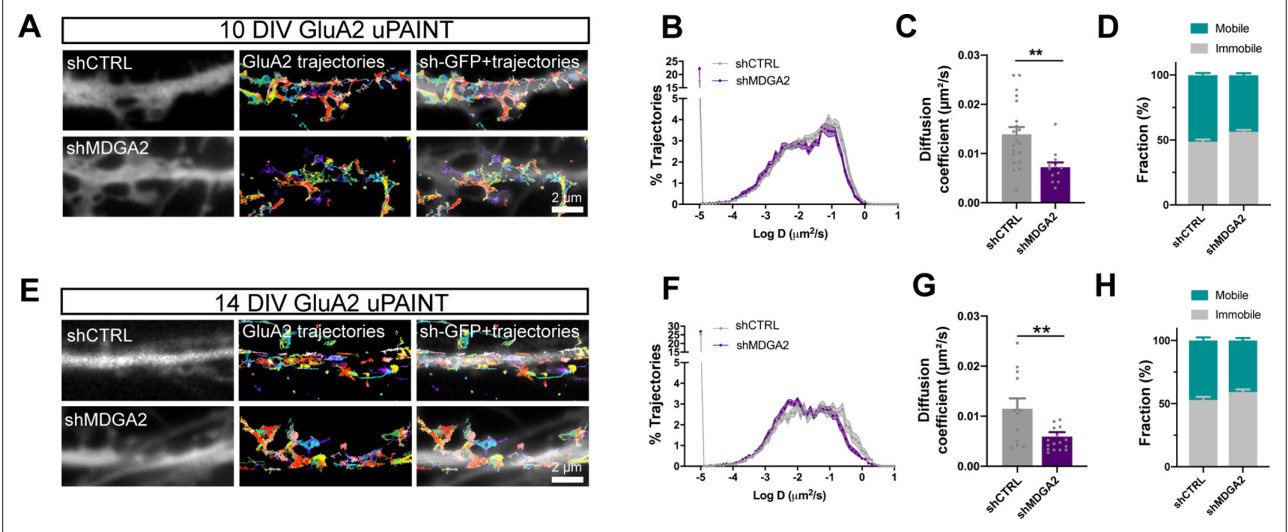

**Figure 7.** Effect of MDGA2 knock-down on AMPAR membrane mobility. Neurons were electroporated at DIV 0 with shCTRL-GFP or shMDGA2-GFP, and imaged at DIV 10 or DIV 14 with uPAINT. (**A, E**) Native AMPARs were sparsely labelled using an anti-GluA2 antibody conjugated to Atto 647 N. Representative trajectories of single GluA2-containing AMPARs are shown in multicolor, super-imposed to the distribution of the GFP reporter (white). (**B, F**) Semi-log plot of the distribution of GluA2 diffusion coefficients at DIV 10 and 14, respectively. The curves represent the averages of at least 12 neurons per condition from three independent experiments. (**C, G**) Median diffusion coefficient of GluA2-containing AMPARs at DIV 10 and 14, respectively. Data represent mean ± SEM of n > 12 neurons per condition from three independent experiments, and were compared by an unpaired t-test (**p < 0.01). (**D, H**) Bar plots of the immobile fraction of GluA2-containing AMPARs in the three conditions, defined as the proportion of single molecules with diffusion coefficient D < 0.01 μm²/s. For the statistics of the data presented in panels (**B,C,D,F,G,H**), see *Supplementary file 1* and *Figure 7—source data 1*.

The online version of this article includes the following source data for figure 7:

**Source data 1.** Excel file containing all raw data and statistical tests used in *Figure 7*.

effect of CRISPR against MDGA1 was observed on primary dendrite branching or outgrowth in these cultures (*Figure 8—figure supplement 1D–F*).

We then evaluated the effects of MDGA1/2 knock-out on the number and surface area of individual excitatory and inhibitory pre- and post-synaptic puncta immunolabeled for VGLUT1 and PSD-95, or VGAT and gephyrin, respectively. At DIV 10, an almost doubling in the number of PSD-95 puncta per unit dendrite length without a change in PSD-95 area, was observed in neurons expressing gRNAs to MDGA1 or MDGA2 relatively to control gRNA (*Figure 8A–C*). In the same conditions, only gRNA to MDGA2 caused a significant increase in the density of VGLUT1 puncta, and no change in area (*Figure 8A, D and E*). Those effects of gRNA to MDGA2 on both PSD-95 and VGLUT1 cluster density were abolished by the co-expression of a rescue MDGA2 vector, demonstrating the specificity of the mechanism. At DIV 14, gRNAs to MDGA1 or MDGA2 did not change PSD-95 cluster density or area relative to control gRNA, but slightly decreased both VGLUT1 puncta density and area (*Figure 8— figure supplement 3*). In contrast, no effects of CRISPR-Cas9 combined with either gRNA to MDGA1 or MDGA2 were found on the density or the area of gephyrin puncta at DIV 10 (*Figure 8—figure supplement 4A-C*). Together, these data show that both MDGA1 and MDGA2 mainly impair excitatory post-synaptic assembly in the early phase of synaptogenesis.

To examine the functional consequences of MDGA1 and MDGA2 knock-out on synaptic assembly, we measured both AMPAR-mediated miniature EPSCs (mEPSCs) and GABA$_A$-receptor-mediated miniature IPSCs (mIPSCs) by performing whole-cell patch-clamp recordings in DIV 10 neurons electroporated with either gRNAs to MDGA1 or MDGA2, or control gRNA (*Figure 8* and *Figure 8— figure supplement 4D-F*). Neurons expressing gRNAs to MDGA1 or MDGA2 showed a threefold increase in the frequency of AMPAR-mediated mEPSCs compared with neurons expressing control gRNA, while the combined expression of gRNA to MDGA2 and rescue MDGA2 abolished this effect (*Figure 8F and G*). No significant change in the amplitude of AMPAR-mediated mEPSCs was observed across conditions (*Figure 8H*). In parallel, endogenous surface AMPARs were live labeled with antibodies to the N-terminal of GluA1 subunits. There was no significant difference in GluA1 or GluA2 synaptic enrichment in neurons expressing Cas9 and gRNAs to MDGA1 or MDGA2, compared to neurons expressing Cas9 and control gRNA, despite an increase in the density of post-synaptic puncta as labeled by the PSD-95 intrabody Xph20 and of GluA1-positive clusters upon MDGA2 KO, but not MDGA1 KO (*Figure 8—figure supplement 5*). Together, these data suggest that knocking out MDGA2 selectively increases the density of AMPAR-containing excitatory synapses, but not the actual amount of AMPARs per synapse. In contrast, no significant change in either the frequency or amplitude of mIPSCs was observed upon single-cell KO of MDGA1 or MDGA2 (*Figure 8—figure supplement 4D-F*), suggesting that MDGAs do not affect inhibitory synapse formation during this developmental period.

## MDGA1 and MDGA2 knock-out selectively enhance AMPA-receptor-mediated synaptic transmission in organotypic slices

Finally, to examine the effects of MDGA KO in a neuronal system with better preserved synaptic connectivity than dissociated cultures, we turned to organotypic hippocampal slices prepared from P2 rats. CA1 neurons were single-cell electroporated at DIV 2 with CRISPR/Cas9 and gRNAs to MDGA1, MDGA2, or CTRL together with a volume marker (td-Tomato). Slices were processed 1 week later for electrophysiological recordings of both AMPAR- and NMDA receptor (NMDAR)- mediated EPSCs evoked in electroporated neurons by stimulation of Schaffer's collaterals (*Letellier et al., 2020*; *Letellier et al., 2018*; *Shipman et al., 2011*). In each experiment, a neighboring non-electroporated CA1 neuron serving as a paired control was recorded simultaneously, allowing for the normalization of EPSC amplitudes (*Figure 9A*). Strikingly, both gRNAs to MDGA1 and MDGA2 increased AMPAR-mediated EPSCs without affecting NMDAR-mediated EPSCs, as compared to non-electroporated controls (*Figure 9B-D* and *Figure 9—figure supplement 1A, B*). As a result, the ratio between AMPAR- and NMDAR-mediated EPSCs was significantly elevated in both MDGA1 and MDGA2 KO neurons (*Figure 9E*), supporting a synaptic unsilencing mechanism. No significant effect of gRNA CTRL on either AMPAR- or NMDAR-mediated EPSCs, or on the AMPA/NMDA ratio was observed when compared to non-electroporated neurons (*Figure 9B-E*), validating the normalization procedure and the absence of off-target effects of the control gRNA. No effect of CRISPR to MDGA1 or MDGA2 on the paired-pulse ratio was observed (*Figure 9—figure supplement 1C, D*), ruling out pre-synaptic

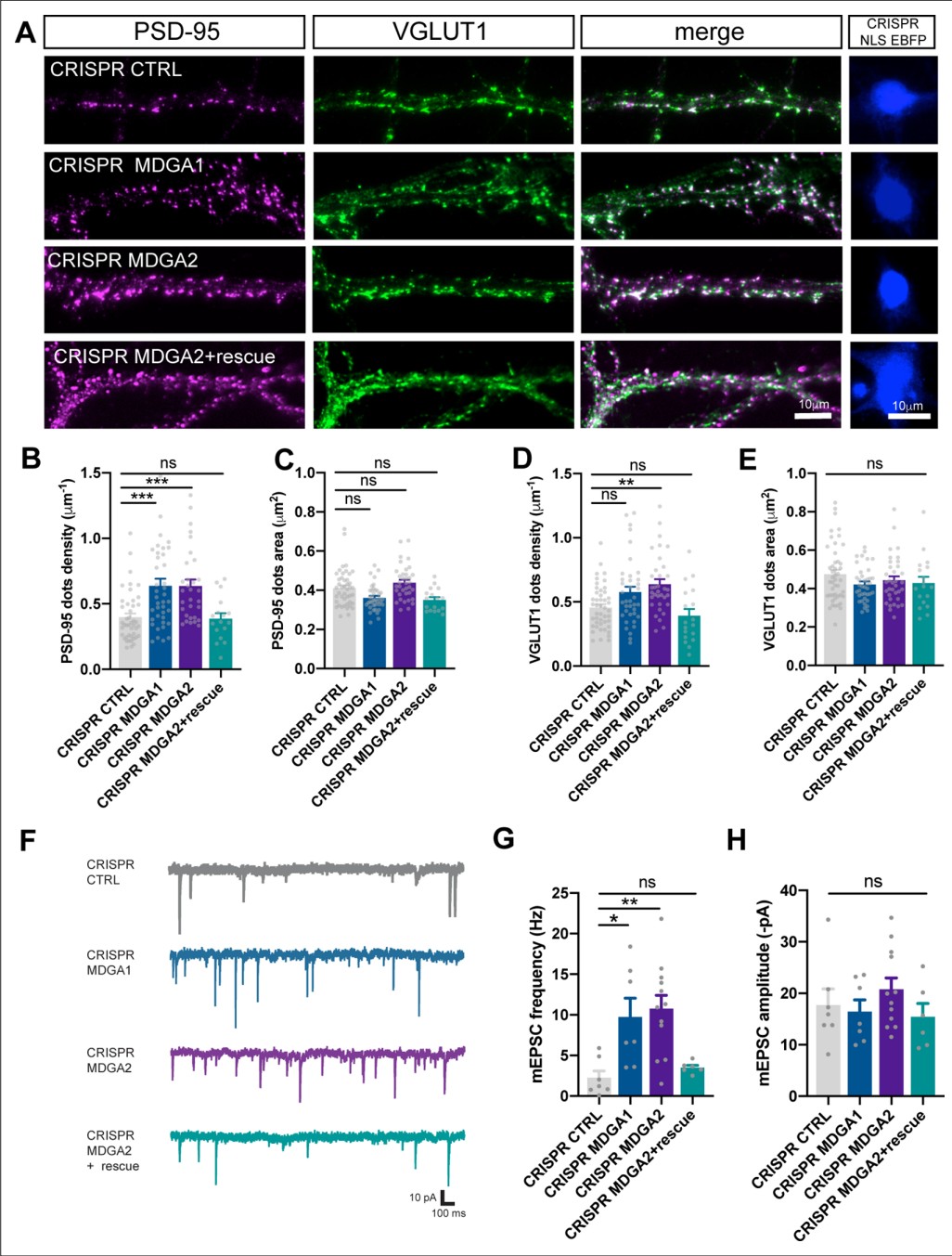

**Figure 8.** Effect of MDGA knock-out on synaptic density and transmission in dissociated neurons. Dissociated neurons were electroporated at DIV 0 with CRISPR/Cas9 CTRL, CRISPR/Cas9 MDGA1, CRISPR/Cas9 MDGA2, or CRISPR/Cas9 MDGA2 plus HA-MDGA2 rescue. Ten days after plating, cultures were fixed, permeabilized, and endogenous PSD-95 and VGLUT1 were immunostained. (**A**) Representative images of dendritic segments showing PSD-95 staining (magenta), VGLUT1 staining (green), the merged images, and the nuclear EBFP control of CRISPR/Cas9 construct expression (blue), in the different conditions. (**B–E**) Bar plots showing the density per unit dendrite length and surface area of individual PSD-95 and VGLUT1 puncta, respectively, in the various conditions. Data represent mean ± SEM of n > 17 cells for each experimental condition and from at least three independent experiments, and were compared by a Kruskal–Wallis test followed by Dunn's multiple comparison test (**p < 0.01; ***p < 0.001). (**F**) Representative traces of AMPAR-mediated mEPSC recordings from DIV 10 neurons expressing CRISPR/Cas9 CTRL, CRISPR/Cas9 MDGA1, CRISPR/Cas9 MDGA2, or CRISPR/Cas9 MDGA2 plus HA-MDGA2 rescue, clamped at –70 mV in the presence of tetrodotoxin and bicuculline. (**G, H**) Bar graphs of mEPSC frequency and amplitude respectively, for each condition. Plots represent mean ± SEM of n > 7 cells for each experimental

*Figure 8 continued on next page*

*Figure 8 continued*

condition from 5 independent experiments, and were compared by a Kruskal–Wallis test followed by Dunn's multiple comparison test (*p < 0.05; **p < 0.01). For the statistics of the data presented in panels (**B,C,D,E,G,H**), see *Supplementary file 1* and *Figure 8—source data 1*.

The online version of this article includes the following source data and figure supplement(s) for figure 8:

**Source data 1.** Excel file containing all raw data and statistical tests used in *Figure 8*.

**Figure supplement 1.** Validation of CRISPR/Cas9 knock-out of MDGA1/2.

**Figure supplement 1—source data 1.** Excel file containing all raw data and statistical tests used in *Figure 8—figure supplement 1*.

**Figure supplement 2.** Patch-seq analysis of MDGA1, MDGA2, and off-target genes in CRISPR/Cas9 expressing neurons.

**Figure supplement 2—source data 1.** Excel file containing all raw data and statistical tests used in *Figure 8—figure supplement 2*.

**Figure supplement 3.** MDGAs knock-out has no effect on synaptic density in DIV 14 neurons.

**Figure supplement 3—source data 1.** Excel file containing all raw data and statistical tests used in *Figure 8—figure supplement 3*.

**Figure supplement 4.** MDGAs knock-out has no effect on inhibitory synapses in DIV 10 neurons.

**Figure supplement 4—source data 1.** Excel file containing all raw data and statistical tests used in *Figure 8—figure supplement 4*.

**Figure supplement 5.** AMPAR membrane localization upon MDGA1/2 knockout.

**Figure supplement 5—source data 1.** Excel file containing all raw data and statistical tests used in *Figure 8—figure supplement 5*.

mechanisms. Together, those results strengthen the concept that both MDGA1 and MDGA2 down-regulate AMPAR recruitment during excitatory synapse development.

## Discussion

In this study, we characterized the membrane localization of MDGAs and their role on the dynamics and signaling of their direct binding partner, NLGN1, as well as associated effects on synaptic differentiation and the recruitment of AMPARs. We demonstrate that both MDGA1 and MDGA2 are essentially non-synaptically enriched molecules that exhibit fast diffusion in the dendritic membrane. Moreover, the knock-down of MDGAs selectively increases excitatory synapse density and as a consequence reduces the surface mobility of both NLGN1 and AMPARs, increases AMPAR-mediated mEPSC frequency and evoked EPSC amplitude, as well as NLGN1 phosphosignaling. Thus, by shielding a fraction of NLGN1 from binding to pre-synaptic NRXNs, MDGAs negatively regulates NLGN function in excitatory synaptic differentiation (*Figure 10*).

We first examined the localization of MDGAs in hippocampal tissue and dissociated cultures. The generation of specific antibodies allowed us to examine the distribution of native MDGA1, which showed strong expression levels in the neuropil of the CA region of the hippocampus, confirming previously shown results of in situ hybridization of MDGA1 mRNAs and staining of β-galactosidase activity expressed from the *Mdga1* locus (*Connor et al., 2017*; *Connor et al., 2016*; *Lee et al., 2013*). MDGA1 immunostaining in dissociated cultures at DIV 14 showed that a large fraction of excitatory synapses did not contain MDGA1. Recombinant MDGA1 and MDGA2 expressed by rescuing endogenous MDGA levels also showed no preferential retention at Homer1c-positive puncta, in contrast with the positive controls NLGN1 and LRRTM2 labeled similarly and showing strong post-synaptic accumulation, as previously described (*Chamma et al., 2016a*). Previous studies also reported a small colocalization extent of either YFP-MDGA1 or HRP-MDGA2 with excitatory synaptic markers (*Loh et al., 2016*; *Pettem et al., 2013*). Our results show that individual MDGAs often localized as transient sub-micron clusters at the periphery of Homer1c puncta, representing the confinement area of a small number of labeled molecules. The nature of these domains is unclear, but might represent a transition zone where NLGN1 could switch between MDGA-bound and NRXN-bound states, with the potential existence of mixed NLGN1 dimers that would exhibit specific mobility properties. In any case, the sum of this MDGA-rich peri-synaptic compartment plus the diffusive pool of MDGAs in synapses

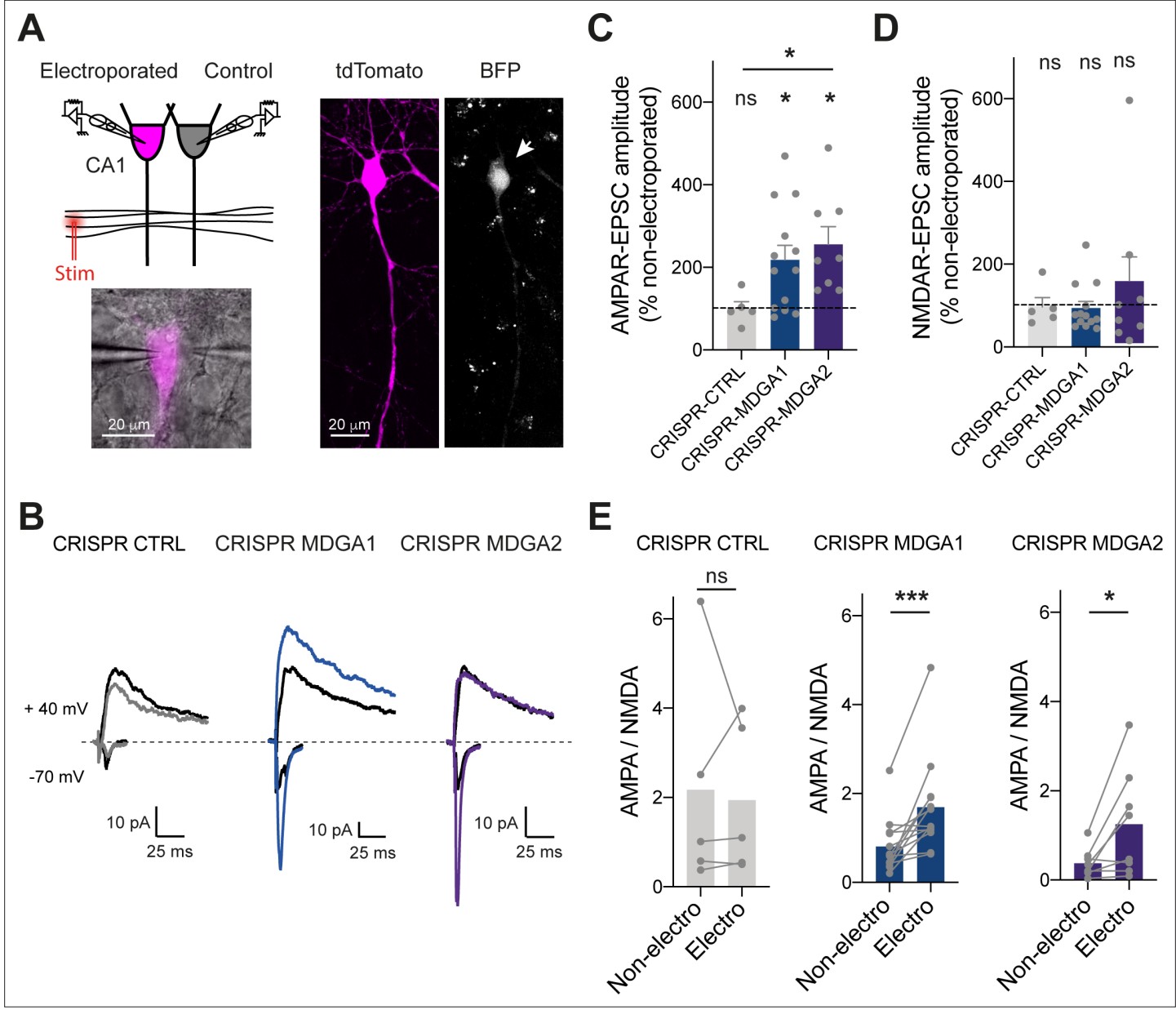

**Figure 9.** Effect of MDGA knock-out on AMPAR- and NMDAR-dependent synaptic transmission in CA1 neurons. CA1 neurons in rat organotypic hippocampal slices were single-cell electroporated at DIV 2 with CRISPR/Cas9 CTRL, CRISPR/Cas9 MDGA1, or CRISPR/Cas9 MDGA2, plus the tdTomato volume marker. One week later, electroporated neurons and non-electroporated control neighbors were processed for dual patch-clamp recordings upon stimulation of Schaffer's collaterals. (**A**) Dual whole-cell recording configuration with corresponding image from an experiment (left) and confocal images showing tdTomato (magenta) and BFP signals in an electroporated neuron (right). (**B**) Representative traces of evoked AMPAR- and NMDAR-mediated EPSCs recorded at −70 mV and + 40 mV, respectively. Color sample traces correspond to electroporated neurons in the three conditions, and black traces correspond to control, non-electroporated neurons. (**C, D**) Average AMPAR- and NMDAR-mediated EPSC amplitudes, respectively, normalized to the control condition (the dashed line indicates 100%). Data were compared to the control condition by the Wilcoxon matched-pairs signed rank test, and between themselves using one-way ANOVA followed by Tukey's multiple comparison (*p < 0.05; ns: not significant). (**E**) Average ratio between paired AMPAR- and NMDAR-mediated EPSCs in the three conditions. Data were compared using the Wilcoxon matched-pairs signed rank test (***p < 0.001; *p < 0.05; ns: not significant). For the statistics of the data presented in panels (**C,D,E**), see ***Supplementary file 1*** and ***Figure 9—source data 1***.

The online version of this article includes the following source data and figure supplement(s) for figure 9:

**Source data 1.** Excel file containing all raw data and statistical tests used in ***Figure 9***.

**Figure supplement 1.** Scatter plots of AMPAR- and NMDAR-dependent EPSCs and paired pulse ratio (PPR) recorded in CA1 neurons expressing CRISPR/Cas9 to MDGA1/2.

**Figure supplement 1—source data 1.** Excel file containing all raw data and statistical tests used in ***Figure 9—figure supplement 1***.

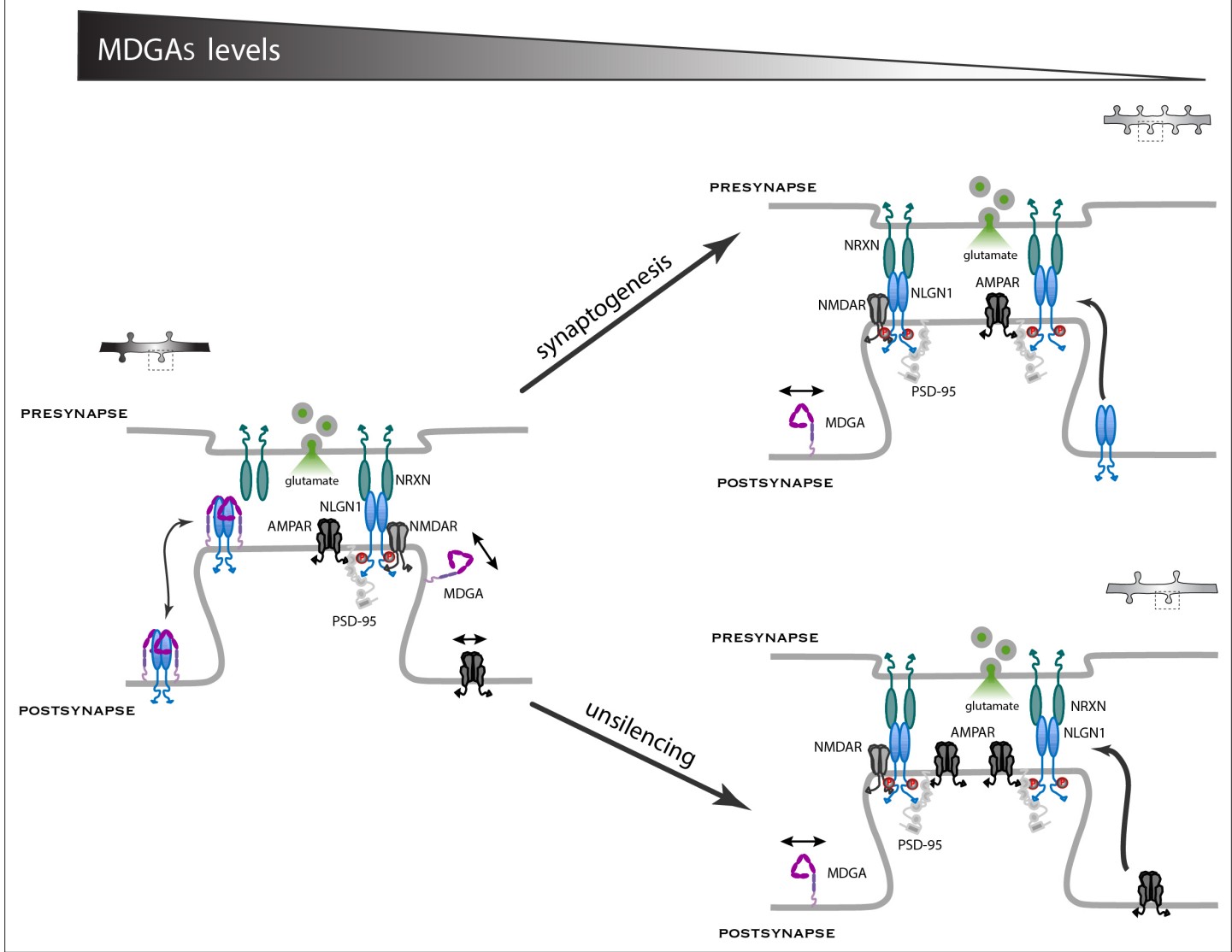

**Figure 10.** Working model for the role of MDGAs in excitatory synapse differentiation. Since NRXNs and MDGAs compete for the binding to NLGN1, the MDGA concentration acts as a key regulator of the signaling events downstream of the NRXN-NLGN1 interaction. When the MDGA level is low in response to KD or KO (right panels), the preferential interaction of NLGN1 with NRXN favors NLGN1 tyrosine phosphorylation and the associated development of excitatory synapses containing AMPARs. When the MDGA level is high (left panel), the NRXN-NLGN1 interaction is weakened and the formation of excitatory synapses is delayed. In dissociated neurons, MDGAs primarily regulate the overall density of NLGN1 and AMPAR modules, but not the actual amount of these molecules at individual synapses (arrow 'synaptogenesis'). In organotypic slices, MDGAs tend to keep synapses in a silent state with low amounts of AMPARs, and single-cell KO of either MDGA1 or MDGA2 in CA1 neurons promotes the selective recruitment of AMPARs (arrow 'unsilencing').

might represent enough material to be detected in synaptosomes (this study) and in the synaptic cleft proteome (*Loh et al., 2016*). When sampled at saturating labeling density using dSTORM, MDGAs showed a rather diffuse localization in the shaft and at synapses both at DIV 10 and 14, with modest synaptic enrichment as the negative control GFP-GPI. As compared to uPAINT, which gives information on the membrane dynamics of a subset of sparsely labeled molecules, dSTORM performed after saturating live labeling provides a snapshot of the whole population of MDGAs that integrates over time many transient confinement areas into a single 2D projection image. Thus the localization of molecules observed in dSTORM seems naturally more homogeneous, as previously reported for NLGN1 (*Chamma et al., 2016a*).

In addition, AP-MDGA1 and AP-MDGA2 exhibited fast membrane diffusion throughout dendrites, even within post-synaptic sites, indicating no particular molecular retention at synapses. These

observations are compatible with the fact that MDGAs lack an intracellular C-terminal domain such as those present in NLGN1 or AMPAR auxiliary subunits, which bind PDZ domain-containing scaffolding proteins that stabilize them at synapses (*Bats et al., 2007*; *Irie et al., 1997*; *Mondin et al., 2011*). A mild confinement of MDGAs outside synapses was observed upon neuronal maturation, which might be due to the fact that a fraction of MDGAs bind to extra-synaptic NLGN clusters (*Gerrow et al., 2006*) or to another unknown protein, e.g. through their Ig4-6 domains (*Lee et al., 2013*). These domains might also be due to interactions of the GPI anchor of MDGAs to the lipid microenvironment present in some membrane microdomains such as lipid rafts (*Díaz-López et al., 2005*; *Renner et al., 2009*). By comparing side by side the distributions of diffusion coefficients for MDGAs and NLGN1, we estimate that around 12% of the highly mobile extra-synaptic MDGAs are not bound to NLGN1, otherwise MDGAs would naturally adopt the slower diffusion of NLGN1. Furthermore, the fact that MDGAs do not accumulate at synapses like NLGN1 over neuronal development (from DIV 10–14) suggests that synaptic NLGN1 is not bound to MDGAs, but instead to NRXNs which display high local concentration at pre-synapses (*Chamma et al., 2016a*; *Chamma et al., 2016b*). Thus, by preventing dendritic NLGN1 from aberrantly binding to the fraction of freely-diffusing NRXNs at the surface of contacting axons (*Chamma et al., 2016a*; *Neupert et al., 2015*), native MDGAs seem to protect neurons from forming synapses too quickly.

Using either previously reported shRNAs or newly generated CRIPSR/Cas9 constructs, we showed that MDGA2 delays excitatory synaptic development and AMPAR-dependent transmission, consistent with the in vivo KO of *Mdga2* (*Connor et al., 2016*). These effects were accompanied by a global decrease of NLGN1 diffusion, supporting the idea that by losing its MDGA partner, NLGN1 is more available to bind axonal NRXNs and thereby accelerates synapse formation. Indeed, the effects of MDGA2 KD and KO were more prominent at DIV 10 during the active phase of synaptogenesis, and barely detectable at DIV 14, in agreement with the lack of effect of shMDGA2 previously seen on excitatory synapse density in DIV 15 neurons (*Loh et al., 2016*). Upon MDGA2 knock-down, there was no significant increase in the synaptic enrichment of either NRXN1β or NLGN1, suggesting that synapses can be newly assembled using a fixed number of those trans-synaptic adhesion molecules. We found more contrasted results with MDGA1, that is the CRISPR/Cas9 strategy significantly increased excitatory post-synaptic density and AMPAR-mediated mEPSC frequency at DIV 10 in dissociated neurons, as well as evoked AMPAR-mediated EPSCs in organotypic slices, while shRNA to MDGA1 had little effect at both DIV 10 and DIV 14, in agreement with previous reports (*Pettem et al., 2013*). The discrepancy might be due to the fact that CRISPR/Cas9 represses MDGA1 expression more strongly than shMDGA1. Indeed, in previous studies, effects of MDGAs on excitatory and inhibitory synapse development were seen only when both MDGA1 and MDGA2 were knocked down (*Lee et al., 2013*; *Loh et al., 2016*), suggesting that MDGA1 and MDGA2 might compensate for each other's loss such that the important parameter in these experiments is the overall MDGA level. The selective increase in inhibitory - but not excitatory - synapses in hippocampal CA1 neurons of full *Mdga1* KO mice (*Connor et al., 2017*) might come from the fact that MDGA1 is silenced throughout the whole animal lifetime, which might lead to compensatory effects. Furthermore, MDGA1 knock-out affects all cell types, including both pyramidal neurons and astrocytes that also express MDGA1 and NLGNs (*Stogsdill et al., 2017*), which might overall cause mixed effects on hippocampal circuitry. A difference in the results obtained by global gene knock-out in mice and sparse single-neuron silencing was also seen for NLGN1 (*Chih et al., 2005*; *Varoqueaux et al., 2006*), suggesting that adhesion-dependent competition between neurons regulates synaptogenesis (*Kwon et al., 2012*), although sparsely knocking out NLGN1 did not reduce synapse number in another study (*Chanda et al., 2017*). Strikingly, the phenotypes we saw with single cell MDGA2 KO (e.g. increases in post-synapse numbers and AMPAR-mediated mEPSC frequency in dissociated neurons) resembles the ones we previously observed by over-expressing NLGN1, thereby reinforcing the view that neurons compete for synapse formation through NRXN-NLGN adhesion. MDGA1 KO had more subtle effects, that is it significantly increased AMPAR-mediated mEPSC frequency but not the density of VGlut1 or GluA1 puncta, which might be due to the fact that electrophysiology is a more sensitive technique than immunostaining to detect AMPAR-dependent synaptic changes (*Letellier et al., 2018*). In any case, crystal structures and affinity measurements support the concept of a stable MDGA1-NLGN1 complex (*Elegheert et al., 2017*; *Gangwar et al., 2017*; *Kim et al., 2017*), which should be compatible with the fact that endogenous MDGA1 can interact with NLGN1 as strongly as MDGA2 to impair excitatory synapse formation.

We also demonstrated a selective increase in the phosphotyrosine level of NLGNs upon MDGA2 knock-down. This observation relates to our previous findings that NLGN1 can be phosphorylated at a unique intracellular tyrosine residue (Y782), and that the NLGN1 phosphotyrosine level regulates the assembly of excitatory post-synaptic scaffolds in a NRXN-dependent fashion (*Giannone et al., 2013*). Moreover, using either the expression of NLGN1 point mutants or the optogenetic stimulation of endogenous NLGN1 phosphorylation, we showed that a high NLGN1 phosphotyrosine level is associated with the selective increase in excitatory synapse number and AMPAR-mediated synaptic transmission (*Letellier et al., 2020*; *Letellier et al., 2018*). Strikingly, single-cell MDGA KO in CA1 neurons selectively enhanced AMPAR- (but not NMDAR-) mediated synaptic transmission, further implicating NLGN in the recruitment of AMPARs at nascent synapses. We thus propose the model that by occupying NLGN1, MDGAs inhibit the NRXN-induced phosphotyrosine signaling pathway associated with NLGN1 and thereby delay the assembly of functional excitatory synapses containing AMPARs. In this context, we were surprised not to observe any impact of MDGA KD on the global diffusion or pre-synaptic accumulation of GFP-NRXN1β in axons contacting neurons expressing shMDGA2, but these effects might have been obscured by the limited number of contacts between these cells or by NRXN1β over-expression. Future single-molecule tracking experiments on NRXN1-α or -β expressed at endogenous levels may allow one to clarify this issue (*Klatt et al., 2021*; *Neupert et al., 2015*). The enhancement of NLGN tyrosine phosphorylation caused by MDGA2 KD might be linked to the increased number of synapses, leading to enhanced local concentration of receptor tyrosine kinases such as Trk family members that are able to phosphorylate native NLGNs (*Letellier et al., 2018*). Since NLGN3 can also be tyrosine phosphorylated in neurons (*Letellier et al., 2018*), a contribution of NLGN3 to the increase in phosphotyrosine level seen upon MDGA2 KD is possible given that we used a pan NLGN antibody to reach efficient immunoprecipitation. However, the fact that MDGA binds 10-fold more weakly to NLGN3 than to NLGN1 in vitro (*Elegheert et al., 2017*), and that the effects of NLGN tyrosine phosphorylation on post-synaptic differentiation are not seen in cultures from *NLGN1* KO mice (*Letellier et al., 2020*) suggest that NLGN3 might only play a minor role in this process.

The decrease in global AMPAR diffusion observed upon MDGA2 knock-down can be linked to the parallel increase in the number of synaptic puncta. Indeed, a similar decrease in AMPAR diffusion was seen across neuronal development in culture or upon the over-expression of NLGN1, which both enhance the number of post-synapses that act as trapping elements for surface diffusing AMPARs (*Czöndör et al., 2012*; *Mondin et al., 2011*). Upon MDGA2 knock-down, a transient immobilization of surface-diffusing AMPARs is expected to occur at newly formed synapses enriched in NLGN1, resembling what was previously observed at micro-patterned dots coated with NRXN1β-Fc (*Czöndör et al., 2013*). Interestingly, the actual content of AMPARs per post-synapse did not seem to be modified by MDGA KO in dissociated neurons, since both the synaptic AMPAR enrichment and the amplitude of AMPAR-mediated mEPSCs remained similar to control conditions. However, the density of synaptic puncta as well as the frequency of AMPAR-mediated mEPSCs were significantly enhanced by MDGA KO, suggesting that the new synapses that had appeared contained functional AMPARs. This situation is similar to NLGN1 over-expression that increases the number of synapses and the frequency, but not the amplitude, of AMPAR-mediated mEPSCs (*Letellier et al., 2018*; *Levinson et al., 2005*). In both cases (MDGA knock-down or NLGN overexpression), AMPARs seem to be inserted in novel synapses as individual units or modules, most likely influenced by the presence of NLGN1 (*Haas et al., 2018*; *Hruska et al., 2018*; *MacGillavry et al., 2013*; *Nair et al., 2013*). In organotypic slices, MDGA KO selectively enhanced the AMPA/NMDA ratio in CA1 neurons but did not affect NMDAR-mediated EPSCs, indicating no net change in synapse number but instead an unsilencing mechanism (*Kerchner and Nicoll, 2008*). This process might also occur in dissociated neurons where the increase in the frequency of AMPAR-mediated mEPSCs upon MDGA KD (3-fold) was much larger than the enhancement in synapse number (50%). In any case, the robust recruitment of AMPARs through synaptogenic and/or unsilencing mechanisms ressembles the effects previously observed upon direct manipulation of NLGN1 expression or signaling (*Letellier et al., 2020*; *Letellier et al., 2018*; *Mondin et al., 2011*), suggesting that MDGAs act through a regulation of NLGN1 function.

Given the strong effects caused by MDGA loss-of-function on synaptic differentiation, the next challenge would be to determine which biological processes regulate endogenous MDGA levels in specific neuron types across development. One interesting factor might be constitutive neuronal activity that can influence synaptic protein expression levels, for example by modulating microRNAs

(*Dubes et al., 2019*; *Letellier et al., 2014*). Indeed, MDGA1 transcripts were recently found to be regulated in response to chronic synaptic activity blockade (*Silva et al., 2019*). Moreover, the action of MDGAs might be finely tuned by other proteins associated to the NRXN-NLGN trans-synaptic complex, including hevin and SPARC that are secreted by astrocytes (*Fan et al., 2021*). Interestingly, the presence of NLGNs in astrocytes offers an additional level of regulation of synapse development through such a network of proteins (*Stogsdill et al., 2017*). Finally, genetic mutations identified in patients with autism and leading to alterations in MDGA levels (*Bucan et al., 2009*), are expected to cause profound changes in synapse differentiation such as the ones shown here.

# Materials and methods

## Key resources table

| Reagent type (species) or resource | Designation | Source or reference | Identifiers | Additional information |
|---|---|---|---|---|
| Sequence-based reagent | gRNA control | This study | 5'-ATATTTCGGCAGTTGCAGCA-3' | CRISPR/Cas9 Construction |
| Sequence-based reagent | gRNA MDGA2 | This study | 5'-ATTTAGTGTACGGTCTCGTG-3' | CRISPR/Cas9 Construction |
| Sequence-based reagent | gRNA MDGA1 | This study | 5'-CTTCAACGTACGAGCCCGGG-3' | CRISPR/Cas9 Construction |
| Sequence-based reagent | AP tag sequence | This study | 5´-GGCCTGAACGAtATCTTCGAGGCCCAG AAGATCGAGTGGCACGAG-3´ | AP tag plasmids |
| Sequence-based reagent | MDGA1 CRISPR validation primers | This study | 5´-GGGAAGAGGTAGAGACCCAAGT-3´ 5´-CCTCCATCAACACATAACGAAA-3´ | CRISPR/Cas9 Validation |
| Sequence-based reagent | MDGA2 CRISPR validation primers | This study | 5´-GCTGATAGGGAAGGACAGACAG-3´ 5´-TAAATCCAAGACTGCAAGAGCC-3´ | CRISPR/Cas9 validation |
| Sequence-based reagent | MDGA1 RTqPCR primers | This study | 5'-GTTCTACTGCTCCCTCAACC-3' 5'-CGTTACCTTTATTACCGCTGAG-3' | RTqPCR |
| Sequence-based reagent | MDGA2 RTqPCR primers | This study | 5'-AAGGTGACATCGCCATTGAC-3' 5'-CCACGGAATTCTTAGTTGGTAGG-3´ | RTqPCR |
| Sequence-based reagent | U6 RTqPCR primers | This study | 5'-GGAACGATACAGAGAAGATTAGC-3' 5'-AAATATGGAACGCTTCACGA-3' | RTqPCR |
| Sequence-based reagent | SDHA RTqPCR primers | This study | 5'-TGCGGAAGCACGGAAGGAGT-3' 5'-CTTCTGCTGGCCCTCGATGG-3' | RTqPCR |
| Sequence-based reagent | Template switching oligo | This study | 5'-AAGCAGTGGTATCAACGCAGAGTACrGrG + G-3' | Reverse transcription for RNA-seq |
| Genetic reagent | pCMV6-XL4 | Origene #pCMVXL4 | | MDGA1 expression |
| Recombinant DNA reagent | V5-MDGA1 (R.norvegicus) | A.M. Craig (University of British Columbia, Canada) *Pettem et al., 2013* | | Neuron electroporation |
| Recombinant DNA reagent | V5-MDGA2 (R.norvegicus) | A.M. Craig (University of British Columbia, Canada) | | Neuron electroporation |
| Recombinant DNA reagent | HA-MDGA1 (R.norvegicus) | A.M. Craig (University of British Columbia, Canada) *Pettem et al., 2013* | | Neuron electroporation |

*Continued on next page*

*Continued*

| Reagent type (species) or resource | Designation | Source or reference | Identifiers | Additional information |
|---|---|---|---|---|
| Recombinant DNA reagent | HA-MDGA2 (R.norvegicus) | A.M. Craig (University of British Columbia, Canada) *Pettem et al., 2013* | | Neuron electroporation |
| Recombinant DNA reagent | shMDGA1 (R.norvegicus) | A.M. Craig (University of British Columbia, Canada) *Pettem et al., 2013* | | Neuron electroporation |
| Recombinant DNA reagent | shMORB (R.norvegicus) | A.M. Craig (University of British Columbia, Canada) *Pettem et al., 2013* | | Neuron electroporation |
| Recombinant DNA reagent | HA-MDGA1 rescue (R.norvegicus) | A.M. Craig (University of British Columbia, Canada) *Pettem et al., 2013* | | Neuron electroporation |
| Recombinant DNA reagent | AP-NLGN1 (*M. musculus*) | A.Ting (Stanford University, USA) | | Neuron electroporation |
| Recombinant DNA reagent | BirA$^{ER}$ (*M. musculus*) | A.Ting (Stanford University, USA) | | Neuron electroporation |
| Recombinant DNA reagent | mApple.V5-MDGA2 rescue (R.norvegicus) | A.Ting (Stanford University, USA) *Loh et al., 2016* | | Plasmid construction |
| Recombinant DNA reagent | shMDGA2 (R.norvegicus) | A.Ting (Stanford University, USA) *Loh et al., 2016* | | Neuron electroporation |
| Recombinant DNA reagent | AP-MDGA1 rescue (R.norvegicus) | This paper | | Obtained with HD-In-Fusion kit |
| Recombinant DNA reagent | AP-MDGA2 rescue (R.norvegicus) | This paper | | Obtained with HD-In-Fusion kit |
| Recombinant DNA reagent | pSpCas9(BB)–2A-GFP (PX458) | Adgene #48,138 | RRID:Adgene_48138 | |
| Recombinant DNA reagent | shNLGN1 (*M. musculus*) | P.Scheiffele (Biozentrum, Basel) | | Neuron electroporation |
| Recombinant DNA reagent | HA-NLGN1 (*M. musculus*) | P.Scheiffele (Biozentrum, Basel) | | Obtained with HD-In-Fusion kit |
| Recombinant DNA reagent | AP-NLGN1 rescue (*M. musculus*) | *Chamma et al., 2016a* | | Neuron electroporation |
| Recombinant DNA reagent | Homer1c-DsRed | *Mondin et al., 2011* | | Neuron electroporation |
| Recombinant DNA reagent | GFP-NRXN1β | M.Missler (Münster University, Germany) *Neupert et al., 2015* | | Neuron electroporation |
| Recombinant DNA reagent | GFP-GPI | *Renner et al., 2009* | | Neuron electroporation |
| Recombinant DNA reagent | pCAG_Xph20-eGFP-CCR5TC | | RRID:Adgene_135530 | Neuron electroporation |
| Recombinant DNA reagent | pCAG_Xph20-mRuby2_CCR5TC | | RRID:Adgene_135531 | Neuron electroporation |
| Peptide, recombinant protein | Anti GFP nanobody | *Chamma et al., 2016a* | | GFP labelling uPAINT, dSTORM |

*Continued on next page*

*Continued*

| Reagent type (species) or resource | Designation | Source or reference | Identifiers | Additional information |
|---|---|---|---|---|
| Peptide, recombinant protein | mSA | *Chamma et al., 2017* | | AP-biotin labelling uPAINT, dSTORM |
| Cell line | COS-7 from ECACC | Sigma-Aldrich Acc Nc 87021302 Lot 15I032 passage + 4 | RRID:CVCL_0224 | Cell surface cluster assays, protein pull-down |
| Cell line | HEK-293T from ECACC | Sigma-Aldrich Acc Nc 12022001 Lot 16G020 Passage + 6 | RRID:CVCL_0063 | MDGA1 peptide production |
| Chemical compound, drug | Bicuculine | TOCRIS #0130/50 | RRID:SCR_003689 | Block inhibitory synaptic transmission |
| Chemical compound, drug | Phosphocreatine | Calcbiochem #2380–5 GM | | RNA extraction |
| Chemical compound, drug | Atto 647 N | Atto-Tec | | Coupled to anti GluA2 antibody |
| Chemical compound, drug | STAR 635 P | Abberior | | Coupled to mSA |
| Chemical compound, drug | Alexa Fluor 647 | Thermo Fischer Scientific | | Coupled to mSA |
| Chemical compound, drug | V5 tag Fab fragment | Abnova #RAB00032 | | Coupled to flurophores |
| Chemical compound, drug | QIAzol Lysis Reagent | Qiazol | | RTqPCR |
| Chemical compound, drug | X- tremeGENE | Transfection Reagent, Roche #6366546001 | RRID:SCR_001326 | COS7 cells transfection |
| Chemical compound, drug | Ribolock | Thermo Fisher Scientific #E00381 | | RNase inhibitor |
| Commercial assay, kit | In-Fusion HD Cloning Kit | Takara Bio #639,642 | | Plasmid construction |
| Commercial assay, kit | MycoAlert Mycoplasma Detection Kit | Lonza # LT07-218 | Lot number: 0000312202 | Mycoplasma detection in cell lines |
| Commercial assay, kit | T7 endonuclease based method | GeneArt Genomic detection kit, Thermo Fisher Scientific #A24372 | | CRISPR validation |
| Commercial assay, kit | Direct-Zol RNA microprep | Zymo Research cat#R2062 | | RTqPCR |
| Commercial assay, kit | Maxima First Strand cDNA Synthesis kit | Thermo Fischer Scientific # K1641 | | RTqPCR |
| Commercial assay, kit | Dynabeads Protein G | Thermo Fisher Scientific #11,004D | RRID:SCR_008452 | NLGN pull down |
| Commercial assay, kit | Dynabeads M-280 | Thermo Fisher Scientific #11,205D | | NLGN pull down |
| Commercial assay, kit | ClarityWestern blot ECL | Bio-Rad #170–5061 | | |
| Antibody | Actin (mouse monoclonal) | Sigma-Aldrich #A5316 | RRID:AB_476743 | (1:10,000 WB) |
| Antibody | βIII tubulin (rabbit polyclonal) | Abcam #18,207 | RRID:AB_444319 | (1:25,000 WB) |
| Antibody | GFP (mouse monoclonal) | Sigma-Aldrich #11814460001 | RRID:AB_390913 | (1:1000 WB) |
| Antibody | HA (rat monoclonal) | Roche #11867423001 | RRID:AB_390918 | (1:1000 WB) |
| Antibody | HA (rabbit monoclonal) | Cell Signaling #3,724 | RRID:AB_10693385 | |

*Continued on next page*

Continued

| Reagent type (species) or resource | Designation | Source or reference | Identifiers | Additional information |
|---|---|---|---|---|
| Antibody | MDGA1 (rabbit polyclonal) | Synaptic Systems #421,002 | RRID:AB_2800520 | 1:50 ICC (1:500 WB) |
| Antibody | GluA1 (rabbit polyclonal) | Agrobio (clone G02141) | - | (1:50 ICC) |
| Antibody | GluA2 (mouse monoclonal) | E.Gouaux (OSHU, Vollum Institute, Portland) | - | (1:200 ICC) |
| Antibody | NLGN1/2/3/4 (rabbit polyclonal) | Synaptic Systems #129,213 | RRID:AB_2619812 | NLGN pull down (1:1000 WB) |
| Antibody | PSD-95 (mouse monoclonal) | Thermo Fischer Scientific #MA1-046 | RRID:AB_2092361 | 1:200 ICC (1:2000 WB) |
| Antibody | Gephyrin (mouse monoclonal) | Synaptic Systems #147,111 | RRID:AB_887719 | (1:2000 ICC) |
| Antibody | VGLUT1 (guinea pig polyclonal) | Merck Millipore #AB5905 | RRID:AB_2301751 | (1:2000 ICC) |
| Antibody | VGAT (guinea pig polyclonal) | Synaptic Systems #131,004 | RRID:AB_887873 | (1:1,000 ICC) |
| Antibody | Synaptophysin (mouse monoclonal) | Sigma-Aldrich # S5768 | Clone SVP-38 RRID:AB_477523 | (1:2000 WB) |
| Antibody | Tubulin (mouse monoclonal) | Sigma-Aldrich #T4026 | RRID:AB_477577 | (1:5000 WB) |
| Antibody | MAP-2 (mouse monoclonal) | Sigma-Aldrich #M4403 | RRID:AB_477193 | (1:2000 ICC) |
| Antibody | p-tyrosine (mouse monoclonal) | Cell Signaling #9,411 | RRID:AB_331228 | (1:1000 WB) |
| Antibody | Anti-mouse HRP (donkey polyclonal) | Jackson Immunoresearch #715-035-150 | RRID:AB_2340770 | (1:5000 WB) |
| Antibody | Anti-mouse IRDye680LT (goat polyclonal) | LICOR #926–68020 | RRID:AB_10706161 | (1:10,000 WB) |
| Antibody | Anti-mouse IRDye 800LT (goat polyclonal) | LICOR #926–32210 | RRID:AB_26218442 | (1:10,000 WB) |
| Antibody | Anti-rabbit HRP (donkey polyclonal) | Jackson Immunoresearch #711-035-152 | RRID:AB_10015282 | (1:5000 WB) |
| Antibody | Anti-rabbit IRDye 680LT (goat polyclonal) | LICOR #926–68021 | RRID:AB_10706309 | (1:10,000 WB) |
| Antibody | Anti-rabbit IRDye800LT (goat polyclonal) | LICOR #926–32211 | RRID:AB_621843 | (1:10,000 WB) |
| Antibody | Anti-mouse Alexa488 (goat polyclonal) | Thermo Scientific #A11001 | RRID:AB_2534069 | (1:1,000 ICC) |
| Antibody | Anti-mouse Alexa 568 (goat polyclonal) | Thermo Scientific #A11031 | RRID:AB_144696 | (1:1,000 ICC) |
| Antibody | Anti-mouse Alexa 647 (goat polyclonal) | Thermo Scientific #A21235 | RRID:AB_2535804 | (1:1,000 ICC) |
| Antibody | Anti-rabbit Alexa488 (goat polyclonal) | Thermo Scientific #A11008 | RRID:AB_143165 | (1:1,000 ICC) |
| Antibody | Anti-rabbit Alexa 568 (goat polyclonal) | Thermo Scientific #A11011 | RRID:AB_143157 | (1:1,000 ICC) |
| Antibody | Anti-rabbit Alexa 647 (goat polyclonal) | Thermo Scientific #A31576 | RRID:AB_10374303 | (1:1,000 ICC) |
| Antibody | Anti-rabbit Alexa 555 (donkey polyclonal) | Invitrogen #A32794 | RRID:AB_2762834 | (1:1,000 ICC) |

Continued

| Reagent type (species) or resource | Designation | Source or reference | Identifiers |
|---|---|---|---|
| Antibody | Anti-rat Alexa 488 (goat polyclonal) | Thermo Scientific #A11073 | RRID:AB_2534117 |
| Antibody | Anti-rat Alexa 568 (goat polyclonal) | Thermo Scientific #A11077 | RRID:AB_2534121 |
| Antibody | Anti Rat Alexa 647 (goat polyclonal) | Thermo Scientific #A21247 | RRID:AB_141778 |
| Antibody | Anti guinea pig DyLight 405 (goat polyclonal) | Jackson ImmunoResearch #106475003 | RRID:AB_2337432 |
| Antibody | Anti-guinea pig Alexa488 (goat polyclonal) | Thermo Scientific #A11008 | RRID:AB_143165 |
| Antibody | Anti-guinea pig Alexa 568 (goat polyclonal) | Thermo Scientific #A11075 | RRID:AB_141954 |
| Antibody | Anti guinea pig Alexa 647 (goat polyclonal) | Thermo Scientific #A21450 | RRID:AB_2141882 |
| Software, algorithm | Metamorph | Molecular devices | RRID:SCR_002368 | Image analysis |
| Software, algorithm | Graphpad | PRISM | RRID:SCR_002798 | Statistical analysis |
| Software, algorithm | ChopChop | Labun et al., 2019 | | CRISPR/Cas9 design |
| Software, algorithm | ImageJ | National Center for Microscopy and Imaging Research | RRID:SCR_001935 | Western-blot quantitation |
| Software, algorithm | cutadapt | Martin, 2011 | | RNA-seq |
| Software, algorithm | fastp | Chen et al., 2018 | | RNA-seq |
| Software, algorithm | STAR | Dobin et al., 2013 | | RNA-seq |
| Software, algorithm | DESeq2 | Love et al., 2014 | | RNA-seq |

**Table 1.** CRISPR off-targets of MDGA1 gRNA with 4 mismatches.

| Name off-target sequence for MDGA1 gRNA | Additional information | CFD score |
|---|---|---|
| intergenic_Vangl2 | (1:1,000 ICC) | 0.36 |
| intron_Tbxas1 | (1:1,000 ICC) | 0.29 |
| intergenic_SCCPDH\|TFB2M | | 0.21 |
| intergenic_Scg2\|ENSRNOG00000097663 | (1:1,000 ICC) | 0.18 |
| intergenic_ENSRNOG00000037633\|Nyap2 | (1:1,000 ICC) | 0.17 |
| intron_Galnt18_1 | | 0.16 |
| intergenic_Ttc7b\|Rps6ka5 | (1:1,000 ICC) | 0.16 |
| intergenic_Six2\|Srbd1 | | 0.12 |
| intron_Slc9a9_8 | | 0.11 |
| intergenic_Adora1\|Myog_13 | (1:1,000 ICC) | 0.1 |

## DNA constructs

Rat V5-MDGA1, V5-MDGA2, HA-MDGA1, HA-MDGA2, shMDGA1, shMORB (shCTRL), sh-RNA resistant HA-MDGA1 (rescue) constructs as described previously (*Elegheert et al., 2017*; *Pettem et al., 2013*) were kind gifts from A.M. Craig (University of British Columbia, Vancouver, OR). Mouse AP-tagged NLGN1, biotin ligase (BirA^ER), shMDGA2, and mApple-V5-MDGA2 rescue plasmids (*Loh et al., 2016*) were gifts from A. Ting (Stanford University, CA). AP-MDGA1 and AP-MDGA2 were generated by replacing the V5 tag of the V5-MDGA1 and V5-MDGA2 constructs, respectively, by the 15 amino acids AP tag (5′-GGCCTGAACGATATCTTCGAGGCCCAGAAGATCGAGTGGCACGAG-3′) using the HD-In-Fusion kit (Takara). The linker 5′-G GAGGATCAGGAGGATCA-3′ was added after the AP tag. AP-MDGA1 and AP-MDGA2 rescue constructs were generated by inserting the muta- tions responsible for the resistance to the respec- tive shRNAs obtained from HA-MDGA1 and mApple-V5-MDGA2 rescue constructs, respec- tively, using the HD-In-Fusion kit. HA-MDGA2 rescue was created by replacing the AP tag from the AP-MGDA2 rescue construct by the HA tag using the HD-In-Fusion kit.

The CRISPR target sequences were all 20-nucleotide long and followed by a proto- spacer adjacent motif (PAM). The first step in the design of gRNAs involved identification of the best sequence to target. For MDGA1, we chose the more efficient gRNA proposed by the online software ChopChop (https://chopchop.cbu.uib. no/). For MDGA2, we chose to target the exon1,

**Table 2.** CRISPR off-targets of MDGA2 gRNA with 4 mismatches.

| Name off target sequence for MDGA2 gRNA | CFD score |
|---|---|
| exon_Pex1_chr4 | 0.44 |
| intron_Ssbp2_chr2 | 0.25 |
| intergenic_Prune2\|LOC102546963_chr1 | 0.2 |
| intergenic_Crygs\|AABR07034636.1_chr11 | 0.17 |
| intron_Ift80_chr2 | 0.15 |
| intergenic_LOC685114\|Ccdc178_chr18 | 0.15 |
| intergenic_U6\|AABR07048636.1_chr5 | 0.14 |
| intron_Tmem47_chrX | 0.13 |
| intron_Camk2b_chr14 | 0.07 |
| intergenic_Samd4a\|Gch1_chr15 | 0.07 |

near ATG (*Labun et al., 2019*). The guide RNA (gRNA) sequence was 5'-CTTCAACGTACGAGCC CGGG-3' for MDGA1 and 5'-ATTTAGTGTACGGTCTCGTG-3' for MDGA2. As a control we used a sequence from a gecko bank: 5'-ATATTTCGGCAGTTGCAGCA-3'. We took particular care to select gRNAs not sharing homology with any other sequence in the genome, even considering 0, 1, 2, or 3 mismatches, and with high efficiency scores (cutting frequency determination, CFD = 98/100). However, there are still a number of genes to which gRNAs for MDGA1 or MDGA2 can hybridize with 4 mismatches, albeit with low scores i.e. CFD < 0.4 (*Tables 1 and 2*). gRNAs were cloned into the vector pSpCas9(BB)–2A-GFP (PX458) (Addgene cat#48138). The CRISPR MDGA2 resitant sequence was the same as for shMDGA2 since the gRNA for MDGA2 was directed to the signal peptide, which is absent in the HA-MDGA2 rescue described above.

Short hairpin RNA to murine NLGN1 (shNLGN1) (*Chih et al., 2005*) and HA-NLGN1 were gifts from P. Scheiffele (Biozentrum, Basel). shRNA-resistant AP-tagged NLGN1 (AP-NLGN1res) was described previously (*Chamma et al., 2016a*; *Letellier et al., 2018*). GFP-NRXN1β (*Chamma et al., 2016a*; *Neupert et al., 2015*) was a gift from M. Missler (Münster University, Germany). Homer1c-DsRed and GFP-GPI constructs were reported earlier (*Mondin et al., 2011*; *Renner et al., 2009*). Xph20-GFP and Xph20-mRuby2 (Addgene#135,530 pCAG_Xph20-eGFP-CCR5TC, #135,531 pCAG_Xph20-mRuby2-CCR5TC) have previously been described (*Rimbault et al., 2019*).

## Production and fluorophore conjugation of probes

The anti-GFP nanobody and mSA were produced as described (*Chamma et al., 2017*). Briefly, the two proteins were expressed in *E. coli* by auto-induction at 16°C. Both proteins were purified by affinity chromatography using their polyhistidine tags in native and denaturing conditions for the nanobody and mSA respectively. After dialysis in PBS and concentration to ~ 1 mg.mL$^{-1}$, the proteins were coupled with 3–6 equivalents of the dyes in their activated ester form. Dyes used were Atto 647 N (Atto-Tec), STAR 635 P (Abberior) and Alexa Fluor 647 (ThermoFisher). Excess unreacted dye was removed using a desalting column and the dye-conjugated probes were further purified to homogeneity by size-exclusion chromatography. Probes were concentrated and flash-frozen for storage at −80 °C until use. The anti-GluA2 antibody, clone 15F1 (gift from E. Gouaux, OSHU, Vollum Institute, Portland), and the V5 tag recombinant Fab fragment (Abnova, RAB00032) were coupled to NHS-derived dyes using the same protocol as above but without the size-exclusion chromatography purification step.

## MDGA1 recombinant protein production and rabbit polyclonal antiserum

For antibody production, mouse MDGA1 cDNA lacking signal peptide, GPI anchor site, and propeptide (amino acids 19–932; Uniprot ID# Q0PMG2) was inserted in-frame in a modified pCMV6-XL4 expression vector containing a prolactin leader peptide (PLP) followed by a N-terminal FLAG tag, MDGA1 insert, a 3CPro cleavage site and the human Fc domain. Secreted dimeric C-terminally Fc-tagged MDGA1 stably expressed in HEK-293T cells was collected in serum-free Opti-MEM (Thermo Fisher Scientific, Inc). HEK-293T cells from the European Collection of Authenticated Cell Cultures (ECACC) were purchased via Sigma-Aldrich (Acc Nc 12022001). Cells were thawed from frozen vials at passage + 6, and maintained up to passage 20. Cells were regularly tested negative for mycoplasma, using the MycoAlert detection kit (Lonza, #LT07-218). Fc-tagged MDGA1 protein was run on an affinity column packed with Protein-G Plus Agarose fast flow resin (Pierce) using a gravity-flow system. Affinity column was washed with 250 mL wash buffer (50 mM HEPES pH 7.4, 300 mM NaCl) and eluted with 10 mL IgG elution buffer (Pierce) per the manufacturer's instructions. For non-Fc-tagged MDGA1 protein used for immunization, following passage of conditioned medium through the column packed with Protein-G Agarose, the column was washed with 250 mL wash buffer (450 mM NaCl, 50 mM Tris, 1 mM EDTA, pH 8.0), the Fc tag was cleaved by overnight incubation with GST-tagged 3 C PreScission Protease (GE Healthcare) in cleavage buffer (150 mM NaCl, 50 mM Tris, 1 mM EDTA, 1 mM DTT, pH 8.0), and the cleaved protein was collected in the eluate. The protease was subsequently separated from the eluted proteins using a Glutathione Sepharose (GE Healthcare) packed column. Fc-tagged and non-Fc-tagged proteins were concentrated using Amicon Ultra 10 kDa MWCO centrifugal filter units (Millipore), dialyzed against PBS, and protein concentration determined by Bradford assay (Bio-Rad).

Immunization of rabbits and harvesting of polyclonal antiserum was performed by Synaptic Systems (MDGA1 polyclonal antiserum #421 002).

## Immunohistochemistry

Vibratome sections (80 μm) from the brains of either adult wildtype mice or *Mdga1* KO mice (*Ishikawa et al., 2011*) (a gift of T. Yamamoto, Kagawa University, Japan) were permeabilized at room temperature (RT) for 40 min in PBS containing 0.5% Triton X-100 (Sigma-Aldrich). Sections were then blocked overnight at 4 °C in PBS containing 10% normal horse serum (NHS), 0.5% Triton X-100, 0.5 M glycine (Sigma-Aldrich, #G8898), 0.2% gelatin (Sigma-Aldrich, #G7041). Sections were then washed in PBS-0.5% Triton X-100 at RT and incubated at 4 °C for 48 hr with MDGA1 antiserum (dilution 1:500) in PBS containing 5% NHS, 0.5% Triton X-100% and 0.2% gelatin. Afterwards, sections were washed in PBS-0.5%Triton X-100 at RT before overnight incubation at 4 °C with Alexa555-conjugated donkey-anti-rabbit antibody (Invitrogen, #A32794) in PBS containing 5% NHS, 0.5% Triton X-100% and 0.2% gelatin. Before mounting coverslips with Mowiol-4–88 (Sigma-Aldrich), sections were washed in PBS-0.5% Triton X-100. Images were acquired using a Leica SP8 confocal microscope (Leica Microsystems).

## Rat hippocampal cultures and electroporation

Gestant Sprague-Dawley rat females were purchased from Janvier Labs (Saint-Berthevin, France). Animals were handled and killed according to European ethical rules. Dissociated neuronal cultures were prepared from E18 rat embryos or P0 mice as previously described (*Kaech and Banker, 2006*). Dissociated cells were electroporated with the Amaxa system (Lonza) using 300,000 cells per cuvette. Depending on the experiments, the following plasmid combinations were used: 1 / Homer1c-DsRed: shMDGA1 or shMDGA2: AP-MDGA1 rescue or AP-MDGA2-rescue: BirA$^{ER}$ (1:3:1:1 μg DNA); 2 / Homer1c-DsRed and GFP-GPI (1:1 μg DNA); 3 / Homer1c-DsRed plus shCTRL (shMORB), shMDGA1, or shMDGA2 (1:3 μg DNA); 4 / in separate electroporations, GFP-NRXN1β (3 μg DNA) and Xph20-mRuby2 plus shCTRL or shMDGA2 (1:3 μg DNA); 5 / Homer1c-DsRed: shCTRL, shMDGA1 or shMDGA2: AP-NLGN1: BirA$^{ER}$ (1:3:1:1 μg DNA); 6 / Homer1c-DsRed: BirA$^{ER}$: shCTRL or shMDGA2: AP-NLGN1: HA-MDGA2 rescue (1:1:3:1:1 μg DNA); 7 / Xph20-mRuby2: CRISPR/Cas9 CONTROL, CRISPR/Cas9 MDGA1, or CRISPR/Cas9 MDGA2: HA-MDGA2 rescue (1:3:1 μg DNA). Electroporated neurons were resuspended in Minimal Essential Medium (Thermo Fisher Scientific, #21090.022) supplemented with 10% Horse serum (Invitrogen) (MEM-HS), and plated on 18 mm glass coverslips coated with 1 mg.mL$^{-1}$ polylysine (Sigma-Aldrich, #P2636) overnight at 37 °C. Three hours after plating, coverslips were flipped onto 60 mm dishes containing 15 DIV rat hippocampal glial cells cultured in Neurobasal plus medium (Gibco Thermo Fisher Scientific, #A3582901) supplemented with 2 mM glutamine and 1 x B27 plus Neuronal supplement (Gibco Thermo Fisher Scientific, #A3582801). Cells were cultured during 8–14 days at 37 °C and 5% CO$_2$. Astrocyte feeder layers were prepared from the same embryos, plated between 20,000 and 40,000 cells per 60 mm dish previously coated with 0.1 mg.mL$^{-1}$ polylysine and cultured for 10 days in MEM containing 4.5 g.L$^{-1}$ glucose, 2 mM L-glutamax (Thermo Fischer Scientific, #35050–038) and 10% horse serum. Ara C (Sigma-Aldrich, #C1768) was added after 3 DIV at a final concentration of 3.4 μM.

## Genomic cleavage of CRISPR constructs

To validate the genomic cleavage, we used a T7 endonuclease based method (GeneArt Genomic detection kit, Thermo Fisher Scientific, #A24372). Briefly, dissociated hippocampal neurons were electroporated as described above with the CRISPR/Cas9 constructs and plated on glass coverslips. After 10 DIV, neurons were scraped in PBS and centrifuged at 1000xg for 5 min. Cells were then resuspended in 50 μL of lysis buffer containing 2 μL of protein degrader to extract genomic DNA. Then, PCRs were run to amplify a 555 bp genomic segment for MDGA1 and 546 bp for MDGA2. The following pairs of primers were used: MDGA1: Forward: 5′-GGGAAGAGGTAGAGACCCAAGT-3′ Reverse: 5′-CCTCCATCAACACATAACGAAA-3′. MDGA2: Forward: 5′-GCTGATAGGGAAGGACAGACAG-3′; Reverse: 5′-TAAATCCAAGACTGCAAGAGCC-3′. After checking the presence of the PCR fragments in an agarose gel, 1 μL of PCR reaction was denatured, reannealed, and digested with T7 endonuclease to reveal the presence of mismatches in the annealed fragments. Cleavage bands were detected in agarose gels.

## RT-qPCR

RNA was extracted from Banker neuronal cultures using the QIAzol Lysis Reagent (Qiagen) and the Direct-Zol RNA microprep (Zymo Research, cat#R2062) per the manufacturer's instructions. cDNA was synthetized using the Maxima First Strand cDNA Synthesis kit (Thermo Fischer Scientific, # K1641). At least three neuronal cultures were analyzed per condition and triplicate qPCR reactions were made for each sample. Transcript-specific primers were used at 600 nM and cDNA at 10 ng in a final volume of 10 µL. The LightCycler 480 SYBR Green I Master qPCR kit (Roche) was used according to manufacturer's instructions. The Ct value for each gene was normalized against that of SDHA and U6. The relative mRNA expression level was calculated using the comparative method (2ΔΔCt) (*Livak and Schmittgen, 2001*). The following set of primers were used: MDGA1 Forward: 5'-GTTCTACTGCTC CCTCAACC-3' Reverse: 5'-CGTTACCTTTATTACCGCTGAG-3' MDGA2 Forward: 5'-AAGGTGACA TCGCCATTGAC-3' Reverse: 5'-CCACGGAATTCTTAGTTGGTAGG-3' U6 Forward: 5'-GGAACGATA CAGAGAAGATTAGC-3' U6 Reverse: 5'-AAATATGGAACGCTTCACGA-3' SDHA Forward: 5'-TGCGG AAGCACGGAAGGAGT-3' SDHA Reverse: 5'-CTTCTGCTGGCCCTCGATGG-3'.

## Culture and transfection of COS-7 cells

COS-7 cells from the ECACC purchased via Sigma-Aldrich, Acc Nc 87021302 were cultured in DMEM (Eurobio) supplemented with 1% glutamax (Thermo Fischer Scientific, #35050–038), 1% sodium pyruvate (Thermo Fischer Scientific, #11360–070), 10% Fetal Bovine Serum (Eurobio). Cells were thawed from frozen vials at passage + 4, and maintained up to passage 20. Cells were regularly tested negative for mycoplasma, using the MycoAlert detection kit (Lonza, #LT07-218). For streptavidin pull-down and Western blots, COS-7 cells were plated in 6-well plates (100,000 cells/well) and transfected the next day using X- tremeGENE 9 DNA (Transfection Reagent, Roche), with HA-NLGN1 + AP-MDGA1 or AP-MDGA2 + BirA$^{ER}$ (total DNA 1 µg/well). Cells were left under a humidified 5% $CO_2$ / 37 °C atmosphere for 2 days before being processed for immunoprecipitation. For imaging experiments and shRNA validation experiments, cells were electroporated with the Amaxa system (Lonza) using the COS-7 ATCC program. Typically, 500,000 cells were electroporated with: 2 µg HA-MDGA1 or HA-MDGA2; 2, 4, 6 µg shMDGA1 or shMDGA2; and 2 µg HA-MDGA1 or HA-MDGA2 rescue. After 24 hr, cells were processed for imaging or biochemistry.

## Neuronal lysates and brain tissue subcellular fractionation

For biochemistry experiments, hippocampal neurons were plated at a density of 500,000 cells per well in a 6-well plate previously coated with 1 mg.mL$^{-1}$ polylysine for 24 hr at 37 °C. Cells were cultured for 7, 14 and 21 DIV in Neurobasal medium supplemented with 2 mM glutamine and 1 x NeuroCult SM1 Neuronal supplement. After 3 DIV, Ara C was added to the culture medium at a final concentration of 3.4 µM. Before lysis, plates were rinsed once in ice cold PBS and then scraped into 100 µL of RIPA buffer (50 mM Tris-HCl pH 7.5, 1 mM EDTA, 150 mM NaCl, 1% Triton-X100) containing protease inhibitor Cocktail Set III (Millipore #539134). Homogenates were kept for 30 min on ice, then centrifuged at 8,000xg for 15 min at 4 °C to remove cell debris. Protein concentration was estimated using the Direct Detect Infrared Spectrophotometer (Merck-Millipore). 100 µg protein were loaded on a gel to detect endogenous MDGA molecules in Western blots. For all other proteins, 20 µg were loaded. Rat brain subcellular fractionation was performed as previously described (*Condomitti et al., 2018*).

## NLGN immunoprecipitation in neuronal cultures

Dissociated cells were electroporated with the Amaxa system (Lonza) using $1.5 \times 10^6$ cells per cuvette and 8 µg of shCONTROL or shMDGA2. Electroporated cells were plated at a density of 500,000 cells per well in a 6-well plate. At DIV 10 cells were treated with 3 mM pervanadate for 15 min at 37 °C before lysis, to preserve phosphate groups on NLGNs. Whole-cell protein extracts were obtained by solubilizing cells in lysis buffer (50 mM HEPES, pH 7.2, 10 mM EDTA, 0.1% SDS, 1% NP-40, 0.5% DOC, 2 mM Na-Vanadate, 35 µM phenylarsine oxide, 48 mM Na-Pyrophosphate, 100 mM NaF, 30 mM phenyl-phosphate, 50 µM NH$_4$-molybdate, 1 mM ZnCl$_2$) containing protease Inhibitor Cocktail Set III (Millipore #539134). Homogenates were kept on ice for 15 min. Lysates were then clarified by centrifugation at 8000 × g for 15 min. For immunoprecipitations, 500–1000 µg of total protein (estimated by Direct Detect Infrared Spectrophotometer assay, Merck Millipore), were incubated overnight with 2 µg of antibody raised against an intracellular epitope in mouse NLGN1 (amino acids 826–843) and

which recognizes all NLGNs 1/2/3/4 (Synaptic Systems, #129 213). Antibody-bound NLGNs were incubated for 1 hr with 20 µL of protein G beads (Dyna- beads Protein G, Thermo Fisher Scientific) precipitated and washed four times with lysis buffer. At the end of the immunoprecipitation, 20 µL beads were resuspended in 20 µL of Laemli Sample Buffer buffer 2 X (Biorad, #1610747), and super-natants were processed for SDS-PAGE and Western blotting.

## Streptavidin pull-down

Biotinylated AP-tagged MDGA1 or NLGN1 expressing COS-7 cells were rinsed once in ice cold PBS and then scraped in 100 µL RIPA buffer (50 mM Tris-HCl pH 7.5, 1 mM EDTA, 150 mM NaCl, 1% Triton-X100) containing protease inhibitor cocktail (Millipore #539134). Homogenates were kept on ice for 30 min, then centrifuged at 8,000xg for 15 min at 4 °C to remove cell debris. The supernatant was recovered and the protein concentration was estimated using the Direct Detect Infrared Spec-trophotometer (Merck-Millipore). 400 µg protein were incubated with 40 µL of streptavidin coupled Dynabeads M-280 (Thermo Fisher Scientific, #11,205D). After 1 hr of incubation at RT on a rotating wheel, tubes were placed in the magnetic column and beads were washed three times with lysis buffer. Proteins were eluted from the beads by directly adding 20 µL of Laemli Sample Buffer buffer 2 X (Biorad, #1610747). Samples were then processed for SDS-PAGE and Western blotting.

## SDS-PAGE and western blotting

Samples were loaded in acrylamide-bisacrylamide 4–20% gradient gels (PROTEAN TGX Precast Protein Gels, BioRad) and run at 100 V for 1 hr. Proteins were transferred to a nitrocellulose membrane for immunoblotting using the TurboBlot system (BioRad). After 1 hr blocking with 5% non-fat dry milk in Tris-buffered saline Tween-20 solution (TBST: 28 mM Tris, 137 mM NaCl, 0.05% Tween-20, pH 7.4), membranes were incubated during 1 hr RT or overnight at 4 °C, with the primary antibody diluted in TBST solution containing 1% dry milk: custom-made rabbit anti-MDGA1 (Synaptic Systems #421002), rabbit anti-HA (Cell Signaling #3,724 (C29F4), 1:1000), rat anti-HA (Roche #1186742300, 1:1000), mouse anti-actin (Sigma-Aldrich #A5316, 1:10,000), mouse anti-GFP (Sigma-Aldrich #11814460001, 1:1000), rabbit anti-ßIII tubulin (Abcam #ab18207, 1:25,000), mouse anti-tubulin (Sigma-Aldrich #T4026, 1:5000), mouse anti-PSD95 (clone 7E3-1B8, Thermo Fisher Scientific #MA1-046, 1:2000), mouse anti-Synaptophysin (SVP-38) (Sigma-Aldrich # S5768, 1:2000), mouse anti-pTyr (Cell Signaling #9411, 1:1000), rabbit anti-panNLGN (Synaptic Systems #129 213, 1:1000). After three washes in TBST, membranes were incubated with horseradish peroxidase-conjugated donkey anti-mouse or anti-rabbit secondary antibodies (Jackson Immunoresearch, #715-035-150 and #711-035-152, respec-tively, concentration: 1:5000) or fluorophore-conjugated goat anti-mouse or anti-rabbit secondary antibodies (IRDye 680LT anti-rabbit #926–6821, IRDye 680LT anti-mouse #926–68020, IRdye-800CW anti-rabbit #926–32211, IRdye-800CW anti-mouse #926–32210 LI-COR) for 1 hr at RT. Target proteins were detected by chemiluminescence with Clarity Western ECL Substrate (Bio-Rad #170–5061) on the ChemiDoc Touch System (BioRad) or Odyssey Fc Imaging System (LI-COR) for fluorescent secondary antibodies. For quantification of band intensities, images were processed with the Gels tool of ImageJ. Normalization of protein loading was done using endogenous actin or tubulin present in the samples.

## Immunocytochemistry

To visualize endogenous MDGA1 proteins and AMPARs at the cell surface, neurons were incubated live for 10 min at 37 °C with the respective antibodies (rabbit anti-MDGA1, Synaptic Systems #421002 1:50; rabbit anti-GluA1, Agrobio, clone G02141, 0.2 mg.mL$^{-1}$, 1:50; mouse anti-GluA2, clone 15F1, gift from E. Gouaux, 1:200), all diluted in Tyrode solution (15 mM D-glucose, 108 mM NaCl, 5 mM KCl, 2 mM MgCl$_2$, 2 mM CaCl$_2$ and 25 mM HEPES, pH = 7.4, 280 mOsm) containing 1% bovine serum albumin (BSA) (Sigma A7030). Then, cultures were fixed for 15 min in 4% paraformaldehyde, 4% sucrose, quenched in NH$_4$Cl 50 mM in PBS for 15 min, permeabilized for 5 min with 0.1% Triton X-100 in PBS. After blocking during 20 min in PBS containing 1% BSA, cells were counter-stained for pre- and post-synaptic markers with a mixture of the following primary antibodies: mouse anti-PSD-95 (Thermo Fisher Scientific, #MA1-046, 1:100); mouse anti-gephyrin (Synaptic Systems, #147111, 1:2000); guinea pig anti-VGAT (Synaptic Systems, #131004, 1:1000), and guinea pig anti-VGLUT1 (Merck Millipore, #AB5905, 1:2000). The neuronal microtubule cytoskeleton was labeled using mouse anti MAP-2 (Sigma Aldrich, #M4403, 1:2000). Following three washes in PBS, cells were incubated

with appropriate secondary antibodies coupled to Alexa fluorophores (405, 488, 564, or 647) (Thermo Fisher Scientific).

MDGA1 immunostainings were visualized on a commercial Leica DMI6000 TCS SP5 microscope using a 63x/1.4 NA oil objective and a pinhole opened to one Airy disk. Images of 1024 × 1024 pixels were acquired at a scanning frequency of 400 Hz. All other immunostainings were visualized using an inverted epifluorescence microscope (Nikon Eclipse TiE) equipped with a 60 x/1.45 NA objective and filter sets for BFP (Excitation: FF02-379/34; Dichroic: FF-409Di03; Emission: FF01-440/40); EGFP (Excitation: FF01-472/30; Dichroic: FF-495Di02; Emission: FF01-525/30); Alexa568 (Excitation: FF01-543/22; Dichroic: FF-562Di02; Emission: FF01-593/40); and Alexa 647 (Excitation: FF02-628/40; Dichroic: FF-660Di02; Emission: FF01-692/40) (SemROCK). Images were acquired with an sCMOS camera (PRIME 95B, Photometrics) driven by the Metamorph software (Molecular Devices). The number of PSD-95, VGLUT1, gephyrin, and VGAT puncta per dendrite length was measured using a custom macro written in Metamorph. Briefly, epifluorescence images of pre- and post- synaptic markers were first thresholded and segmented using the morphometric image analysis module of MetaMorph for structures bigger than 4 pix² (0.137 μm²). Then, the total length of the dendrite was measured with the free line drawing tool of MetaMorph, and the linear pre- and postsynaptic density was calculated.

## Single molecule tracking (UPAINT)

Universal Point Accumulation in Nanoscale Topography (uPAINT) experiments were performed as previously described (*Chamma et al., 2016a*). In brief, neuronal cultures were placed in a Inox Ludin chamber (Life Imaging Services) containing pre-warmed Tyrode solution supplemented with 1% biotin-free BSA (Roth #0163.4). The chamber was placed on a motorized inverted microscope (Nikon Ti-E Eclipse) enclosed in a thermostatic box (Life Imaging Services) providing air at 37 °C. Biotinylated AP tags in MDGA1, MDGA2 and NLGN1 were labelled with STAR 635P-conjugated mSA at a concentration of 1 nM; N-terminal V5 tags in MDGA1, MDGA2 and LRRTM2 were labeled with 1 nM recombinant Fab fragment coupled to STAR 635 P (Abnova, #RAB00032). GFP-GPI or GFP-NRXN1β were labeled with 1 nM anti-GFP nanobody conjugated to Atto 647 N. Endogenous AMPARs were labelled with a low concentration (~1 nM) of Atto 647N-conjugated anti-GluA2 antibodies. A four-color laser bench (405; 491; 561; and 647 nm, 100 mW each; Roper Scientific) is connected through an optical fiber to the TIRF illumination arm of the microscope and laser powers are controlled through acousto-optical tunable filters driven by Metamorph. The fluorophores STAR 635 P and Atto 647 N were excited with the 647 nm laser line through a four-band beam splitter (BS R405/488/561/635, SemRock). Samples were imaged by oblique laser illumination, allowing the excitation of individual fluorescent probes (mSA, V5 Fab, anti-GluA2) bound to the neuron surface, with minimal background coming from the probes in solution. Fluorescence light was collected through a 100 X/1.49 NA PL-APO objective using a FF01-676/29 nm emission filter (SemRock), placed on a filter wheel (Suter Instruments). Image stacks of 2000–4000 consecutive frames with an integration time of 20 ms, were acquired with an EMCCD camera working at 10 MHz and Gain 300 (Evolve, Photometrics).

## dSTORM

AP-tagged proteins were labelled for dSTORM using 100 nM mSA-Alexa 647 in Tyrode solution containing 1% biotin free-BSA (Roth #0163.4) for 10 min at 37 °C. V5-tagged proteins were labelled using 100 nM Alexa 647-conjugated anti-V5 Fab. GFP-GPI was labeled using 100 nM anti-GFP nanobody conjugated to Alexa 647. Cells were rinsed and fixed with 4% PFA–0.2% glutaraldehyde in PBS-sucrose 4% for 10 min at RT, then kept in PBS at 4 °C until dSTORM acquisitions. Neurons were imaged in Tris-HCl buffer (pH 7.5), containing 10% glycerol, 10% glucose, 0.5 mg.mL⁻¹ glucose oxidase (Sigma), 40 mg.mL⁻¹ catalase (Sigma C100-0,1% w/v) and 50 mM β-mercaptoethylamine (MEA) (Sigma M6500) (*Heilemann et al., 2008*). The same microscope described above for uPAINT was used. This microscope is further equipped with a perfect focus system preventing drift in the z-axis during long acquisition times. Pumping of Alexa 647 dyes into their triplet state was performed for several seconds using ~ 60 mW of the 647 nm laser at the objective front lens. Then, a lower power (~30 mW) was applied to detect the stochastic emission of single-molecule fluorescence, which was collected using the same optics and detector as described above. Multicolour Tetraspec fluorescent 100 nm beads (Invitrogen, #T7279) or nano-diamonds (Adamas Nanotechnologies #ND-NV140 nm)

were added to the sample for later registration of images and lateral drift correction. Single-molecule detection was performed online with automatic feedback control of the lasers using the WaveTracer module running in Metamorph, enabling optimal single-molecule density during the acquisition. Acquisition sequences of 64,000 frames were acquired in streaming mode at 50 frames per second (20 ms exposure time), thus representing a total time of 1280 s = 21 min.

## Offline single-molecule detection, trajectory analysis, and image reconstruction

Analysis of the image stacks generated by uPAINT and dSTORM was made offline under Metamorph, using the PALM-Tracer program based on wavelet segmentation for single-molecule localization and simulated annealing algorithms for tracking (*Izeddin et al., 2012*; *Kechkar et al., 2013*). For the analysis of uPAINT experiments, the instantaneous diffusion coefficient, D, was calculated for each trajectory from linear fits of the first 4 points of the mean square displacement (MSD) function versus time, for trajectories containing at least 10 points. For very confined trajectories, the fit of the MSD function can give negative values for diffusion coefficients: in that case, D is arbitrarily set at $10^{-5}$ µm²/s. The uPAINT sequences were also represented as density maps integrating all individual molecule detections. These super-resolved images were constructed using a zoom factor of 5, that is with a pixel size of 32 nm which is five times smaller than that of the original image (0.16 µm) and corresponds to the pointing accuracy of our system. To sort individual trajectories among synaptic and extra-synaptic compartments, post-synapses were identified by wavelet-based image segmentation (*Racine et al., 2006*) of the Homer1c-DsRed signal, and the corresponding binary masks were transferred to the single-molecule images for analysis. Synaptic coverage was determined from super-resolved detection maps as the ratio between segmented areas containing detections over the whole synaptic region determined from the low resolution Homer1c-DsRed image. dSTORM stacks were analyzed using the PALM-Tracer program, allowing the reconstruction of a unique super-resolved image of 32 nm pixel size (zoom 5 compared to the original images) by summing the intensities of all localized single molecules (1 detection per frame is coded by an intensity value of 1). The localization precision of our imaging system in dSTORM conditions is around 60 nm (FWHM) (*Lagardère et al., 2020*). To analyze protein enrichment at post-synapses, the average number of detections within Homer1c puncta was divided by the the average number of extra-synaptic detections, both normalized per unit area.

## Organotypic slice culture and single-cell electroporation

Organotypic hippocampal slice cultures were prepared from postnatal day 2 Sprague-Dawley rats (*Stoppini et al., 1991*). Animals were quickly decapitated and brains placed in ice-cold Gey's balanced salt solution under sterile conditions. Hippocampi were dissected out and coronal slices (350 µm) were cut using a tissue chopper (McIlwain) and incubated at 35 °C with serum-containing medium on Millicell culture inserts (CM, Millipore). The medium was replaced every 2–3 days. After 2 days in culture, organotypic slices were transferred to an artificial cerebrospinal fluid (ACSF) containing (in mM): 130 NaCl, 2.5 KCl, 2.2 CaCl₂, 1.5 MgCl₂, 10 D-glucose, 10 HEPES (pH 7.35, osmolarity adjusted to 300 mOsm). CA1 pyramidal cells were then processed for single cell electroporation using glass micropipets containing 20 ng.µL⁻¹ plasmids encoding CRISPR/Cas9 + gRNA to MDGA1, MDGA2, or CTRL containing a nuclear BFP reporter + 6 ng.µL⁻¹ plasmid encoding tdTomato. Micropipets were pulled from 1 mm borosilicate capillaries (Harvard Apparatus) with a vertical puller (Narishige). Electroporation was performed by applying 4 square pulses of negative voltage (–2.5 V, 25 ms duration) at 1 Hz, then the pipet was gently removed. 10–20 neurons were electroporated per slice, and slices were placed back in the incubator for 7 days before electrophysiology. We checked by confocal microscopy performed on sister slices that tdTomato-positive neurons also expressed the nuclear BFP reporter of CRISPR constructs.

## Electrophysiology

Electrophysiological recordings were carried out at RT using an upright microscope (Nikon Eclipse FN1) equipped with a motorized 2D stage and micromanipulators (Scientifica), and amplifiers driven by software (Axon Instruments). Whole-cell patch-clamp was performed using micropipettes pulled from borosilicate glass capillaries (Clark Electromedical) using a micropipette puller (Narishige). Pipettes had a resistance in the range of 4–6 MΩ. The recording chamber was continuously perfused

with aCSF containing (in mM): 130 NaCl, 2.5 KCl, 2.2 $CaCl_2$, 1.5 $MgCl_2$, 10 D-glucose, 10 HEPES, and 0.02 bicuculline (pH 7.35, osmolarity adjusted to 300 mOsm), while the internal solution contained (in mM): 135 Cs-MeSO$_4$, 8 CsCl, 10 HEPES, 0.3 EGTA, 4 MgATP, 0.3 NaGTP, and 5 QX-314. Salts were purchased from Sigma-Aldrich and drugs from Tocris.

For the measurement of miniature excitatory and inhibitory currents, primary hippocampal neurons electroporated with CRIPSR/Cas9 and either CTRL, MDGA1, or MDGA2 gRNAs (all containing a GFP volume reporter) were cultured on 18 mm coverslips. GFP-positive neurons were voltage-clamped at a membrane potential of −70 mV to record AMPAR-mediated mEPSCs for 5 min, then clamped at +10 mV to record GABA$_A$ receptor-mediated IPSCs for another 5 min, in the presence of 0.5 µM tetrodotoxin to block action potentials. We verified that CNQX (20 µM) and bicuculline (20 µM) abolished the recorded mEPSCs and mIPSCs, respectively.

For the measurement of evoked AMPAR- and NMDAR-mediated currents in organotypic slices, CA1 pyramidal neurons were imaged with DIC and electroporated neurons were identified from the tdTomato fluorescence. The recording chamber was continuously perfused with ACSF bubbled with 95% $O_2$ / 5% $CO_2$ containing (in mM): 125 NaCl, 2.5 KCl, 26 NaHCO$_3$, 1.25 NaH$_2$PO$_4$, 2 $CaCl_2$, 1 $MgCl_2$, and 25 glucose. 20 µM bicuculline and 100 nM DNQX were added to block inhibitory synaptic transmission and reduce epileptiform activity, respectively. The series resistance Rs was left uncompensated, and recordings with Rs higher than 30 MΩ were discarded. EPSCs were evoked in an electroporated neuron and a nearby non-electroporated neuron (non-electro) every 10 s for 5 min using a bipolar electrode in borosilicate theta glass filled with ACSF and placed in the stratum radiatum. The stimulating electrode was linked to a generator (ISO-STIM 01D, NPI) and voltage pulses of 10–100 V and duration 100–500 µs were imposed. Voltage-clamp recordings were digitized using the Multiclamp 700B amplifier (Axon Instruments) and acquired using the Clampex software (Axon Instruments). AMPAR-mediated currents were recorded at −70 mV and NMDAR-mediated currents were recorded at + 40 mV and measured 50 ms after the stimulus, when AMPAR-mediated EPSCs are back to baseline. EPSCs amplitude measurements were performed using Clampfit (Axon Instruments).

## Patch-Seq analysis of neurons expressing CRISPR-Cas9 against MDGAs

To investigate the efficiency and potential off-target effects of the CRISPR-Cas9 strategy against MDGA1 and MDGA2, we used the Patch-seq technique (*Cadwell et al., 2016*; *Fuzik et al., 2016*). Briefly, CRISPR-Cas9 positive neurons were selected based on the expression of the GFP reporter and cells were patched using the following RNA-preserving internal solution, which contained (in mM): 120 CsMeSO$_4$, 10 BAPTA, 3 TEA-Cl, 2 Na$_2$-ATP, 2 Mg-ATP, 0.2 Na-GTP, 10 Na$_2$-Phosphocreatin (Calbiochem 2380), 320 U/mL Ribolock, 290 mOsm, pH = 7.34. The cytoplasm and the nucleus were harvested using gentle negative pressure and expelled in a lysis solution containing RNase inhibitor (Ribolock, ThermoFisher). Reverse transcription into double stranded cDNA was performed using template switching oligo (LNA-TSO) with sequence 5'-AAGCAGTGGTATCAACGCAGAGTACrGrG + G-3', in which rG indicates riboguanosines and + G indicates a locked nucleic acid (LNA)-modified guanosine, Eurogentec. The quality of these cDNA libraries was first tested using capillary-gel electrophoresis system (Labchip GX Touch, Perkin Elmer), then sequencing libraries were synthesized using NEXTFLEX Rapid 2.0 (Perkin Elmer). Sequencing was performed on a NextSeq 2000 system (Illumina) with an average of 20 millions of reads per sample. Reads were cleaned using the software cutadapt (version 1.18) (*Martin, 2011*) and fastp (version 0.20.0) (*Chen et al., 2018*) then aligned on the rat genome (Rnor_6.0) and quantified using STAR (version 2.7.1 a) (*Dobin et al., 2013*). Finally, differential gene analysis was performed with the R package DESeq2 (*Love et al., 2014*).

## Statistics

Statistical values are given as mean ± s.e.m., unless otherwise stated. Statistical significance was calculated using GraphPad Prism 8.0 (San Diego, CA). For most experiments, data did not pass the D'Agostino and Pearson tests for normality, so comparisons were made using the non-parametric Mann–Whitney test. For data sets containing more than two conditions, comparisons were made by one-way analysis of variance (ANOVA) with the Kruskal-Wallis test for non-parametric samples, followed by a post hoc multiple comparison Dunn's test. The number of experiments performed and the number of cells examined are indicated in each figure.

## Acknowledgements

We thank AM Craig (University of British Columbia, Vancouver), M Missler (Munster University), P Scheiffele (Biozentrum, Basel), and A Ting (Stanford University, Palo Alto) for the generous gift of plasmids, T Yamamoto (Kagawa University, Japan) for providing *Mdga1* KO mice lines, E Gouaux (OSHU, Vollum Institute, Portland) for the gift of anti-GluA2 antibody, S Benquet and M Munier in the team for molecular biology, R Sterling and J Girard for logistics, V Rouglan for RNA sequencing at the Transcriptomics facility (Neurocentre Magendie INSERM), the Cell Biology Core facility of the Institute (C Breillat, N Retailleau, E Verdier, N Chevrier), C Lemoigne for probe production, JB Sibarita, A Kechkar and C Butler (IINS) for the generous gift of the PALM-Tracer and WAVE-Tracer single molecule detection programs, M Mondin and C Poujol (Bordeaux Imaging Center) for providing image analysis macros, AM Craig, J Elegheert (IINS) and N Brose (Max Planck Institute, Goettingen) for scientific discussions. Confocal microscopy was done at the Bordeaux Imaging Center, part of the FranceBioImaging national infrastructure (ANR-10-INBS-04–0). The protein quantitation and Western blot analysis were done in the Biochemistry and Biophysics Core Facility of the Bordeaux Neurocampus (BioProt, funded by the Labex BRAIN ANR-10-LABX-43), with the help of Y Rufin.

This work received funding from the Fondation pour la Recherche Médicale ("Equipe FRM" DEQ20160334916), French Ministry of Research, Agence Nationale de la Recherche (grants « SynSpe » ANR-13-PDOC-0012–01, and « SyntheSyn » ANR-17-CE16-0028-01), ERA-NET Neuron "Synpathy" (ANR-15-NEUR-0007–04), Investissements d'Avenir Labex BRAIN (« SynOptoGenesis » ANR-10-LABX-43), ERC grant DynSynMem (787340), and grants from the Conseil Régional de Nouvelle Aquitaine.

## Additional information

### Funding

| Funder | Grant reference number | Author |
|---|---|---|
| Agence Nationale de la Recherche | ANR-13-PDOC-0012-01 | Mathieu Letellier |
| Agence Nationale de la Recherche | ANR-17-CE16-0028-01 | Olivier Thoumine |
| Agence Nationale de la Recherche | ANR-15-NEUR-0007-04 | Daniel Choquet |
| Agence Nationale de la Recherche | ANR-10-LABX-43 | Daniel Choquet |
| Fondation pour la Recherche Médicale | DEQ20160334916 | Olivier Thoumine |

The funders had no role in study design, data collection and interpretation, or the decision to submit the work for publication.

### Author contributions

Andrea Toledo, Mathieu Letellier, Conceptualization, Data curation, Formal analysis, Investigation, Methodology, Writing – review and editing; Giorgia Bimbi, Data curation, Formal analysis, Investigation, Methodology, Writing – review and editing; Béatrice Tessier, Data curation, Formal analysis, Investigation, Methodology; Sophie Daburon, Joris de Wit, Conceptualization, Methodology, Writing – review and editing; Alexandre Favereaux, Conceptualization, Data curation, Formal analysis, Methodology, Writing – review and editing; Ingrid Chamma, Jeroen Vanderlinden, Data curation, Methodology; Kristel Vennekens, Methodology; Matthieu Sainlos, Conceptualization, Methodology, Resources, Writing – review and editing; Daniel Choquet, Conceptualization, Project administration, Resources, Supervision, Writing – review and editing; Olivier Thoumine, Conceptualization, Methodology, Project administration, Resources, Supervision, Methodology, Writing – review and editing

**Author ORCIDs**
Mathieu Letellier (iD) http://orcid.org/0000-0003-4008-298X
Matthieu Sainlos (iD) http://orcid.org/0000-0001-5465-5641
Olivier Thoumine (iD) http://orcid.org/0000-0002-8041-1349

**Decision letter and Author response**
Decision letter https://doi.org/10.7554/eLife.75233.sa1
Author response https://doi.org/10.7554/eLife.75233.sa2

## Additional files

### Supplementary files
• Supplementary file 1. Statistics table. The number of experimental replicates ( = number of independent experiments performed), number of biological replicates ( = number of different cells analyzed), result of the normality test, statistical tests used to compare the data, and P-values associated to each test are given for each figure panel.

• Supplementary file 2. Cell line authentication. Labels of the vials of COS-7 and HEK-293T cell lines from ECACC purchased through Sigma-Aldrich in 2015 and 2016, respectively.

• Transparent reporting form

### Data availability
All data generated or analysed during this study are included in the manuscript and supporting file; source data files are provided.

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
