## [Editor Report]

The authors used immunostaining and single-molecule tracking analyses in cultured hippocampal neurons to address some unresolved issues on MDGA molecules that are regarded as negative regulators of synapse development. MDGAs were highly mobile and homogenously distributed in cultured neurons with localization and dynamics of NLGN1 and GluA2 altered upon loss of MDGA2.

---

## [Decision Letter]

**Decision letter after peer review:**

[Editors’ note: the authors submitted for reconsideration following the decision after peer review. What follows is the decision letter after the first round of review.]

Thank you for submitting your work entitled "MDGAs are fast-diffusing molecules that delay excitatory synapse development by altering neuroligin behavior" for consideration by *eLife*. Your article has been reviewed by 3 peer reviewers, one of whom is a member of our Board of Reviewing Editors, and the evaluation has been overseen by a Senior Editor. The reviewers have opted to remain anonymous.

Comments to the Authors:

We are sorry to say that, after consultation with the reviewers, we have decided that your work will not be considered further for publication by *eLife*. Although all three reviewers agreed on the quality and impact of the datasets, they raised substantial concerns about the limitations in key control experiments and validation of the approaches taken. However, if the authors choose to address all of the review comments and resubmit the manuscript, the manuscript could be re-considered for review.

*Reviewer #1:*

This study reports the roles of MDGAs in the regulation of surface dynamics and synaptic localization of NLGN1 and GluA1 in cultured neurons. The results suggest that endogenous MDGAs are largely non-synaptic and strongly diffusable and that MDGAs negatively regulate surface dynamics of NLGN1 and GluA1 and excitatory synapse development. The authors use multiple high-end imaging approaches such as single-molecule dynamics (uPAINT) and nano-scale synapse imaging (dSTORM) to support the main conclusions, which are largely convincing. The authors succeeded in demonstrating, by live imaging experiments in cultured neurons, that how MDGAs negatively regulate excitatory synapse organization mediated by NLGN1 and GluA1. Given the increasing importance of molecular and cellular synapse organization, negative regulators of excitatory synapse development, and related brain disorders such as autism, this study is an important step forward in synapse-related neuroscience fields.

1. The authors mainly use knockdown and knockout approaches to draw the main conclusions, including the negative influences of MDGAs on excitatory synaptic trafficking and retention of NLGN1 and GluA1. However, the manuscript is lacking data from MDGA overexpression, which could substantially increase the strength of the study.

2. This study uses several rescue MDGA constructs that resistant to the knockdown/knockout. But it could be argued that the rescue effects could still be indirect. Given that the crystal structure of MDGA-NLGN complexes is known, if some of the rescue constructs with mutations disrupting the MDGA-NLGN interaction fail to rescue diffusion/synaptic localization of NLGN1/GluA1, it could be strong support for the main conclusions.

*Reviewer #2:*

This manuscript by Toledo, Thoumine and colleagues employed immunostaining and single molecule tracking analyses in cultured hippocampal neurons to address some previously unresolved issues on MDGA molecules that are regarded as negative regulators of synapse development. The authors also employed both knockdown and CRISPR/CAS9-based knockout mice for MDGA1 and MDGA2. They found that MDGAs are highly mobile and homogenously distributed in cultured neurons. They further showed that localization and dynamics of NLGN1 and GluA2 are altered upon loss of MDGA2. Overall, the data appear to be okay, but I am not persuaded that of the conceptual advance given the limitation of major approaches employed.

1. All the experiments were performed in dissociated cultured hippocampal neurons. I do not see that major questions the authors tried to address in the current study could be convincingly tackled in this rather limited system.

2. The authors used AP (biotin acceptor peptide)-conjugated MDGAs, instead of MDGA2 antibodies that are not suitable for detecting endogenous proteins. However, this approach is essentially similar to that used by Ting and colleagues (Loh et al., 2016, Cell) that obviously contrasts with conclusion of Connors et al. (2016 Neuron). Without further rigorous validation of recombinant MDGA1 and MDGA2, concern about overexpression artifact cannot be eliminated.

3. The authors did not provide evidence to eliminate potential off-target effects from CRISPR/CAS9-based deletion of MDGA proteins, apart from showing knockout efficiency. Thus, conclusion that deletion of both MDGA1 and MDGA2 alters excitatory postsynapse maturation cannot be justified. These results obtained from dissociated cultured neurons also contrast with Connors et al. 2017 Cell Reports, showing that MDGA1 specifically acts as a negative regulator at GABAergic hippocampal CA1 synapses in vivo.

*Reviewer #3:*

This work by Toledo and colleagues characterizes the synaptic localization of MDGA1/2 and their effects on Neuroligin 1 and AMPA receptor surface dynamics during development in dissociated neuronal cultures. MDGA1/2 are analyzed using KO, knockdown with molecular replacement, and CRISPR targeting. The data support that endogenous MDGA1 is partially synaptic but that MDGA1/2 are not enriched at excitatory synapses and become less abundant at excitatory postsynaptic sites as neurons mature. In agreement, they diffuse very fast in dendritic membranes. Detection of Homer density and tracking of Nlgn1 and GluA2 after MDGA2 knockdown support that both MDGAs (tested by CRISPR) restrict in young neurons synapse number and that MDGA2 loss reduces Nlgn1 and AMPA receptor surface mobility in more mature neurons. A role of MDGA2 in restricting Nlgn1 Tyr phosphorylation is shown as well. Functional effects on synapse maturation are measured by electrophysiological recordings in the cultures. In agreement with the Homer data representing an increase in functional synapse number, CRISPR targeting of MDGA1 or MDGA2 increased the frequency of mEPSCs. No effect on mEPSC amplitudes was found.

Maturation-dependent changes in the localization and dynamics of MDGAs, Nlgn1, and AMPA receptors are well characterized, including by strong super-resolution imaging. The model that MDGAs serve as restrictive factors in synapse development through altering Nlgn1 mobility is interesting to gain new insights into the negative control of synapse assembly.

Several points need to be addressed to further support the conclusions.

1. The data support that MDGAs restrict the formation of excitatory synapses. The authors propose that this due to an effect on Neurexin/Neuroligin complex stability, as expected from the literature. Can the authors show that postsynaptic manipulation of MDGAs indeed alters the mobility of Neurexins and the extent of their presynaptic retention in maturing neurons?

2. Why does targeting MDGA1 or 2 not alter synaptic strength measured as mEPSC amplitudes? Their role in controlling AMPA receptor mobility should predict that.

[Editors’ note: further revisions were suggested prior to acceptance, as described below.]

Thank you for submitting your article "MDGAs are fast-diffusing molecules that delay excitatory synapse development by altering neuroligin behavior" for consideration by *eLife*. Your article has been reviewed by 3 peer reviewers, and the evaluation has been overseen by Gary Westbrook as the Senior Editor. The reviewers have opted to remain anonymous.

The reviewer are overall satisfied that the manuscript will eventually be publishable in *eLife*, providing that you can address the remaining concerns. We expect that you will be able to address these concerns with existing data or with careful revisions of the text. Depending on your response, we will decide whether to handle the revisions editorially or request re-review.

Essential revisions:

1. Rescue experiments for MDGA2 knockdown: there are cases where rescue effects were not clear (e.g. Figure 5F). Statistics should be performed between shMDGA2 and shMDGA2+res to ensure significance in this comparison.

2. Neuron morphologies: the authors argue that the cultures were relatively of early stages for some images. However, it is difficult to discern clear dendritic spine structures in many fluorescent images obtained from DIV14 cultures (e.g., Figure 5—figure supplement 1). This is still a lingering problem.

3. Validation of CRISPR knockout approach: the authors nicely provided with convincing in silico evidences, arguing against potential no off-target effects. However, the authors need to provide data (e.g. qPCRs), showing no changes in expression levels of some neuronal genes from cultured neurons or N2a cells expressing sgMDGA1 or sgMDGA2.

4. Interpretation of Figure 6—figure supplement 1: the reasonable conclusion seems to be that MDGA2 is NOT required for Nlgn1/Nrxn1beta contact. Please discuss.

5. Discussion: The authors' argument is not convincing about adhesion-dependent competition between neurons with citing Kwon et al. 2012. Indeed, Sudhof and colleagues later demonstrated in cultured neurons that sparsely knocking out Nlgn1 did not produce any phenotypes in synapse numbers (Chandra et al., 2017 J Neurosci). The authors need to consider revising the Discussion section to reconcile the discrepancy with previous observations.

6. The authors could discuss in the manuscript the new data in Figure 6 – —figure supplement 1. While they do not show an effect of MDGA knockdown on Nrx1beta, future studies can test whether the Nrx overexpression approach obscures effects of MDGA knockdown, or whether MDGA knockdown preferentially affects Nrx1alpha. The added discussion points provide interesting ideas into developmental mechanisms controlling MDGA functions.

---

## [Author Response]

[Editors’ note: the authors resubmitted a revised version of the paper for consideration. What follows is the authors’ response to the first round of review.]

Reviewer #1:This study reports the roles of MDGAs in the regulation of surface dynamics and synaptic localization of NLGN1 and GluA1 in cultured neurons. The results suggest that endogenous MDGAs are largely non-synaptic and strongly diffusable and that MDGAs negatively regulate surface dynamics of NLGN1 and GluA1 and excitatory synapse development. The authors use multiple high-end imaging approaches such as single-molecule dynamics (uPAINT) and nano-scale synapse imaging (dSTORM) to support the main conclusions, which are largely convincing. The authors succeeded in demonstrating, by live imaging experiments in cultured neurons, that how MDGAs negatively regulate excitatory synapse organization mediated by NLGN1 and GluA1. Given the increasing importance of molecular and cellular synapse organization, negative regulators of excitatory synapse development, and related brain disorders such as autism, this study is an important step forward in synapse-related neuroscience fields.

We thank this reviewer for those very positive comments about our study.

1. The authors mainly use knockdown and knockout approaches to draw the main conclusions, including the negative influences of MDGAs on excitatory synaptic trafficking and retention of NLGN1 and GluA1. However, the manuscript is lacking data from MDGA overexpression, which could substantially increase the strength of the study.

We initially did not consider over-expressing MDGA1 or MDGA2 in cultured neurons since this has extensively been done in several previous reports. Indeed, *Pettem et al. (J. Cell Biol, 2013)* found that over-expressing HA-MDGA1 caused a 40% decrease in VGAT and gephyrin puncta in DIV 14 hippocampal neurons, without affecting PSD-95/VGLUT1 puncta. In agreement with those results, *Kim et al.* (PNAS, 2013) found by electrophysiology that MDGA1 overexpression decreased by 40% the density of VGAT puncta (but not VGLUT1 puncta) and mIPSC (but not mEPSC) frequency in cultured cortical neurons. In contrast, *Loh et al. (Cell, 2016)* reported that over-expressing Venus-MDGA1 did not induce any change in either excitatory or inhibitory synapse density, while over-expressing Venus-MDGA2 surprisingly increased by 50% inhibitory synapse density and did not affect excitatory synapse density. The cause of the discrepancy between those studies is unclear, and might come from the fact that the N-terminal Venus tag (25 kD) used by *Loh et al.* is quite large compared to MDGA itself (130 kD) and might interfere with NLGN binding, thereby affecting the localization and/or function of the recombinant MDGAs expressed in neurons.

Nevertheless, in initial exploratory experiments we transfected DIV 7 hippocampal cultures with V5-MDGA1 or V5-MDGA2 using calcium phosphate, to explore the effect of MDGA overexpression on AP-NLGN1 localization and mobility (Author response image 1). We hypothesized that overexpressing MDGAs would compete with NLGN1 binding to NRXN and thus lead to a lower NLGN1 synaptic enrichment and a corresponding increase in NLGN1 diffusion at the membrane. However, opposite to our predictions, MDGA1 or MDGA2 overexpression both led to a reduction in NLGN1 diffusion, with no apparent increase of NLGN1 confinement in synapses. One explanation for these counter-intuitive effects might be that NLGN1 and MDGAs form extrasynaptic aggregates in the membrane due to the strong overexpression of MDGAs achieved by calcium phosphate transfection, which could result in a decrease of NLG1 mobility. Whichever the mechanism, the difficulty to interpret those results led us for the rest of the study to stay away from over-expression approaches, focusing instead on the dynamics of MDGAs using replacement strategies, and designing new CRISPR/Cas9 constructs to achieve full single cell KO of MDGAs compared to partial KD induced by previously characterized shRNAs.

**Author response image 1. sa2fig1:** AP-NLGN1 mobility upon MDGA1 and MDGA2 overexpression. Neurons were transfected using calcium phosphate with AP-NLGN1, Homer1c-GFP, BirAER, and V5-MDGA1 or V5-MDGA2 at DIV 7 and imaged at DIV 14. Different DNA ratios of AP-NLGN1 to MDGA were used (1:1; 1:2; 1:3). (A) APNLGN1 sparsely labelled with mSA conjugated to STAR 635P for single molecule tracking. (B,C) Semi-log plots of the distribution of AP-NLGN1 diffusion coefficients. DNA ratios are indicated in the inset of each plot. Curves represent the mean ± SEM of at least 4 neurons per condition from two independent experiments. (D) Bar plot of median diffusion coefficients averaged per cell in the different conditions. (E) Plot of the mobile and immobile fractions of AP-NLGN1 as a function of MDGA overexpression. The threshold between mobile and immobile molecules was set at D = 0.01 µm²/s.

2. This study uses several rescue MDGA constructs that resistant to the knockdown/knockout. But it could be argued that the rescue effects could still be indirect. Given that the crystal structure of MDGA-NLGN complexes is known, if some of the rescue constructs with mutations disrupting the MDGA-NLGN interaction fail to rescue diffusion/synaptic localization of NLGN1/GluA1, it could be strong support for the main conclusions.

We appreciate this very interesting suggestion of the reviewer. Indeed, two recent papers that described the crystal structure of the MDGA1-NLGN2 complex *(Kim et al., Neuron 2017; Gangwar et al., Neuron 2017)* have reported a series of critical mutations in MDGA1 that abolish binding to NLGN2, including R105A/D/E, Y107A, and R123D in the Ig1 domain, and F154A, R156A, Y187A, and K200E in the Ig2 domain. Based on these data, and focusing on MDGA2 which shows the most robust effects in our assays, we designed an MDGA2 construct carrying mutations R107E and Y109E in Ig1, and F156E and Y189A in Ig2, knowing that there is a 2 amino-acid difference between MDGA1 and MDGA2 sequences. Considering the remote positions of the mutations, we adopted a molecular strategy based on outsourced gene synthesis, but unfortunately, we experienced a significant delay in the reception of this mutant gene (> 2 months). Given the amount of work required for the characterization of this new MDGA2 mutant (in particular regarding its binding capacity to NLGN1) before running the actual experiments on NLGN1 and AMPAR diffusion, and considering that the post-doctoral fellow who is first author on the paper was reaching the end of her contract in the lab, we decided to stop the generation of this construct. Therefore, we are sorry will not be able to provide these data for the revisions of this paper.

Instead, we had initiated uPAINT and STORM experiments with a NLGN1 mutant which was reported not to bind MDGA1 in SPR experiments *(Elegheert et al., Neuron 2017)*. This mutant called NLGN1-ΔsiteII carries 5 mutations in the extracellular domain (D429A, F430A, S433A, N434A and R450A). Given the common sequence similarity between MDGA1 and MDGA2 and similar binding mechanisms to NLGN1 and NLGN2, we reasoned that these mutations in NLGN1 would impair binding to both MDGA1 and MDGA2. We were expecting this NLGN1-ΔsiteII mutant to show lower mobility and stronger synaptic enrichment than NLGN1 WT. However, even though the data show a trend towards higher NLGN1 synaptic enrichment in neurons expressing NLGN1ΔsiteII compared to NLGN1-WT (in replacement conditions), these differences turned out not be significant (Author response image 2). One possible explanation can be that mutant NLGN1-ΔsiteII forms heterodimers with either a fraction of wild type NLGN1 which is not totally silenced by the shRNA, or with endogenous NLGN3 *(Shipman et al., Neuron 2012; Bemben et al., Trends in Neurosci 2015)* that could still interact with MDGAs, thereby attenuating the effect of the mutation. For these reasons, we did not include those data in the manuscript, but present them here for the reviewer’s perusal.

**Author response image 2. sa2fig2:** AP-NLGN1 ΔsiteII mobility and localization. Hippocampal neurons were electroporated at DIV 0 with a combination of shRNA to NLGN1 (GFP or EBFP2 reporter), rescue AP-NLGN1 WT or rescue AP-NLGN1 ΔsiteII, biotin ligase BirAER, and Homer1c-DsRed or Xph20-EGFP as post-synaptic merkers. (A-F) uPAINT experiments were performed at DIV 10 and 14, after labelling neurons expressing AP-NLGN1 WT or ΔsiteII with 1 nM STAR 635P-conjugated mSA. (A, B) Representative images of dendritic segments showing the shNLGN1 reporter GFP signal, Homer1c-DsRed and the corresponding single molecule detections (magenta) and trajectories (random colors) acquired during an 80 s stream, for the indicated time in culture. (C, E) Corresponding semi-log plots of the distributions of diffusion coefficients for AP-NLGN1 WT or ΔsiteII at DIV 10 and 14 respectively. (D, F) Graph of the median diffusion coefficient, averaged per cell, in the different conditions. Data represent mean ± SEM of n > 14 neurons for each experimental condition from at least three independent experiments, and were compared by a Mann-Whitney test (ns: not significant). (G-J) dSTORM experiments were performed at DIV 10 or 14, after labelling neurons with 100 nM Alexa 647-conjugated mSA. (G, I) Representative images of dendritic segments showing the shNLGN1 reporter EBFP2, Xph20-EGFP positive synapses, the super-resolved localization map of all APNLGN1 (WT or ΔsiteII) single molecule detections (gold), and merged images (Xph20-EGFP in red and detections in green). Scale bars, 20 µm. (H, J) Bar plots representing the enrichment of AP-NLGN1 WT or AP-NLGN1 ΔsiteII localizations at synapses. Values were obtained from at least three independent experiments and n > 6 neurons for each experimental condition. Data were compared by a Mann-Whitney test (ns: not significant).

Reviewer #2:This manuscript by Toledo, Thoumine and colleagues employed immunostaining and single molecule tracking analyses in cultured hippocampal neurons to address some previously unresolved issues on MDGA molecules that are regarded as negative regulators of synapse development. The authors also employed both knockdown and CRISPR/CAS9-based knockout mice for MDGA1 and MDGA2. They found that MDGAs are highly mobile and homogenously distributed in cultured neurons. They further showed that localization and dynamics of NLGN1 and GluA2 are altered upon loss of MDGA2. Overall, the data appear to be okay, but I am not persuaded that of the conceptual advance given the limitation of major approaches employed.

We are sorry to hear that, in contrast to reviewers 1 and 3, this second reviewer did not appreciate more our experimental approaches. Nevertheless, we believe that the data we provide on MDGA diffusion and nanoscale localization using high-end single molecule detection of MDGAs expressed at near-endogenous levels are highly original with respect to the existing literature. Finally, the design and characterization of new CRISPR/Cas9 constructs offered us the possibility to achieve single-cell KO of MDGAs over a specific developmental window, and thereby reveal the role played by both MDGA1 and MDGA2 in excitatory synapse differentiation, by regulating NLGN1 signaling and AMPAR dynamics. We discuss below the specific points raised by the reviewer, in particular the discrepancy between our results and previous in vivo studies, which we partly addressed by performing new electrophysiology experiments in organotypic hippocampal slices.

1. All the experiments were performed in dissociated cultured hippocampal neurons. I do not see that major questions the authors tried to address in the current study could be convincingly tackled in this rather limited system.

First, the original questions we raised regarding the nanoscale localization and surface mobility of MDGAs, NLGN1, and AMPARs can only be addressed in dissociated neurons due to technical limitations, in particular the use of oblique illumination that allows us to detect single molecules at the cell membrane (uPAINT and dSTORM paradigms). At present, these types of measurements in more complex 3D tissues are technically extremely challenging. On the other hand, in vivo studies where proteins are knocked-out throughout mouse development are prone to compensatory effects and lack the temporal resolution accessible in culture systems where subtle roles of those molecules can be identified. For example, primary neuronal cultures allowed us to reveal that MDGAs operate in a narrow developmental window, between DIV 10 and 14, where they delay excitatory synaptic differentiation. Another aspect of full KO studies is that MDGAs are silenced in all cell types, including neurons and astrocytes. However, recent reports indicate that NLGNs are expressed not only in neurons but also in astrocytes where they play an important role in astrocyte morphogenesis and synapse development *(Stogsdill et al., Nature 2017)*. Interestingly, we found by qRT-PCR that both MDGA1 and MDGA2 are expressed in astrocytic cultures at the mRNA level (Figure 1 —figure supplement 1A,B). Thus, the KO of MDGAs in astrocytes, by affecting selective NRXN-NLGN interactions at tripartite synapses, might differentially regulate excitatory and inhibitory synaptic transmission in vivo. This might be one reason why MDGA1 KO in vivo affects essentially inhibitory synaptic transmission *(Connor et al., Cell Rep 2017)*. In our Banker cultures, MDGAs were selectively silenced in neurons since astrocytes do not survive or proliferate on the coverslips where neurons are plated after electroporation.

Finally, global vs sparse KD of proteins have shown different effects in diverse systems. Notably, full KO of NLGN1 in hippocampus was shown not to alter synaptic density *(Varoqueaux et al., Neuron 2006),* while KD of NLGN1 in individual primary neurons causes a strong reduction in synapse density *(Chih et al., Science 2005)*. Sparse silencing of NLGN1 in the hippocampus further revealed that NLGN1-dependent competition between neurons regulates synaptogenesis *(Kwon et al., Nat Neurosci. 2012)*. In a similar way, we can speculate that when MDGA1 is lost in all cells of the hippocampus *(Connor et al., Cell Reports 2017)*, the effects observed mostly on inhibitory synaptic transmission would be different from those observed when MDGA1 levels are reduced only in isolated cells (while remaining constant in surrounding neighbors), revealing instead a role in excitatory synapse development *(this study)*. Strikingly, the phenotypes we saw with single cell MDGA1/2 KO (e.g. increases in post-synapse numbers and AMPAR-mediated mEPSC frequency) resembles the ones we previously observed by over-expressing NLGN1, thereby reinforcing the concept that neurons compete for synapse formation through NRXN-NLGN adhesion.

Between these two extreme systems (in vivo and in vitro), organotypic slice cultures are an ex vivo preparation which offer the possibility to manipulate protein expression level in a relatively restricted time frame, while preserving synaptic connectivity. To address the reviewer’s concern, we thus decided to down-regulate MDGAs in organotypic hippocampal slices. We used single cell electroporation to deliver CRISPR/Cas9 constructs against MDGA1 or MDGA2 into CA1 pyramidal neurons, and studied the effects on excitatory synaptic transmission using dual-patch clamp electrophysiological recordings. Specifically, we measured the amplitude of both AMPA and NMDA-receptor mediated EPSCs in CA1 neurons, upon stimulation of Schaffer’s collaterals, normalized to values measured simultaneously in neighboring non-electroporated neurons. Strikingly, both gRNAs to MDGA1 and MDGA2 increased AMPAR-mediated EPSCs without affecting NMDA receptor-mediated EPSCs, as compared to non-electroporated controls (Figure 9B-D and Figure 9 —figure supplement 1A,B). As a result, the ratio between AMPAR and NMDAR-mediated EPSCs was significantly elevated in both MDGA1 and MDGA2 KO neurons (Figure 9E), supporting a synaptic unsilencing mechanism (Figure 10, lower panel). No significant effect of gRNA CTRL on either AMPAR- or NMDAR-mediated EPSCs, or on the AMPA/NMDA ratio was observed when compared to non-electroporated neurons (Figure 9B-E), validating the normalization procedure and the absence of off-target effects of the control gRNA. No effect of CRISPR to MDGA1 or MDGA2 on the paired pulse ratio was observed (Figure 9 —figure supplement 1C,D), ruling out pre-synaptic mechanisms. Together, those results strengthen the concept that both MDGA1 and MDGA2 down-regulate AMPAR recruitment during excitatory synapse development. These new data are presented in the Results section pp. 11-12 and discussed p. 15.

2. The authors used AP (biotin acceptor peptide)-conjugated MDGAs, instead of MDGA2 antibodies that are not suitable for detecting endogenous proteins. However, this approach is essentially similar to that used by Ting and colleagues (Loh et al., 2016, Cell) that obviously contrasts with conclusion of Connors et al. (2016 Neuron). Without further rigorous validation of recombinant MDGA1 and MDGA2, concern about overexpression artifact cannot be eliminated.

We appreciate this comment and share some of the reviewer’s concerns. *Loh et al. (Cell, 2016)* used constructs where a large HRP protein (44 kDa) was fused to the N-terminus of MDGA1 and MDGA2, allowing their subsequent biotinylation and detection with tetrameric fluorescent streptavidin. Alternatively, they used a Venus N-terminal tag, which is also quite prominent (25 kDa). These large tags compared to MDGA itself (130 kDa) can potentially hinder the binding of MDGAs to NLGN1 since the MDGA N-terminus lies very close to the binding interface *(Kim et al., Neuron 2017; Gwangar et al., Neuron 2017; Elegheert et al., Neuron 2017)*. This concern, in addition to the fact that the authors over-expressed HRP- or Venus-tagged MDGA proteins might be responsible for mislocalization artifacts. In contrast, we were careful in our study to use much smaller N-terminal tags (V5 and AP, 15 aa each), and introduced a flexible linker so as to preserve MDGA binding to NLGN1, even in the presence of labeling probes such as V5 Fab or monomeric streptavidin (mSA). We checked by streptavidin pull-down and surface cross-linking experiments with anti-biotin antibodies that biotinylated AP-tagged MDGAs were indeed able to bind NLGN1 (Figure Supplement 6).

Regarding MDGA1 expression levels, we now clearly show by MDGA1 immunostaining that shMDGA1 reduces the expression levels of endogenous MDGA1 by 75% in dissociated hippocampal neurons (Author response image 3). When rescuing the expression of native MDGA1 with shRNAresistant AP-MDGA1 in the same conditions as the ones shown in Figures 2 and 3 of the manuscript, there is only a mild overexpression (~1.5 fold) of AP-MDGA1 with respect to endogenous MDGA1, while over-expression leads to a 3-fold higher level (Author response image 4). Although we lack a good antibody against MDGA2 to show that rescue AP-MDGA2 is expressed at similar levels as native MDGA2, our electrophysiological recordings clearly demonstrate that the expression of rescue AP-MDGA2 in combination with CRISPR/Cas9 + gRNA to MDGA2 blocks the increase in AMPAR-mediated mEPSC frequency observed upon MDGA2 single-cell KO (Figure 8F,H). This indicates that the expression level and functionality of recombinant APMDGA2 are likely to match those of endogenous MDGA2. Figures IV and V are now merged into Figure 2—figure supplement 2.

**Author response image 3. sa2fig3:** Validation of shRNA MDGA1. Immunodetection of endogenous MDGA1 in non-electroporated neurons and in neurons electroporated with shCTRL or shMDGA1 at DIV 0. (A) Images show MDGA1 in magenta, the shRNA GFP reporter in green, and MAP-2 in blue. Scale bar: 20 µm. (B) Bar plot showing the anti-MDGA1 fluorescence intensity normalized to the average signal measured on non-electroporated neurons (non electro). Data represent the mean ± SEM of n > 40 cells for each condition from two independent experiments, and were compared by a Kruskall-Wallis test followed by Dunn’s multiple comparison test (ns: not significant; **** p < 0.001).

**Author response image 4. sa2fig4:** Validation of MDGA1 rescue. Immunodetection of endogenous MDGA1 in non-electroporated neurons and in neurons electroporated at DIV 0 with shCTRL or shMDGA1, plus rescue AP-MDGA1. (A) Images show MDGA1 in magenta, the shRNA GFP reporter in green, and MAP-2 in blue. Scale bar: 10 µm. (B) Bar plot showing the anti-MDGA1 fluorescence intensity normalized to the average signal measured on non-electroporated neurons (non electro). Data represent the mean ± SEM of n > 50 cells for each condition from three independent experiments, and were compared by a Kruskall-Wallis test followed by Dunn’s multiple comparison test (**p < 0.01, ****p < 0.001).

3. The authors did not provide evidence to eliminate potential off-target effects from CRISPR/CAS9-based deletion of MDGA proteins, apart from showing knockout efficiency. Thus, conclusion that deletion of both MDGA1 and MDGA2 alters excitatory postsynapse maturation cannot be justified.

Although RNA-guided genome editing via the CRISPR/Cas9 system is now widely used in biomedical research, genome-wide target specificities of Cas9 nucleases still remain controversial. Many studies have shown the CRISPR/Cas9 system to be highly specific, however others show a substantial number of off-target activity *(Anderson et al. Nat Methods 2018; Haeussler, Cell Biol Toxicol. 2020; Kim et al., Nat Methods 2015).* A search of the genome for sequence homology with the gRNA used to knock-out a certain gene with CRISPR/Cas9, might give several results if mismatches in this sequence are considered. However, even if some mismatches in the target sequence can be tolerated for the binding of the gRNA, mutations closer to the PAM sequence (protospacer adjacent motif sequence) are much less tolerated and a maximum of three mismatches can be tolerated for cleavage *(Cong et al., Science 2013)*. Interestingly, whole genome sequencing of CRISPR/Cas9 edited genomes have shown that indels are rarely detected above deep-sequencing error rates (*Kim et al., Nat Methods 2015)*. However, a more recent study of high-throughput screening showed that similar sequences to the gRNA are recognized but not cleaved in the genome, and the tolerance for sequence mismatches here rises up to 5 base pairs (*Tsai et al., Nat Biotechnol 2015*). In this sense, during the design of our CRISPR tools with the Chochop software, we took particular care to select potential gRNAs that do not share homology with any other sequence in the genome, even considering 0, 1, 2 or 3 mismatches (MM, MM2, MM3 in TABLES 1 and 2), and with high efficiency scores (based on the “G20” system) (Moreno-Mateos et al. Nat Methods 2015). Also, lower values of GC% content were chosen when possible *(Wu et al., Nat Biotechnol 2014; Haeussler, Cell Biol Toxicol. 2020).*

Detailed information regarding the target sequence for MDGA1 and MDGA2 to design gRNAs is shown in Author response tables 1 and 2. Highlighted in yellow are the sequences used in the manuscript.

**Author response table 1. sa2table1:** MDGA1 potential gRNAs [ID Pub Med: 309659; ENSRNOG00000000536].

Rank	Target sequence	Genomic location	Stra nd	GC cont ent (%)	Self-complementarity	MM0	MM1	MM2	MM3	Efficie ncy
1	TGCAGCGTCTCCAACGACGTGGG	chr20:8526603	-	60	1	0	0	0	0	70.24
2	GAGCTTCAGTGCGAAGTGCGCGG	chr20:8527460	-	60	1	0	0	0	0	66.16
3	CTTCAACGTACGAGCCCGGGAGG	chr20:8527275	-	65	0	0	0	0	0	64.49
4	CGTCCGAGGCAACTTCTACCAGG	chr20:8534783	-	60	1	0	0	0	0	61.12
5	ACACGTTACGCACAGACACCTGG	chr20:8533927	+	55	0	0	0	0	0	59.11

**Author response table 2. sa2table2:** MDGA2 potential gRNAs [ID Pub Med: 314180; ENSRNOG00000000618].

Rank	Target sequence	Genomic location	Stra nd	GC cont ent (%)	Self-complemetarity	MM0	MM1	MM2	MM3	Efficie ncy
1	ATTTAGTGTACGGTCTCGTGTGG	chr6:89291822	-	45	0	0	0	0	0	67.1
2	GTGCACAATCCGAACCGTCGGGG	chr6:88917044	+	60	0	0	0	0	0	63.54
3	AGTCAGCACACGTATCGGATAGG	chr6:88556575	+	50	1	0	0	0	0	62.59
4	TGCACAATCCGAACCGTCGGGGG	chr6:88917045	+	60	1	0	0	0	0	62.15
5	AGGGTCTATACCATCCGGGAAGG	chr6:88916979	-	55	1	0	0	0	0	62.06

We further analyzed the sequences of interest using CRISPOR (http://crispor.tefor.net/), a software that helps to design, evaluate and clone guide sequences for the CRISPR/Cas9 system. With this algorithm, we obtained different score values to assess gRNA specificity and efficiency. As shown in Author response tables 3 and 4, the gRNAs used to KO MDGA1 and MDGA2 (highlighted in yellow) exhibit high specificity according to specificity in score *(Doench et al., Nat Biotechnol. 2016)* (also shown in Author response tables 1 and 2 as Efficiency value). The selected gRNAs also showed high “out of frame” and “lindel” scores which indicate the efficiency of the gRNA to induce out of frame deletions or frame shifts, the goal of the CRISPR strategy *(Bae et al., Bioinformatics 2014; Chen et al., Nucleic Acids Res 2019)*.

**Author response table 3. sa2table3:** CRISPOR evaluation of MDGA1 gRNAs.

gene	targetSeq	MITSpecScore	CFDSpecScore	Off-targetCount	Doench '16-Score	Out-of-Frame-Score	LindelScore
MDGA1	CGCGTGAGCCGCGAAATGAGCGG	99	99	4	49	54	83
MDGA1	AGCCATTATAGCGAGCCGTCTGG	98	98	21	57	55	80
MDGA1	ATGGCTTCAACGTACGAGCCCGG	98	99	28	53	74	78
MDGA1	TGCCAGACGGCTCGCTATAATGG	97	99	22	32	59	81
MDGA1	CTTCAACGTACGAGCCCGGGAGG	97	98	17	64	70	78

**Author response table 4. sa2table4:** CRISPOR evaluation of MDGA2 gRNAs.

Gene	targetSeq	MITSpecScore	CFDSpecScore	Off-targetCount	Doench '16-Score	Out-of-Frame-Score	LindelScore
MDGA2	TGTACGCTCCCCCGACGGTTCGG	100	100	4	41	67	77
MDGA2	TGCACAATCCGAACCGTCGGGGG	99	99	10	62	69	84
MDGA2	GTGCACAATCCGAACCGTCGGGG	98	98	14	64	72	73
MDGA2	GAGTGCACAATCCGAACCGTCGG	98	98	18	61	63	79
MDGA2	AGTGCACAATCCGAACCGTCGGG	97	98	21	48	68	79
MDGA2	ATTTAGTGTACGGTCTCGTGTGG	96	98	25	67	43	77

We also explore the amount and nature of possible off-target sequences by increasing the number of tolerated mismatches (MM) in the target sequence. When considering higher number of MM by using the CRISPR OFF finder tool (http://www.rgenome.net/cas-offinder/), a number of off-targets appear for both gRNAs (Author response table 5).

**Author response table 5. sa2table5:** CRISPR OFF TARGET analysis with 4 and 5 mismatch values.

	Target Sequence	Mismatch	Number of Found Targets
**MDGA1**	CTTCAACGTACGAGCCCGGGNGG	0	1
**MDGA1**	CTTCAACGTACGAGCCCGGGNGG	4	17
**MDGA1**	CTTCAACGTACGAGCCCGGGNGG	5	209
**MDGA2**	ATTTAGTGTACGGTCTCGTGNGG	0	1
**MDGA2**	ATTTAGTGTACGGTCTCGTGNGG	4	23
**MDGA2**	ATTTAGTGTACGGTCTCGTGNGG	5	321

A more detailed analysis of the off-target candidates for both genes shows that most of them are either intergenic or intron sequences (except for Pex1exon in gRNA MDGA2), and they exhibit very low CDF scores, pointing at low cutting frequency probability (Author response tables 6 and 7).

**Author response table 6. sa2table6:** Author response table.

Name off target sequence for MDGA1 gRNA	CFD Score
intergenic_Vangl2	0,36
intron_Tbxas1	0,29
intergenic_SCCPDH|TFB2M	0,21
intergenic_Scg2|ENSRNOG00000037663	0,18
intergenic_ENSRNOG00000037633|Nyap2	0,17
intron_Galnt18_1	0,16
intergenic_Ttc7b|Rps6ka5	0,16
intergenic_Six2|Srbd1	0,12
intron_Slc9a9_8	0,11
intergenic_Adora1|Myog_13	0,1
intergenic_Clstn2|Nmnat3_8	0,09
intergenic_ENSRNOG00000047256|ENSRNOG00000031960	0,07
intron_Wasf2_5	0,07
intergenic_ENSRNOG00000031681|ENSRNOG00000024649/ENSRNOG00000036386	0,06
intergenic_Ffar2|Ffar3	0,05
intron_Apbb2	0,05
intron_Fshr_6	0,02

**Author response table 7. sa2table7:** Author response table.

Name off target sequence for MDGA2 gRNA	CFD Score
exon_Pex1_chr4	0,44
intergenic_AABR07041418.1|RGD1563975_chrX	0,28
intron_Ssbp2_chr2	0,25
intergenic_Prune2|LOC102546963_chr1	0,2
intergenic_Crygs|AABR07034636.1_chr11	0,17
intron_Ift80_chr2	0,15
intergenic_LOC685114|Ccdc178_chr18	0,15
intergenic_U6|AABR07048636.1_chr5	0,14
intron_Tmem47_chrX	0,13
intergenic_AABR07001025.1|AABR07001035.1_chr1	0,1
intergenic_AC241705.1|Gap_chrY_KL568150v1_random	0,09
intron_Camk2b_chr14	0,07
intergenic_Samd4a|Gch1_chr15	0,07
intergenic_AABR07028381.1|Klf6_chr17	0,06
intron_Map4_chr8	0,05
intron_Pik3ca_chr2	0,03
intergenic_AC139950.1|U6	0,02
intergenic_AC111678.2|AC111678.1_chr18	0,02
intergenic_Cxcl13|AABR07014424.1_chr14	0,01
intron_Mipep_chr15	0
intergenic_AABR07048878.2|AABR07048892.1_chr5	0
intron_Tmem97_chr10	0
intergenic_SNORD22|AABR07033162.1_chr11	0

In our experimental system, we electroporate dissociated neurons with gRNA and Cas9, so even if other intron or intergenic sequences are targeted by the Cas9 activity, this happens with very low probability thus the affected genes are likely to be different from one cell to the other, and from experiment to experiment. Moreover, neurons do not divide in culture so there is no possibility of clonal expansion of potential off-target mutated genes. Considering all the above-mentioned evidence, we think that even though the occurrence of off-target events cannot be completely ruled out, we are confident that our results come from the specific KO of MDGA1 and MDGA2. Of note is the rescue of the phenotype found in MDGA2 KO when we expressed a gRNA resistant variant of MDGA2 (Figure 8).

These results obtained from dissociated cultured neurons also contrast with Connors et al. 2017 Cell Reports, showing that MDGA1 specifically acts as a negative regulator at GABAergic hippocampal CA1 synapses in vivo.

Indeed, our results in dissociated cultures using single cell KO of MDGA1 by CRISPR/Cas9 show an enhancement of the density of excitatory post-synapses and the frequency of AMPARmediated mEPSCs (Figure 8). Our latest data obtained using organotypic slices confirm these results by showing that both CRISPR/Cas9 against MDGA1 and MDGA2 selectively enhance AMPAR-dependent synaptic transmission in CA1 neurons (Figure 9 and Figure 9 —figure supplement 1). Finally, we did not observe any change in either gephyrin puncta density or mIPSC frequency in neurons expressing gRNA to MDGA1, compared to neurons expressing control gRNA (Figure 1 —figure supplement 3), thereby not supporting a role of MDGA1 in inhibitory synapse differentiation during this developmental window (DIV 0-10). Taken together, our data are at odds with the study of Connor et al. (Cell Reports 2017), which shows a selective increase of inhibitory synaptic transmission in the hippocampus upon full MDGA1 KO.

The discrepancies between the two studies might come from the quite different experimental paradigms, i.e. single cell knock-down/out of MDGAs in neuronal cultures for a limited period of time, versus full KO of MDGAs in all cell types throughout the mouse lifetime. In the full KO, MDGA1 is silenced in all cell types, including both pyramidal neurons and astrocytes which also express MDGA1 (Author response image 5 and Figure 1 —figure supplement 1A) and NLGNs (Stogsdill et al., Nature 2017), which might overall cause mixed effects on hippocampal circuitry and increase inhibitory synaptic transmission. In this context, it would be interesting to generate MDGA1/2 Flox mice to be able to knock-out MDGAs for determined periods of time in single neurons or astrocytes using cre-recombinase, as done for NLGNs (Chanda et al., J. Neurosci 2017; Wu et al., Neuron 2019), and determine cell-specific effects of silencing MDGAs on synaptic differentiation. This is now mentioned in the discussion p. 14.

**Author response image 5. sa2fig5:** MDGA1 expression in astrocytes. (A) Live immunolabelling of endogenous MDGA1 (magenta) in astrocytes cultured for 28 DIV. Astrocytes were counterstained with GFAP (green) and DAPI (blue). Scale bars, 15 µm. (B) RT-qPCR data showing the relative expression of MDGA1 mRNA in hippocampal neurons at DIV 7, 14 and 21, and astrocyte cultures at DIV 28.

Reviewer #3:This work by Toledo and colleagues characterizes the synaptic localization of MDGA1/2 and their effects on Neuroligin 1 and AMPA receptor surface dynamics during development in dissociated neuronal cultures. MDGA1/2 are analyzed using KO, knockdown with molecular replacement, and CRISPR targeting. The data support that endogenous MDGA1 is partially synaptic but that MDGA1/2 are not enriched at excitatory synapses and become less abundant at excitatory postsynaptic sites as neurons mature. In agreement, they diffuse very fast in dendritic membranes. Detection of Homer density and tracking of Nlgn1 and GluA2 after MDGA2 knockdown support that both MDGAs (tested by CRISPR) restrict in young neurons synapse number and that MDGA2 loss reduces Nlgn1 and AMPA receptor surface mobility in more mature neurons. A role of MDGA2 in restricting Nlgn1 Tyr phosphorylation is shown as well. Functional effects on synapse maturation are measured by electrophysiological recordings in the cultures. In agreement with the Homer data representing an increase in functional synapse number, CRISPR targeting of MDGA1 or MDGA2 increased the frequency of mEPSCs. No effect on mEPSC amplitudes was found.Maturation-dependent changes in the localization and dynamics of MDGAs, Nlgn1, and AMPA receptors are well characterized, including by strong super-resolution imaging. The model that MDGAs serve as restrictive factors in synapse development through altering Nlgn1 mobility is interesting to gain new insights into the negative control of synapse assembly.Several points need to be addressed to further support the conclusions.1. The data support that MDGAs restrict the formation of excitatory synapses. The authors propose that this due to an effect on Neurexin/Neuroligin complex stability, as expected from the literature. Can the authors show that postsynaptic manipulation of MDGAs indeed alters the mobility of Neurexins and the extent of their presynaptic retention in maturing neurons?

We thank the reviewer for this interesting suggestion, which has indeed been part of our concerns. We have tried to detect changes in endogenous NRXN localization in axons facing MDGA KO dendrites but none of the antibodies tested (anti NRXN1β Abcam cat#77596 or anti pan-NRXN1 Millipore cat#ABN161-I) gave specific signals in our hands. Thus, we decided to detect NRXN by electroporating recombinant GFP-NRXN1β. The introduction of a relatively large GFP tag on the N-terminal part of NRXN1β does not alter its axonal function or binding to NLGN1, as previously demonstrated *(Schneider et al., J. Neurosci 2015; Lagardère et al., Sci Rep 2020; Klatt et al., Cell Reports 2021*). We co-plated the GFP-NRXN1β electroporated neurons with neurons electroporated with shCTRL or shMDGA2, both containing an EBFP2 reporter, plus a postsynaptic marker (Xph20-mRuby2). At DIV 10, we searched for GFP-NRXN1β positive axons contacting EBFP2-positive dendrites. Then, we analyzed the GFP-NRXN1β enrichment at axondendrite contact sites or we sparsely labelled it with an anti-GFP nanobody conjugated to the Atto647N fluorophore, to perform uPAINT as previously reported *(Chamma et al., Nat Commun 2016; Lagardère et al., Sci Rep 2020)*.

Our results do not show any change in NRXN1β enrichment at pre-synapses, nor a change in NRXN1β dynamics at the cell membrane (Figure 6—figure supplement 1). GFP-NRXN1β expressing axons contact several neurons, some of them expressing sh-EBFP2 and others not being electroporated. One explanation for this lack of change in global NRXN1β mobility is that the average number of GFP-NRXN1β presynaptic dots on axons is not increased in the shMDGA2 condition in such sparse electroporating conditions. On the other hand, at the level of individual synapses, we did not see an increase in the enrichment of NRXN1β, which is in accordance with the absence of AP-NLGN1 enrichment change found in shMDGA2 expressing neurons (Figure 6 A-C). Overall our results point to a global effect of MDGA KD/KO resulting in an increase in the total number of synapses in dissociated neurons, without affecting the protein enrichment/localization of either NRXN1β, NLGN1, or AMPARs at individual synapses (Figure 10, upper panel).

2. Why does targeting MDGA1 or 2 not alter synaptic strength measured as mEPSC amplitudes? Their role in controlling AMPA receptor mobility should predict that.

We appreciate this point raised by the reviewer, which apparently we failed to explicit well enough in the discussion or model schematics (Figure 10, upper panel). We explain the reduction in global AMPAR mobility observed upon MDGA2 KD in dissociated cultured neurons by an increase in the total number of synapses, not by an increase in the actual AMPAR content per synapse. Indeed, MDGA2 KD caused no change in AMPAR accumulation at each synapse (labeled by GluA1 antibody), and correspondingly no change in AMPAR-mediated mEPSC amplitude. A twofold reduction in global AMPAR diffusion was previously observed as a function of neuronal age in culture (from DIV 8-15), or upon NLGN1 over-expression, both of which resulting in a doubling of the number of synapses *(Czöndör et al. PNAS 2012)*. In these experimental situations, the AMPAR mEPSC frequency strongly increased, but the mEPSC amplitude stayed constant *(Chanda et al. J. Neurosci 2017; Letellier et al. Nat Commun 2018*). Thus, these three experimental paradigms (neuronal maturation, NLGN1 OE, and MDGA2 KD) all increase the number of synapses, but not the actual content of AMPARs per synapse. It seems therefore that AMPARs come to individual synapses as quantal packets, potentially in the form of nanodomains, associated to the post-synaptic scaffold, as previously reported *(McGillavry et al. Neuron 2013; Nair et al., J. Neurosci 2013)*.

In relation to this point (also motivated by the response to reviewer 2), we now provide additional electrophysiology data obtained in organotypic slices, where we knocked out MDGA1 or MDGA2 in single CA1 neurons, and recorded both AMPAR- and NMDAR-mediated EPSCs upon stimulation of Schaffer’s collaterals, normalized to values measured simultaneously in neighboring non-electroporated neurons. Strikingly, both gRNAs to MDGA1 and MDGA2 increased AMPARmediated EPSCs without affecting NMDAR-mediated EPSCs, as compared to non-electroporated controls (Figure 9B-D and Figure 9—figure supplement 1A,B). As a result, the ratio between AMPAR- and NMDAR-mediated EPSCs was significantly elevated in both MDGA1 and MDGA2 KO neurons (Figure 9E), supporting a synaptic unsilencing mechanism (Figure 10, lower panel). No effect of CRISPR to MDGA1 or MDGA2 on the paired pulse ratio was observed (Figure 9 —figure supplement 1C,D), ruling out pre-synaptic mechanisms. Together, those results strengthen the concept that both MDGA1 and MDGA2 down-regulate AMPAR recruitment during excitatory synapse development, although the mechanisms might be slightly different between dissociated and organotypic cultures. These points are discussed p. 15.

[Editors’ note: what follows is the authors’ response to the second round of review.]

Essential revisions:1. Rescue experiments for MDGA2 knockdown: there are cases where rescue effects were not clear (e.g. Figure 5F). Statistics should be performed between shMDGA2 and shMDGA2+res to ensure significance in this comparison.

In Figure 5F, the average median diffusion coefficient is 0.021 µm²/s for shCTRL, 0.0098 µm²/s for shMDGA2, and 0.0148 µm²/s for shMDGA2 + rescue MDGA2 which lies between the two other conditions. One can see from the average distribution in Figure 5E that the shMDGA2 reduces the peak of mobile NLGN1 molecules compared to the two other conditions. But statistically, there is only a significant difference between shCTRL and shMDGA2 conditions (P < 0.05) and not between shMDGA2 and rescue MDGA2, or between shCTRL and rescue MDGA2. This is likely due to the smaller data sample (n = 9 neurons) in the shMDGA2 + rescue MDGA2 condition, associated to the variable degree of expression level of the rescue MDGA2 construct. Indeed, the standard deviation is 0.006 µm²/s for the shMDGA2 data while it is 0.011 µm²/s for sh + rescue MDGA2, i.e. almost double. The difference in sample size comes from the difficulty to find neurons expressing the three plasmids simultaneously (shRNA + post-synaptic marker + rescue construct).

2. Neuron morphologies: the authors argue that the cultures were relatively of early stages for some images. However, it is difficult to discern clear dendritic spine structures in many fluorescent images obtained from DIV14 cultures (e.g., Figure 5—figure supplement 1). This is still a lingering problem.

We agree with the reviewer that the dendritic segment chosen to illustrate the shMDGA1 condition at DIV14 did not exhibit very clear dendritic spines. We chose another image of this condition, with nice spines bulging out of the dendrite shaft, all containing Homer1c-dsRed positive puncta (new Figure 5 —figure supplement 1). The other panels look good to us.

3. Validation of CRISPR knockout approach: the authors nicely provided with convincing in silico evidences, arguing against potential no off-target effects. However, the authors need to provide data (e.g. qPCRs), showing no changes in expression levels of some neuronal genes from cultured neurons or N2a cells expressing sgMDGA1 or sgMDGA2.

We thank the reviewer for this interesting remark, which stimulated us to perform a thorough patchseq analysis of neurons electroporated with CRISPR MDGA1/2 constructs. It took us a while to perform these experiments and analysis, thereby explaining the reason why we come back quite late with this revised manuscript. Because there are a variety of off-target genes potentially recognized by MDGA1 or MDGA2 gRNAs and consequently excised by the Cas9 enzyme, we turned to global RNA sequencing instead of classical qRT-PCR. Moreover, because of the large size of the plasmids used (Cas9 + gRNA + GFP reporter), the fraction of electroporated neurons can be quite modest in dissociated cultures. Therefore, to make sure of the CRISPR effects in those cells, we decided to isolate a limited number of GFP-positive neurons by patch-clamp, extract their RNA, and perform RNA-sequencing, as previously described (*Fusik et al., Nat Biotech 2015; Cadwell et al., Nat Biotech 2015*). The results are shown in new Figure 8 – figure supplement 2.

First, we checked that all patched neurons contained detectable mRNA levels of the GFP reporter, validating the single-cell mRNA extraction procedure (Figure 8 – figure supplement 2A). Second, MDGA1 and MDGA2 mRNA levels were selectively reduced by CRISPR MDGA1 and CRISPR MDGA2, respectively, compared to the CRISPR CTRL condition where those two transcripts are well detected (Figure 8 – figure supplement 2B,C). These results confirm for MDGA1 the loss in protein level seen previously (Figure 8 —figure supplement 1B,C), and establishes the efficacy of MDGA2 KO by CRISPR at the mRNA level. Third, to address potential off-target effects, we performed a principal component analysis (PCA) of the top 200 variable genes expressed in all neurons, which shows that genes are not particularly affected by CRISPR MDGA1 or MDGA2 compared to CRISPR CTRL (Figure 8 – figure supplement 2D). Third, we plotted the mRNA levels of the genes detected in neurons expressing CRISPR MDGA1 or CRISPR MDGA2, versus the same genes measured in neurons expressing CRISPR CTRL (Figure 8 —figure supplement 2E). The data are tightly correlated (r > 0.7) , giving no evidence for off-target genes that would be selectively affected by MDGA1 or MDGA2 KO. Finally, we analyzed a subset of genes theoretically predicted to be potential off-target genes of CRISPR MDGA1 and CRISPR MDGA2 (i.e. with > nucleotide 4 mismatches) (Tables 1 and 2), and found no significant difference in mRNA levels when compared to neurons expressing CRISPR CTRL (Figure 8 —figure supplement 2F,G). All this evidence strengthens the selectivity of our CRISPR MDGA1 and MDGA2 strategy.

4. Interpretation of Figure 6—figure supplement 1: the reasonable conclusion seems to be that MDGA2 is NOT required for Nlgn1/Nrxn1beta contact. Please discuss.

We added a sentence in the Results section p. 9 under the paragraph entitled “MDGA knock-down enhances NLGN tyrosine phosphorylation”. The end of the paragraph now reads : “Even though these results might suggest that MDGA KD does not directly affect the trans-synaptic NRXN1β-NLGN interaction, we have to moderate this explanation by considering that GFP-NRXN1β expressing axons make simultaneous contacts with dendrites from many neurons, such that the effect of shMDGA2 in sparsely electroporated cells is diluted”. Other possibilities are given in the discussion p. 15, as mentioned in our response to point 6 below.

5. Discussion: The authors' argument is not convincing about adhesion-dependent competition between neurons with citing Kwon et al. 2012. Indeed, Sudhof and colleagues later demonstrated in cultured neurons that sparsely knocking out Nlgn1 did not produce any phenotypes in synapse numbers (Chandra et al., 2017 J Neurosci). The authors need to consider revising the Discussion section to reconcile the discrepancy with previous observations.

We modified the sentence in the discussion p. 14 to address this controversial point. It now reads : “A difference in the results obtained by global gene knock-out in mice and sparse single-neuron silencing was also seen for NLGN1 (*Chih et al., 2005; Varoqueaux et al., 2006*), suggesting that adhesiondependent competition between neurons regulates synaptogenesis (*Kwon et al., 2012*), although sparsely knocking out NLGN1 did not reduce synapse number in another study (*Chanda et al., 2017*)”.

6. The authors could discuss in the manuscript the new data in Figure 6 – figure supplement 1. While they do not show an effect of MDGA knockdown on Nrx1beta, future studies can test whether the Nrx overexpression approach obscures effects of MDGA knockdown, or whether MDGA knockdown preferentially affects Nrx1alpha. The added discussion points provide interesting ideas into developmental mechanisms controlling MDGA functions.

We thank the reviewer for this interpretation and added a sentence in the discussion about these points (p. 15). It now reads : “In this context, we were surprised not to observe any impact of MDGA KD on the global diffusion or pre-synaptic accumulation of GFP-NRXN1β in axons contacting neurons expressing shMDGA2, but these effects might have been obscured by the limited number of contacts between these cells or by NRXN1β over-expression. Future single molecule tracking experiments on NRXN1-α or -β expressed at endogenous levels may allow one to clarify this issue (*Klatt et al., 2021; Neupert et al., 2015*).”